# Streamflow forecast sensitivity to air temperature forecast calibration for 139 Norwegian catchments

Trine J. Hegdahl[1], Kolbjørn Engeland[1,2], Ingelin Steinsland[3], Lena M. Tallaksen[2]

[1]Norwegian Water Resources and Energy Directorate, Hydrological Modelling, 0301 Oslo, Norway
[2]University of Oslo, Department of Geosciences, 0316 Oslo, Norway
[3]Norwegian University of Science and Technology, Department of Mathematical Sciences, 7034 Trondheim, Norway

*Correspondence to*: Trine J. Hegdahl (tjh@nve.no)

**Abstract.** In this study, we used meteorological ensemble forecasts as input to hydrological models to quantify the uncertainty
in forecasted streamflow, with a particular focus on the effect of temperature forecast calibration on the streamflow ensemble
forecast skill. In catchments with seasonal snow cover, snowmelt is an important flood generating process. Hence, high quality
air temperature data are important to accurately forecast streamflows. The sensitivity of streamflow ensemble forecasts to the
calibration of temperature ensemble forecasts was investigated using ensemble forecast of temperature from ECMWF covering
a period of nearly three years, from 01.03.2013 to 31.12.2015. To improve the skill and reduce biases of the temperature
ensembles, the Norwegian Meteorological Institute provided parameters for ensemble calibration, derived using a standard
quantile mapping method where Hirlam, a high resolution regional weather prediction model, was used as reference. A lumped
HBV model, distributed on 10 elevation zones, was used to estimate the streamflow. The results show that temperature
ensemble calibration affected both temperature and streamflow forecast skill, but differently depending on season and region.
We found a close to 1:1 relationship between temperature and streamflow skill change for the spring season, whereas for
autumn and winter large temperature skill improvements were not reflected in the streamflow forecasts to the same degree.
This can be explained by streamflow being less affected by sub-zero temperature improvements, which accounted for the
biggest temperature biases and corrections during autumn and winter. The skill differs between regions. In particular, there is
a cold bias in the forecasted temperature during autumn and winter along the coast, enabling a large improvement by
calibration. The forecast skill was partly related to elevation differences and catchment area. Overall, it is evident that
temperature forecasts are important for streamflow forecasts in climates with seasonal snow cover.

## 1    Introduction

Floods can severely damage infrastructure, buildings and farmland, and can have high economic impacts on society
(Dobrovičová et al., 2015). Early warnings based on hydro-meteorological forecasts are an important flood mitigation measure
and provide time to reduce flood damage. A flood forecasting system consists of a hydro-meteorological forecasting chain

with three main components, all affected by uncertainties: (i) observations used to establish the initial conditions for the catchment, (ii) meteorological forecasts used as forcing, and (iii) the hydrological model.

The Norwegian flood forecasting system, operated by the Norwegian Water Resources and Energy Directorate (NVE), uses deterministic forecasts of air temperature and precipitation as forcing for hydrological models in 145 catchments across the country. Meteorological forecasts from the AROME-MetCoOp operational weather prediction model (Müller et al., 2017) are used for short range forecasts (day 1 and 2), whereas forecasts from the European Centre for Medium-Range Weather Forecasts (ECMWF, 2018a) high resolution model are used for medium range forecasts (day 3 to 9). All forecasts are provided by the Norwegian Meteorological Institute (MET Norway). The Hydrologiska Byråns Vattenbalans model (HBV) (Bergström, 1976; Sælthun, 1996; Beldring, 2008) is used as the hydrological forecasting model, which combined with statistical uncertainty models (Langsrud et al., 1998a; Langsrud et al., 1998b), provides probabilistic streamflow forecasts. The uncertainty model accounts for the strong autocorrelation in forecast errors and estimates an uncertainty band around the deterministic temperature, precipitation and streamflow forecasts.

An alternative approach to estimate probabilistic streamflow forecasts is to use meteorological ensemble forecasts from numerical weather prediction models as a means to account for uncertainty in the forcing. The meteorological ensemble forecasts are created by perturbing both the initial states and the physics tendencies of the original deterministic forecast. The spread of the ensemble members can be interpreted as the uncertainty of the forecasts, where a large spread indicates large uncertainty (Buizza et al., 1999; Persson, 2015). Subsequently, the meteorological ensemble is used as forcing for a hydrological model to produce an ensemble of forecasted streamflow, referred to as a hydrological ensemble prediction system (HEPS). HEPS are increasingly being used in flood forecasting (Cloke and Pappenberger, 2009; Wetterhall et al., 2013). A HEPS adds value to a flood forecast by assessing the forecast uncertainty caused by uncertainties in one or several parts of the modelling chain.

Raw (unprocessed) ensembles are rarely reliable in a statistical sense (Buizza, 1997; Wilson et al., 2007). Reliability means that the observation behaves as if it belongs to the forecast ensemble probability distribution (Leutbecher and Palmer, 2008). To improve reliability, the ensemble forecasts can be calibrated by applying statistical techniques correcting bias and under/over-dispersion (Hamill and Colucci, 1997; Persson 2015). Examples of methods used to calibrate meteorological ensembles include ensemble model output statistics (EMOS) (Gneiting et al., 2005; Wilks and Hamill, 2007), Bayesian model averaging (BMA) (Raftery et al., 2005; Wilson et al., 2007), ensemble Kalman filters (Evensen, 2003; Verkade et al., 2013), non-homogenous Gaussian regression ( Gneiting et al., 2005; Wilks and Hamill, 2007), quantile mapping (Bremnes, 2007), and kernel dressing (Wang and Bishop, 2005). These methods differ in their sensitivity to length of training data and ensemble size, and how the spread and bias are corrected. Pre-processing (from a hydrological perspective) refers to all techniques used to change the output from a meteorological model, and includes calibration (described above) and downscaling. Downscaling implies resampling from the original forecast grid size to a grid of higher resolution, and both statistical (e.g. interpolation) and dynamical (e.g. a regional weather forecast model) techniques can be used (Schaake. et. al., 2010). A recent review of pre-processing methods are given in Li et al (2017) and the textbook edited by Vannitsem et al (2018).

In climates with seasonal snow cover, snowmelt during the spring season is an important flood-generating process. In these climates, temperature is a key variable to classify the precipitation phase and to estimate the snowmelt rate. The sensitivity of daily streamflow to temperature is non-linear since streamflow depends on temperature thresholds for rain/snow partitioning and for snow melt/freeze processes. The latter depends on the state of the system i.e. snow is needed to generate snowmelt.

For temperatures well below zero degrees, the streamflow is not sensitive to temperature, whereas for temperatures around zero degrees relatively small changes in temperature might control if precipitation falls as rain or snow, and consequently whether streamflow is generated or not. Most Norwegian catchments experience a seasonal snow cover, but are otherwise diverse in terms of the length of the snow season and topographic complexity (Rizzie et al., 2017).

Downscaling and interpolating air temperature in complex topography are both challenging, mostly because temperature lapse
rates depend on several factors, i.e. altitude, time and place, as well as specific humidity and air temperature (Aguado and Burt, 2010; Pagès and Miró, 2010; Sheridan et al., 2010). Errors in forecasted temperature might result in a misclassification of precipitation phase and/or cause the hydrological forecasting system either to miss a flood event or provide a false alarm, caused by too high or too low snowmelt rates. It is therefore important to assess the relationship between temperature and streamflow forecasts. The importance of reliable temperature forecasts for streamflow forecasts is demonstrated for two Alpine
catchments during a heavy precipitation event in Ceppi et al. (2013). An interesting finding in this paper is that catchment elevation distribution, and by this area above the snowline, was important for how streamflow forecasts were affected by temperature uncertainty. Verkade et al. (2013), on the other hand, found only modest effects of temperature calibration on streamflow forecast skill as an average over several years for Rhine catchments.

As far as the authors know, the isolated effect of the uncertainties in temperature forecasts has not yet been systematically
investigated for a larger number of catchments in a cold climate. The large spatial and seasonal variations in snow accumulation and snowmelt processes found in cold regions with complex terrain require that both spatial and seasonal patterns in the performance of temperature and streamflow forecasts are evaluated.

The main objective of this study is to investigate the effect of temperature forecast calibration on the streamflow ensemble forecasts skill in catchments with seasonal snow cover, and to identify potential improvements in the forecasting chain. In
particular, we address the following research questions:
- Are there seasonal effects of temperature calibration on the temperature ensemble forecast skill?
- Are there seasonal effects of temperature calibration on the streamflow ensemble forecast skill?
- Are there spatial patterns in the temperature and streamflow ensemble forecast skill and if so, can these be related to catchment characteristics?
To answer these questions, we applied temperature ensemble forecasts from ECMWF combined with the pre-processing setup from MET Norway, to 139 catchments in Norway. Three years of operational ECMWF forecasts from 2013-2015 were used to re-generate streamflow forecasts, and the skill of temperature and streamflow forecasts were systematically evaluated for these catchments. To investigate the isolated effect of the temperature ensembles on the streamflow forecasts, the observed SeNorge precipitation (Tveito et al., 2005) was used instead of the precipitation ensemble forecasts, to run the hydrological

model. Finally, a case study is presented, demonstrating the effect of temperature calibration on a single snowmelt induced flood event. We start by presenting the study area, data and hydrological model (HBV) used (Sect. 2). In Sect. 3, methods used to establish the hydro-meteorological forecasting chain, the skill metrics and evaluation strategy are presented. Section 4 contains the results, followed by a discussion in Sect. 5. Finally, in Sect. 6, the findings are summarized, conclusions are drawn, and further research questions are discussed.

## 2     Study area, data and model

### 2.1     Study area

In Norway there are spatial variations in climate and topography, and a recent overview of past, current and future climate is given in Hanssen-Bauer et al. (2017). The western coast has steep mountains, high annual precipitation (4000-5000 mm/year) and a temperate oceanic climate. Inland areas have less precipitation, larger differences between winter and summer temperatures, and climatic zones from humid continental, to subarctic and mild tundra (according to the Köpper-Geiger system, see (Peel et al., 2007)). The mean annual runoff follows to a large degree the spatial patterns of precipitation. The two basic flood generating processes are snowmelt and rainfall (Vormoor et al., 2015). Most catchments in Norway have prolonged periods of sub-zero temperatures during winter, resulting in a seasonal snow storage, winter low flow, and increased streamflow during spring due to snowmelt. The relative importance of rainfall and snowmelt processes are decided by the duration of the snow accumulation season and the share of annual precipitation stored as snow. Across Norway two basic runoff regimes can be identified, (i) coastal regions with high flows during autumn and winter due to heavy rainfall and (ii) inland regions with high runoff during spring due to snowmelt (Vormoor et al., 2015). However, there are many possible transitions between these two basic patterns (Gottschalk et al., 1979).

The national flood-forecasting system builds on hydrological models providing streamflow forecasts in 145 catchments, covering most parts of Norway, varying in size (~3 to 15447 km$^2$) and elevation difference (103 to 2284 m). The latter is calculated as the difference between the lowest and highest point on the hypsographic curve, $\Delta H = (H_{100} - H_0)$. The flood forecasting catchments are mostly pristine, although some do have minor (hydropower) regulations. Fourteen catchments have a glacier coverage of 5 % or more. Of the 145 flood forecasting catchments, 139 were chosen as the basis for the study (Fig. 1). The catchments were grouped into five *regions* based on their location; North (N), South (S), West (W), Mid (M), and East (E) following Hanssen-Bauer et al. (2017) and Vormoor et al. (2016) (Fig. 1, right). These regions are defined by the boundaries of the major watersheds, and reflect major hydro-climatological zones. Rainfall floods dominate in South, West, and Mid, whereas snowmelt floods dominate in East and North. There is still a large variability in hydrological regimes within individual regions. Figure 1 includes the location of four catchments, for which results that are more detailed will be presented. Gjuvaa (E), Foennerdalsvatn (W) and Viksvatn (W) were used to visualize the challenges in temperature forecasts, and both uncalibrated and calibrated ensemble values will be presented for these three catchments. Viksvatn (W) and Foennerdalsvatn (W) are located in Western Norway and are both catchments with some glaciers (~3 % and 47 % respectively). Gjuvaa (E) is

non-glaciered and located inland (Fig. 1, left). The Bulken (W) catchment was chosen to demonstrate the effect of temperature calibration on the streamflow forecast for a snowmelt driven flood event.

## 2.2 Observations, hydrological model and forecasts

### 2.2.1 Interpolated precipitation and temperature observations– SeNorge data

In Norway, a network of about 400 precipitation stations and 240 temperature stations provides daily temperature and precipitation values. These *in situ* observations are interpolated to create a gridded ($1\times1$ km$^2$) product, referred to as *SeNorge* (available at SeNorge.no, Tveito et al., 2005). In this study, we used version 1.1. For this version, gridded temperature is calculated by kriging, where both the elevation and location of temperature stations are accounted for. The observed daily precipitation is corrected for under-catch at the gauges, and triangulation is used for spatial interpolation to a $1\times1$ km$^2$ grid. A constant gradient of 10 % per 100 m beneath 1000 meter above sea level (masl) and 5% per 100 m above 1000 masl is applied to account for elevation gradients in precipitation (details can be found in Tveito (2002), Tveito et al. (2005), and Mohr (2008)). The SeNorge data are available from 01.01.1957, and in this study, we used data for the period 01.03.2013 to 31.12.2015 in the forecasting mode and 01.01.1958 to 31.12.2012 to calculate the temperature and streamflow climatology (Sect. 3.2). The SeNorge precipitation substitute the precipitation forecasts in the ensemble forecasting chain, and hence the isolated effect of temperature calibration on streamflow forecasts was obtained. We hereby denote SeNorge temperature and precipitation, $T_{o[lat, lon, t]}$ and $P_{o[lat, lon, t]}$ respectively, where $t$ is an index for observation time. Latitude (*lat*) and longitude (*lon*) represent the grid indexing.

### 2.2.2 Hydrological model – HBV

The HBV model (Bergström, 1976) as presented in Sælthun (1996) and Beldring (2008) constitutes the basis for this study. The vertical structure of the HBV model consists of a snow routine, a soil moisture routine, and a response function that includes a nonlinear reservoir for quick runoff and a linear reservoir for slow runoff. The model uses catchment average temperature and precipitation as input. Each catchment is divided into 10 elevation zones, each covering 10% of the total catchment area. The catchment average precipitation and temperature are elevation adjusted to each elevation zone using catchment specific lapse rates. In this study, we used the operational model set-up which has been calibrated, for each catchment individually, using the PEST software for parameter estimation (Doherty, 2015 ), with Nash-Sutcliffe (Nash and Sutcliffe, 1970) and volume bias as calibration metrics. The calibration, 1996-2012, gives mean Nash-Sutcliffe 0.77, with zero volume bias for the 139 catchments. The validation period, 1980-1995, shows mean Nash-Sutcliffe 0.73, with a mean volume bias of 5% (Gusong , 2016). We used one optimal parameter set for each catchment and ignored therefore uncertainty arising from parameter estimation and the hydrological model.

### 2.2.3    Reference Streamflow

Reference streamflow, $Q_{o(c,t)}$, where $c$ is an index for catchment, was derived using SeNorge precipitation and temperature, aggregated to the catchment scale, as forcing to the HBV model (Fig. 2, see "Reference mode" in the green frame). In order to isolate the effect of temperature calibration on forecasted streamflow and avoid effects of hydrological model deficiencies, reference streamflow was used as a benchmark when the streamflow forecasts were evaluated. Similarly, operational flood warning levels (here demonstrated for the case study basin, Bulken), are based on return-periods from reference streamflow.

### 2.2.4    Temperature ensemble forecasts

We used the ECMWF temperature forecast ensemble (ENS) for the period 01.03.2013 to 31.12.2015 from an original grid resolution of 0.25° (i.e. model cycles/versions 38r1/2, 40r1, and 41r1 (ECMWF, 2018b)). This period covers model cycles/versions for which temperature grid calibration parameters are trained (40r1 and 41r1, see section 3.1.2) plus spring 2013 (cycle 38r1/2) in order to include one more snowmelt season. In short, 50 ensemble members of ENS are generated by adding small perturbations to the forecast initial conditions and model physics schemes, subsequently running the model with different perturbed conditions. The ensemble represents the temperature forecast uncertainty. A more detailed description of the ECMWF ENS system is provided in e.g. Buizza et al.(1999, 2005), and Persson (2015). For each issue date $d$, 51 ensemble members $T_{ens[lat,\ lon,\ m,\ l*]}$ are provided for a lead time up to 246 hours, where $m$ is the ensemble member and $l^*$ the lead time in 6 hours intervals. In this study, we used the forecasts issued at 00:00 and aggregated daily values for the meteorological 24-hour period defined as 06:00-06:00 to provide forecast for lead times up to nine days. The observational time $t$ for a forecast is $d + l^*$.

## 3    Methods

### 3.1    Ensemble forecasting chain

Figure 2 shows the forecasting modelling chain designed for this study. The green frame presents the observational reference mode that determines the internal states for the forecasting issue date, $d$, in the red frame. This reference mode was also used to estimate reference streamflow $Q_{o[c,t]}$ (see Sect. 2.2.3). SeNorge temperature and precipitation ($T_{o[c,t]}$ and $P_{o[c,t]}$), aggregated to each catchment $c$, were used to force the hydrological model in the observational reference mode. The red frame illustrates the forecasting mode, including the post-processing of temperature forecasts. The hydrological ensemble forecasts were estimated using downscaled raw temperature ensemble forecasts ($T_{ens[c,m,l]}$, see Sect. 3.1.1) or downscaled and calibrated temperature ensemble forecasts ($T_{cal[c,m,l]}$, see Sect. 3.1.2) and observed precipitation ($P_{o[c,\ d+1]}$) as forcing, where $m$ is ensemble member and $l$ is lead time in days. All temperature forecasts were aggregated to daily time steps since the operational HBV model runs on a daily time step and the SeNorge data used as a reference provide only daily values. In the forecasting

mode, each temperature ensemble member was used as input and run as a separate deterministic forecast. All hydrological forecasts were estimated for all 9 lead times. Note that for each issue date $d$, the same internal states of the HBV model were used for all ensemble member runs. Thus two sets of streamflow ensemble forecasts ($Q_{ens[c,m,l]}$ and $Q_{cal[c,m,l]}$) that differ only by the applied temperature calibration, were derived. The following subsections provide details on the approach used for

downscaling and calibration of the ensemble temperature forecasts (ENS).

### 3.1.1  Temperature forecast downscaling

In this paper the term downscaling refers to the interpolation of temperature from a low resolution grid to a high resolution grid where vertical temperature gradients are accounted for. The ECMWF grid temperature, which represent the average temperature for the grid cell, was interpolated from a horizontal resolution of 0.25° (~ 30 km) to the 1×1 km² SeNorge grid,

using the nearest neighbour method and aggregated to daily values to match the spatial and temporal resolution of the SeNorge data. Due to elevation differences between the ECMWF and SeNorge grid elevations, we corrected the ensemble temperature at the 1×1 km² scale by applying a standard atmospheric lapse rate of -0.65 °C/100 m. Finally, the downscaled temperature ensemble was aggregated to daily values and averaged over the catchment areas to provide $T_{ens[c,m,l]}$ for a given lead time and ensemble member.

### 3.1.2  Temperature grid calibration

The grid temperature is calibrated using quantile mapping (Seierstad, 2016; Bremnes, 2007) to remove biases by moving the ENS forecast climatology closer to the observed climatology. MET Norway provided temperature grid calibration parameters used in this study. This grid calibration was used in the operational post-processing chain for meteorological forecasts including the forecasts published on yr.no. MET Norway uses Hirlam (Bengtsson et al., 2017) temperature forecasts (on a 4×4 km²) to

provide a reference for parameter estimation (calibration). Hirlam is suitable as a reference since it provides a continuous field covering all of Norway at a sub daily time step. In addition, Hirlam gives higher skill and is less biased than ENS (Engdahl et al., 2015). To establish the calibration parameters MET Norway used both ENS reforecasts (Owens, 2018) and Hirlam data from July 2006 to December 2011 interpolated to a 5×5 km² grid. The ENS reforecast is a 5 member ensemble generated from the same model cycle (40r1 and 41r1) as ENS. For each grid cell, monthly unique quantile transformation coefficients are

determined by using data from a three-month window centred on the target month, e.g. the May analysis consists of April, May and June (Seierstad, 2017). The same coefficients, based on mapping the first 24 hours, were applied to all lead times and members. For forecasts outside the observation range, a 1:1 extrapolation was used. I.e. if a forecast is 2°C higher than the highest mapped forecasted temperature, then the calibrated forecast is 2°C higher than the highest mapped reference temperature.

For this study, we applied the calibration coefficients provided by MET Norway to the temperature forecasts for the period 2013-2015. Accordingly, ENS was interpolated to the 5×5 km² grid for which the quantile mapping coefficients were used to

obtain the calibrated temperature ensembles ($T_{cal}$). Subsequently, the calibrated ensembles on the 5×5 km$^2$ grid were downscaled to the 1×1 km$^2$ grid following the same procedure as for the uncalibrated temperature ensemble ($T_{ens}$, Sect. 3.1.1). Finally, the calibrated temperature ensemble was aggregated to daily values and averaged over the catchment areas to provide $T_{cal[c,m,l]}$.

## 3.2    Validation scores and evaluation strategy

The evaluation focused on the performance of the temperature forecast ensembles, and the effect of both uncalibrated and calibrated temperature forecasts on the performance of the streamflow ensembles. A well performing ensemble forecast should be reliable and sharp, where reliability has the first priority (Gneiting et al., 2007). A forecast is considered reliable if it is statistically consistent with the observed uncertainty, i.e. 90% of the observations should verify within the 90% forecast interval. Rank-histograms are often used for visual evaluation of reliability, and show the frequencies of observations amongst ranked ensemble-members. For reliable ensemble forecasts, the rank-histogram will be uniform (horizontal). A bias in the ensemble forecast is recognized as a slope in the rank-histogram, where a negative slope indicates too warm temperature forecasts, and  positive slope too cold forecasts. A U-shape indicates that the ensemble forecast is under-dispersed, whereas a convex shape indicates over-dispersion (Hamill, 2001). In order to quantify the reliability, a decomposition of the chi-square test statistics for the rank-histogram was used to describe the rank-histograms slope (bias) and convexity (dispersion) (Jolliffe and Primo, 2008). Both rank-histogram slope and convexity are negatively oriented, i.e. lower values are better, with an optimal value of zero for un-biased and uniformly distributed data. The sharpness of a reliable forecast is described by the spread between the ensemble members, where a sharp forecast has a small spread and is the most useful (Hamill, 2007). In this study, the temperature sharpness was assessed by first estimating the range between the 5$^{th}$ and the 95$^{th}$ percentile of the ordered ensemble forecasts for all issue dates, lead times and catchments. For streamflow, we estimated a relative sharpness by dividing the 5$^{th}$ to 95$^{th}$ percentile range by the ensemble mean. Thereafter, sharpness was determined for each catchment and lead time as the average range of all issue dates. The continuous rank probability score (CRPS) is a summary of reliability, sharpness and uncertainty (Hersbach, 2000). CRPS (denoted as $S_{CRP}$ in Eq. 1) measures the distance between the observation $x_a$ and the ensemble forecast, where the latter is expressed by the cumulative density function $F_x(x)$:

$$S_{CRP}(F_x, x_a) = \int_{-\infty}^{\infty}[F_x(x) - H(x - x_a)]^2 dx, \tag{1}$$

where $H$ is the Heaviside function that is zero when the argument is less than zero, and one otherwise (Hersbach, 2000). $\overline{CRPS}$ was calculated as the average CRPS ($S_{CRP}$) over the study period (01.03.2013 to 31.12.2015). $\overline{CRPS}$ is similar to the mean absolute error for deterministic forecasts. The temperature $\overline{CRPS}$ was computed using the SeNorge temperature $T_o$, as observations, whereas streamflow $\overline{CRPS}$ used $Q_{o[c,t]}$ as observations. This evaluation approach allowed us to evaluate the isolated effect of the uncertainties in the temperature forecasts since we can then, to a large degree, ignore uncertainties in the HBV model itself.

Skill scores are convenient for comparison between forecast variables (e.g. temperature versus streamflow) and catchments since these scores are dimensionless. To calculate the continuous ranked probability skill score (CRPSS denoted as $S_{CRPS}$ in Eq. 2), a benchmark score ($\overline{CRPS}_B$ denoted as $\bar{S}_{B\_CRP}$) which a skilful forecast score ($\overline{CRPS}_F$ denoted as $\bar{S}_{F\_CRP}$) should outperform, is needed. For both temperature and streamflow, ensembles representing daily climatology were used as

benchmarks. Daily SeNorge temperature ($T_{o[c,t]}$) from 1958 to 2012 (i.e. 55 years) were used to create a climatological temperature ensemble of 55 members for each day of the year. Similarly, a daily streamflow climatology was established from reference streamflow ($Q_{o[c,t]}$) calculated by the HBV model, forced with the 55 years of temperature and precipitation ($T_{o[c,t]}$ and $P_{o[c,t]}$) from the SeNorge data.

CRPSS ($S_{CRPS}$) was calculated for each catchment according to Eq. (2) (Hersbach, 2000).

$$S_{CRPS} = \frac{\bar{S}_{B\_CRP} - \bar{S}_{F\_CRP}}{\bar{S}_{B\_CRP}}, \tag{2}$$

CRPSS varies from $-\infty$ to 1, where one is a perfect score. Negative values mean that the forecast performs worse than climatology, and CRPSS equal to zero implies that it performs similarly to the benchmark (climatology in this case). The seasonal skill score was calculated by averaging the daily CRPS only for the months belonging to the target season. The effect of the grid calibration on the temperature and streamflow forecast skill was evaluated by comparing the validation scores using both the uncalibrated ($T_{ens}$) and the calibrated ($T_{cal}$) ensembles to generate the streamflow ensembles. For readability, the

abbreviations $S_{CRP}$ and $S_{CRPS}$ used in the equation will be substituted with CRPS and CRPSS in the text hereafter.

*Spatial* patterns in the forecast performance for all 139 catchments, i.e. CRPSS and differences in CRPSS between calibrated and uncalibrated temperature, were mapped for Norway. Further, box plots for the five regions (see Fig. 1) were drawn to reveal potential regional patterns. Finally, we used linear regression, to identify relationships between catchment characteristics (*elevation difference* and catchment *area*) and the skill score ($T_{cal}$ and $Q_{cal}$ CRPSS). The linear regression analysis was done

for combinations of seasons and regions. S*easonal* variations in skill score were assessed by calculating CRPSS for the two seasons, spring (April to June) and autumn (October to December). This definition of seasons is used to better capture a snowmelt season, which for most Norwegian catchments is in the period April to June. For this paper, we chose to focus on the results for autumn and spring. Summer (July to September) was excluded due to the relatively small changes in CRPSS explained by (i) the skill of uncalibrated temperature forecasts is higher and the potential for improvement is lower, and (ii)

there is less or no snow in summer, resulting in a reduced streamflow sensitivity to temperature. Winter (January to March) was excluded since it performs similarly to autumn.

Finally, the effect of temperature calibration on the flood warning level is illustrated for a snowmelt induced flood event in the Bulken catchment. In the operational flood warning system at NVE, the predefined flood thresholds are catchment specific and calculated return-periods are based on reference streamflow, which is also the approach used herein.

## 4 Results

Temperature and streamflow forecasts were estimated for 139 catchments, 1036 issue dates and 9 lead times. Figure 3 presents a summary of the validation scores, CRPSS and the rank-histogram decomposition, in addition to sharpness, for all lead times. Each box plot shows the variations in the validation scores between the catchments. The rank-histogram slope and convexity describes bias and dispersion in the forecasts, respectively, both can be considered a measure for the reliability. As shown in Fig. 3, temperature slope and convexity improve with increasing lead time, whereas CRPSS and sharpness get poorer. For streamflow, slope and sharpness get poorer, convexity improves, whereas CRPSS shows small changes with lead time. To reduce the amount of presented results, the remaining part of this paper focuses on CRPSS for a lead time of 5 days. CRPSS was the chosen validation score since it contains information of reliability, uncertainty and sharpness, and enables a comparison between catchments. A lead time of 5 days was chosen since reliability (convexity and slope) has improved and some sharpness is maintained, i.e. too large ensemble spread will increase the reliability but the forecast value will be reduced.

### 4.1 Temperature forecasts

Time series of SeNorge daily temperature $T_o$, the range of raw (uncalibrated) temperature ensembles $T_{ens}$ (left panels), and scatter plots of ensemble mean for both raw $T_{ens}$ and calibrated $T_{cal}$ versus $T_o$ (right panels) are shown for three selected catchments in Fig. 4. For Gjuvaa (E), a high altitude catchment (Fig. 1), $T_o$ lies within the range of $T_{ens}$ for most days, and temperature forecast $T_{cal}$ was improved by the temperature calibration. The well performing raw temperature forecasts for this catchment are representative for most catchments in eastern Norway. Representing western Norway, raw $T_{ens}$ in Viksvatn (W) has a seasonal cold bias that is reduced by the temperature calibration. The cold bias is typical for several catchments in the coastal regions West, Mid and North. Another western catchment, Foennerdalsvatn (W), has a similar cold bias in $T_{ens}$ to Viksvatn (W), but for Foennerdalsvatn the bias is notable for all seasons and even increases for $T_{cal}$ (Fig. 4).

### 4.2 Skill – relations to season, spatial location, and catchment characteristics

Scatter plots of the difference between CRPSS for calibrated and uncalibrated forecasts for the temperature ($T_{cal}$ and $T_{ens}$) and streamflow ($Q_{cal}$ and $Q_{ens}$) ensembles are shown in Fig. 5. Each dot represents a catchment and the color indicates the region. The two panels in Fig. 5 show how the change in temperature CRPSS affects the change in streamflow CRPSS for spring and autumn. For spring, the relationship is close to the 1:1 line, whereas for autumn streamflow is less sensitive to the temperature calibration.

Catchment CRPSSs for spring and autumn were sorted according to increasing CRPSS for $T_{ens}$ and $Q_{ens}$ in Fig. 6. The figure reveals that $T_{ens}$ is more skillful in spring than in autumn when $T_{ens}$ has no skill (i.e. CRPSS<0) for about half of the catchments (i.e. they perform poorer than the climatology). In spring, 97% of catchments have skillful temperature forecasts. Temperature

calibration improved the temperature skill for most catchments in autumn, whereas for many catchments in spring, the skill worsened. For streamflow, $Q_{ens}$, there are only small differences in CRPSS between spring and autumn (Fig. 6 right panels). Calibration of temperature improved the skill for streamflow, $Q_{cal}$, in autumn. Whereas for spring, the streamflow forecast skill followed the temperature skill change, and are both reduced and improved.

CRPSS for uncalibrated temperature and streamflow forecasts, and the change in CRPSS, calculated as the difference in CRPSS between calibrated and uncalibrated forecasts, were mapped for all catchments. Fig. 7 and 8 show the CRPSS values for spring and autumn, respectively. The figures include box plots showing the variations in skill within each region, for both calibrated and uncalibrated forecasts. Neither $T_{ens}$, nor $Q_{ens}$ skill show any clear spatial pattern in spring (Fig. 7 left panel). For autumn, however, $T_{ens}$ has the lowest skill for the coastal catchments (Fig. 8 left panel). A coastal low CRPSS in autumn

is also seen for $Q_{ens}$, even though less distinct compared to $T_{ens}$. Both temperature and streamflow CRPSS were improved by calibration for the coastal regions (Fig. 8 right panel).

Table 1 summarizes the result of the linear regression analysis between catchment characteristics (i.e. catchment area and elevation difference) and skill. By indicating the significance and sign of the relationships, significant relationships were found for 12 out of 40 regression equations (5% significance level). Elevation difference is negatively correlated to streamflow

CRPSS for the regions East and Mid. Region East also has a negative correlation between streamflow CRPSS and catchment area as opposed to the other regions that have a positive correlation. The correlation does not change sign between the seasons for any of the regions. Calibrated temperature and streamflow CRPSS plotted as a function of catchment area are presented for East and South in Fig. 9.

### 4.3   Snowmelt flood 2013

Forecasts and observations for a snowmelt driven flood are presented in Fig. 10 for Bulken (W), located in western Norway. The figure shows forecasted streamflow for lead times 2, 5 and 9 days for the target dates May 16-26 2013. Note that for the lead times 2, 5 and 9 days, the forecasts for e.g. May 18, are issued on May 16, 13 and 9, respectively. The horizontal grey dotted lines represent the mean annual, the 5-year and the 50-year floods (i.e. the operational flood warning levels) in this catchment. Figure 10 reveals how temperature calibration increases the streamflow for Bulken, leading to a change in warning

level for all lead times. In addition we see how the ensemble spread increases with lead time (from lower to upper panel), from a narrow range around the ensemble mean for the lead time of 2 days, to a very wide range for lead time of 9 days.

### 5   Discussion

Box plots of validation scores for all catchments and lead times in Fig. 3 show that, on average, both raw $T_{ens}$ and calibrated $T_{cal}$ temperature ensembles were more skillful with a higher CRPSS, for shorter as compared to longer lead times, and that

$T_{cal}$ was more skillful than $T_{ens}$. Even though both bias and dispersion (i.e. reliability) as measured by rank histogram slope and convexity improved with longer lead times, the reduced sharpness and increased uncertainty, resulted in a reduced skill (CRPSS). For streamflow, the bias increased with longer lead times, while dispersion improved. Further, $Q_{cal}$ was slightly more skillful than $Q_{ens}$. Overall, the grid calibration of temperature had a positive effect on both temperature and streamflow

for most validation scores and lead times. The calibration procedure applied in this study involves many interpolations and downscaling steps that increases the uncertainty in temperature forecasts. We believe that a catchment specific temperature calibration, tailored to the needs for hydrological forecasting, would solve this challenge.

## 5.1     Effect of temperature calibration for the temperature forecast skill

The skill for both raw (uncalibrated) $T_{ens}$ and calibrated $T_{cal}$ temperature ensembles varies with season (Fig. 5 – 8). The

relatively small temperature skill improvements in spring, and large skill improvements in autumn, can be explained by the skill of the raw ensembles $T_{ens}$. The low skill for $T_{ens}$ in autumn and winter is caused by a cold bias, and lays the ground for the large improvements seen for $T_{cal}$. The seasonal differences in skill and response to calibration show the importance of using seasonal calibration parameters. It is also apparent that the applied methods do not perform optimally for all seasons. For spring, the results show that several catchments have a reduction in the forecast skill after calibration. By inspecting the

forecasts in detail, we found a too extensive correction of temperature for some days and catchments. Quantile mapping, as most statistical techniques, is sensitive to forecasts outside the range of calibration values and period (Lafon et al. 2013), which can be an explanation for too high a correction in the highest $T_{ens}$ quantile. The use of forecasts from different model cycles might affect the consistency in the forecasts. Moreover, the calibration parameters are sensitive to the representativeness of the calibration period.

The most pronounced spatial pattern is the low autumn CRPSS for uncalibrated ensembles $T_{ens}$ in the coastal areas. This is seen from the boxplots for the regions West, Mid and North (Fig. 8) and in the plots of the western catchments Viksvatn and Foennerdalsvatn during winter months (Fig. 4). This cold bias is documented for the Norwegian coastal areas in the cold seasons by Seierstad et al (2016), and is mainly caused by the radiation calculations in the ECMWF model (Hogan et al., 2017). The coarse radiation grid results in warmer sea points being used to compute longwave fluxes applied over colder land

points, causing too much cooling. This effect is seen for the temperature forecasts for winter 2014 and 2015 for the coastal catchments in fig 4 (b) and (c), in contrast to the inland catchment (a) which is less biased. The radiation resolution is improved in later model cycles (Hogan et al., 2017; Seierstad et al., 2016). In addition, the challenging steep coastal topography is not well represented by the spatial resolution in the ECMWF model (Seierstad et al., 2016). For inland catchments, and the regions South and East, CRPSS shows that the uncalibrated $T_{ens}$ is skillful for both autumn and spring; hence, the calibration has a

smaller effect in these catchments.

## 5.2    Effect of temperature calibration for the streamflow forecast skill

The skill of the temperature calibrated streamflow ensemble forecasts, $Q_{cal}$, improved for most of the catchments for autumn, while both improved and reduced skill were seen for spring (Fig. 5 – 8). Autumn streamflow skill was improved by temperature calibration for all regions, the largest improvement was seen for the coast, and the regions West and Mid. Two possible explanations for this spatial pattern are (i) the improvement in temperature forecast skill during autumn in these regions, and (ii) that many coastal catchments are more sensitive to calibration of temperatures since the temperatures are more frequently around zero degrees compared to the colder and dryer inland catchments. In spring, no clear spatial patterns are seen, neither for $Q_{ens}$, nor for the change in skill.

It is also evident that, independent of the sign of the temperature skill change (Fig. 5), a change in temperature has a larger impact on streamflow in spring than in autumn. During spring, temperatures are often close to the two threshold temperatures that control the phase of precipitation and the onset of snowmelt. Such periods are challenging to simulate correctly (Engeland et al., 2010). Of additional importance, for spring as opposed to autumn, is the snow storage at the end of winter, and the snowmelt contribution to streamflow. Hence, estimated streamflow has a high sensitivity to changes in temperature during spring, a sensitivity also described for Alpine snow covered catchments by Ceppi et al. (2013). Verkade et al. (2013), on the other hand, found only marginal effects of pre-processing temperature and precipitation for the streamflow skill in the Rhine catchments. The results presented herein and in the cited papers, indicate that the effect of pre-processing depends on the hydrological regime (i.e. sensitivity to temperature), the initial skill of the forcing variables, and for which temporal periods (i.e. for specific events, seasons, or the whole year) the sensitivity is evaluated. The same lead time was used to relate improvement in streamflow to temperature, we consider this robust since most catchments in this study have a concentration time of less than a day.

In summary, it can be concluded that to further improve streamflow forecasts during the snowmelt season, improved temperature forecasts are essential. Streamflow forecasts during spring have the highest potential for improvements since the temperature forecasts were not, for a majority of the catchments, improved by the applied calibration. For autumn, the substantial improvement in temperature forecast skill by grid calibration improves streamflow forecasts, but the sensitivity is less than for spring.

## 5.3    Catchment characteristics and skill

Only a few significant relationships between the catchment characteristics, e.g. catchment area and elevation gradient, and skill were found (Table 1). We expected to find the highest temperature skill in large catchments, due to averaging, and in catchments with small elevation differences, due to less elevation correction inaccuracy. No significant relationships between temperature skill and elevation difference was found for any combination of region or season. A positive relationship between temperature skill and catchment area was found for five out of ten regression equations. This result is not conclusive, but

indicates that (i) the smallest catchments are smaller than the grid size of the ECMWF model and therefore sensitive to the pre-processing and (ii) it is more challenging to forecast weather on small spatial scales than large spatial scales.

It was expected that streamflow skill would increase with catchment area due to averaging effects. Significant linear regression coefficients were found for East and South but with different signs, the same tendencies for both spring and autumn. The interpretation of this result is therefore ambiguous. For elevation difference, a significant negative correlation was found for three out of ten datasets. This suggest that the downscaling approach has a potential to improve the streamflow forecasts. These results are not conclusive, and studies that are more detailed are needed to determine any significant relationships to catchment characteristics.

Forecasting in small catchments with particular characteristics may be challenging since they may not be well represented, neither by the numerical weather prediction model, nor by the calibration methods. In our dataset, Foennerdalsvatn (fig 4c) is such an example. The catchment area is only 7.1 km2, elevation is high, topography is steep, glaciers cover 47% of the catchment area, and it is located close to the coast.

## 5.4    Snowmelt flood 2013

The snowmelt flood event (Fig. 10) illustrates clearly how temperature calibration affects forecasted ensemble streamflow. The increase in forecasted temperature by grid calibration, results in additional snowmelt and thus increased streamflow. The increased streamflow led to a change in the warning level, from below to above the 5-year flood. For this event, however, the use of calibrated temperature reduced the performance of the forecasted streamflow, $Q_{cal}$. The reference streamflow, $Q_o$ is better captured by the streamflow forecasts based on uncalibrated temperature forecasts, $Q_{ens}$. The deterioration in the forecast performance using calibrated temperature is particular for this event. Other results provided in this study show clearly that the calibrated temperature ensembles improve the streamflow forecasts on average.

Figure 10 reveals how the ensemble range for the snowmelt event clearly increases with increasing lead time. For a lead time of 2 days (lower panel) the range is too narrow, while for a lead time of 9 days (upper panel), the wide forecasting intervals capture the events, but there is little information left in the forecasts.

## 6    Summary and conclusion

The main objective of this study was to investigate the effect of temperature forecast calibration on the streamflow ensemble forecast skill, and to identify potential improvements in the forecasting chain.  We applied a gridded temperature calibration method, and evaluated its effect on both temperature and streamflow forecasting skill. The seasonality in skill was evaluated and correlations to catchment characteristics and spatial patterns were investigated. Supported by the results presented in this paper, our answers to the research questions listed in the introduction are summarized as follows:

Are there seasonal effects of temperature calibration on the temperature ensemble forecast skill?

- The largest temperature skill improvements by calibration were found for low performing coastal catchments in autumn and winter.
- The effect of calibration on temperature skill was less clear in spring. In spring, the calibrated temperature resulted in reduced skill for many catchments.
- Smaller bias in spring explained a higher $T_{ens}$ skill and hence, less room for improvements by calibration.

Are there seasonal effects of temperature calibration on the streamflow ensemble forecast skill?

- In autumn and winter, streamflow skill improved for most catchments. For spring, the calibration resulted in both better and worse skill.
- In spring, changes in temperature skill had a higher effect on streamflow skill, compared to autumn and winter. .

Are there spatial patterns in the ensemble forecast skill and if so, can these be related to catchment characteristics?

- The skill in temperature forecasts was the lowest in coastal catchments along the coast in West, Mid and North in autumn, caused by a cold bias in the forecasts (this was also the case for winter, although these results are not shown).
- The largest improvement in skill for both temperature and streamflow was found for catchments with a cold bias in the temperature forecasts.
- A regional division seemed useful to identify spatial patterns in temperature forecasts, whereas for streamflow the spatial patterns were not so obvious.
- It was not possible to conclude a relationship between the catchment characteristics and skill.

Snowmelt flood

- Streamflow increased by temperature calibration, changing the flood warning level, clearly showing the importance of correct temperature calibration for catchments with snow during snowmelt season

This study showed that the applied gridded temperature calibration method improved the temperature skill for most catchments in autumn and winter. Temperature forecasts have an impact on streamflow, and are important for seasons where temperature determines snowmelt and discriminates between rain and snowfall. The improvement in temperature skill propagated to streamflow skill for some, but not all, catchments. This was to a large degree dependent on region, and the skill of the uncalibrated ensemble.

The most obvious improvement in the forecasting chain is to use the same temperature information, the SeNorge temperature, for calibrating the temperature forecast that is used for calibrating the hydrological model, generating the initial conditions for the hydrological system, and evaluating the performance. In particular, the calibrated temperature forecast could be improved during spring when the streamflow forecasts are the most sensitive to temperature. The pre-processing of temperature includes both an elevation correction depending on lapse rate and the calibration method. Lapse rate in this study is defined as a constant, but actually depends on weather conditions, location and elevation. In addition, the calibration method, here the quantile mapping, is sensitive to forecasted values outside the observation range, and other methods should be considered. In this study, we have investigated the isolated effect of uncertainties in temperature forecasts. For a more complete assessment of forecast

uncertainties, error in initial conditions, hydrological model parameters and structure need to be accounted for. In particular, we might expect a strong interaction between uncertainties in temperature forecasts and model parameters controlling snow accumulation and snow melt processes.

The conclusions in this study are based on a testing period of almost three years. Even if this is a relatively short testing period,
we believe that the large number of catchments to a large degree compensates for the short testing period and that the results and conclusions are therefore relatively robust. We suggest that some of the main conclusions can be valid for regions with a similar climate. The most important general conclusion is that streamflow forecasts are sensitive to the skill of temperature forecasts, especially in the snowmelt season. In addition, this study shows that reducing the cold temperature bias in coastal areas results in improved streamflow forecasts, and that the pre-processing needs to account for seasonal differences in
temperature forecasts (biases).

## 7    Data

Processed data is available by contacting corresponding author. Raw meteorological data can be obtained directly from ECMWF.

## 8    Authors contribution

15  T. J. Hegdahl prepared the data, set up the forecasting chain (including writing new code for non-available functionalities), did the data simulations and analysis, and wrote the manuscript. K. Engeland contributed to the writing. K. Engeland, I. Steinsland and L. M. Tallaksen contributed to the design of the study, by advice during the work, and in the revision of the manuscript.

## 9    Acknowledgement

The authors would like to thank B. Grønbech at NVE for the work done with setting up the hydrological model for ensemble forecasting. We would also like to thank A. Singleton at MET Norway for his comments during the work and proofreading of the manuscript. T. Nipen and I. Seierstad at MET Norway we thank for their support and sharing of precipitation and temperature ensemble forecast calibration knowledge. In addition, their aid was valuable during the implementation of https://github.com/metno/gridpp in the forecasting chain. Thanks also to colleagues at NVE working in the project "Better
uncertainty estimation in flood forecasting", led by E. Langsholt.
We are also very thankful for in depth reviews and valuable comments provided by J. Verkade and three anonymous referees, and from Editor J. Seibert. The review process helped improve the quality of the finished paper.

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

**Table 1: Summary of significant correlations between CRPSS for calibrated temperature ($T_{cal}$) and streamflow ($Q_{cal}$) ensembles and catchment characteristics, i.e., area and elevation difference ($\Delta H$), for the five regions. Blue indicates a significant positive relationship, red a significant negative relationship, and grey a non-significant relationship. Results are for a lead time of 5 days.**

| | | $T_{cal}$ | $Q_{cal}$ | $T_{cal}$ | $Q_{cal}$ |
|---|---|---|---|---|---|
| | | SPRING | | AUTUMN | |
| Area (km²) | East | grey | red | grey | red |
| | South | blue | blue | grey | blue |
| | West | blue | grey | grey | grey |
| | Mid | blue | grey | blue | grey |
| | North | grey | grey | blue | grey |
| $\Delta H$ (m) | East | grey | red | grey | red |
| | South | grey | grey | grey | grey |
| | West | grey | grey | grey | grey |
| | Mid | grey | grey | grey | red |
| | North | grey | grey | grey | grey |

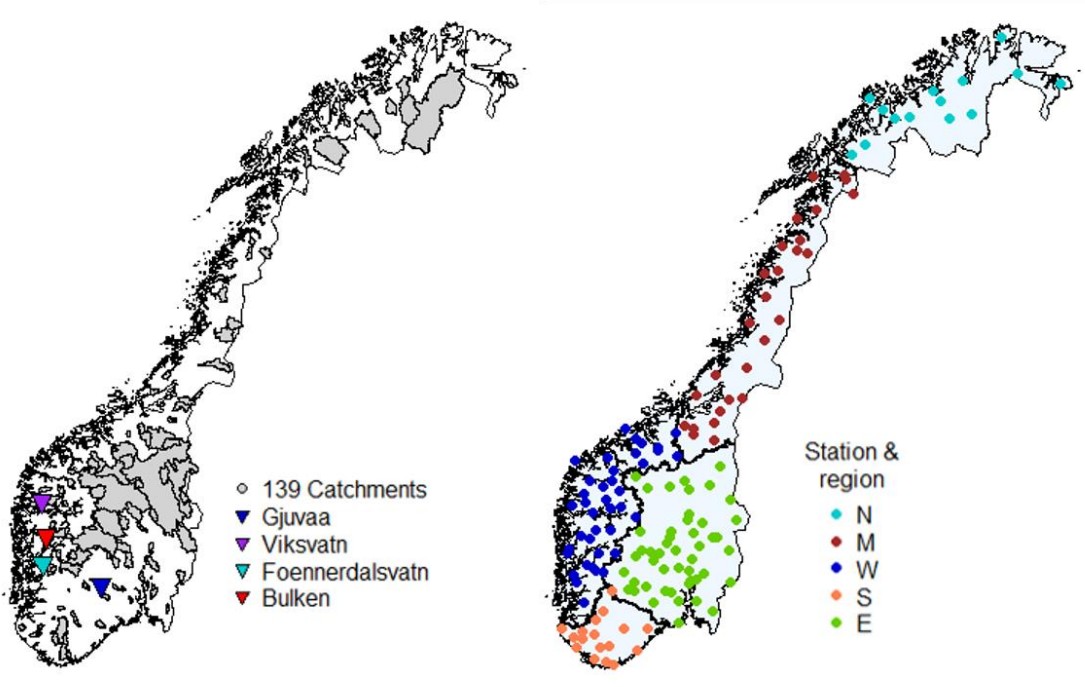

**Figure 1: The maps for Norway indicate the 139 catchments used in this study. The left map shows the catchment boundaries including the location of four selected catchments. Please note that many catchments are relatively small and difficult to detect. The location of the catchments gauging stations are shown in the map on the right. Norway was grouped into five regions (N=North, M=Mid, W=West, S=South, and E=East), all regions are marked with different colors and regional boundaries.**

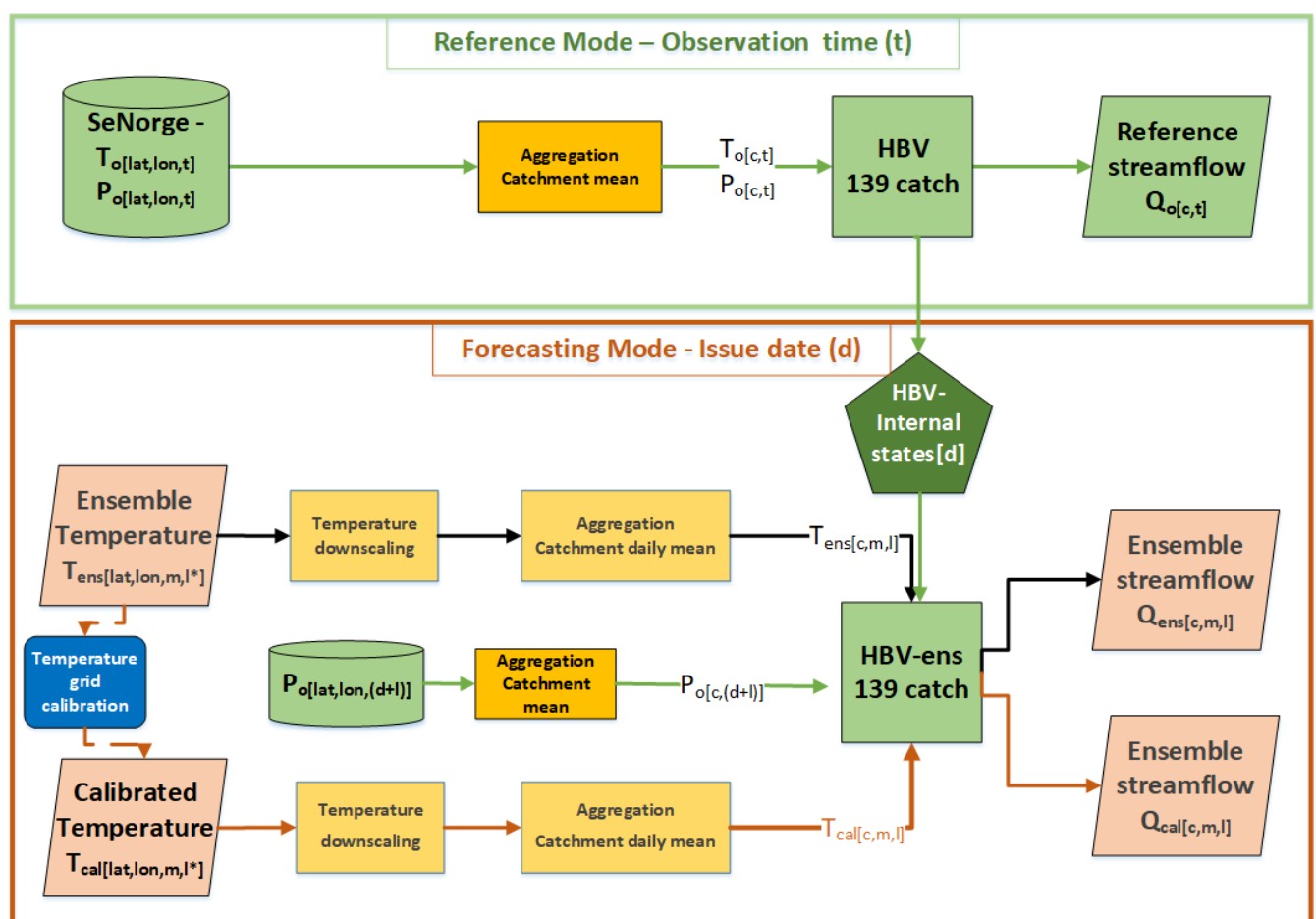

**Figure 2: Conceptual diagram of the ensemble forecasting chain. The upper green frame shows the reference mode that is the calculation of reference streamflow using the HBV model with catchment aggregated daily mean values of SeNorge temperature**

5 **($T_o$) and precipitation ($P_o$). In the forecasting mode, the lower red frame, ECMWF temperature ensembles are downscaled to 1×1 km² prior to catchment aggregation. Calibrated temperature ($T_{cal}$) is estimated from $T_{ens}$, applying a grid calibration at 5×5 km resolution. Daily average forecast values ($T_{ens}$ or $T_{cal}$) and observed precipitation ($P_o$) are used to force the hydrological model at forecasting issue date ($d$), with internal states from the reference mode.**

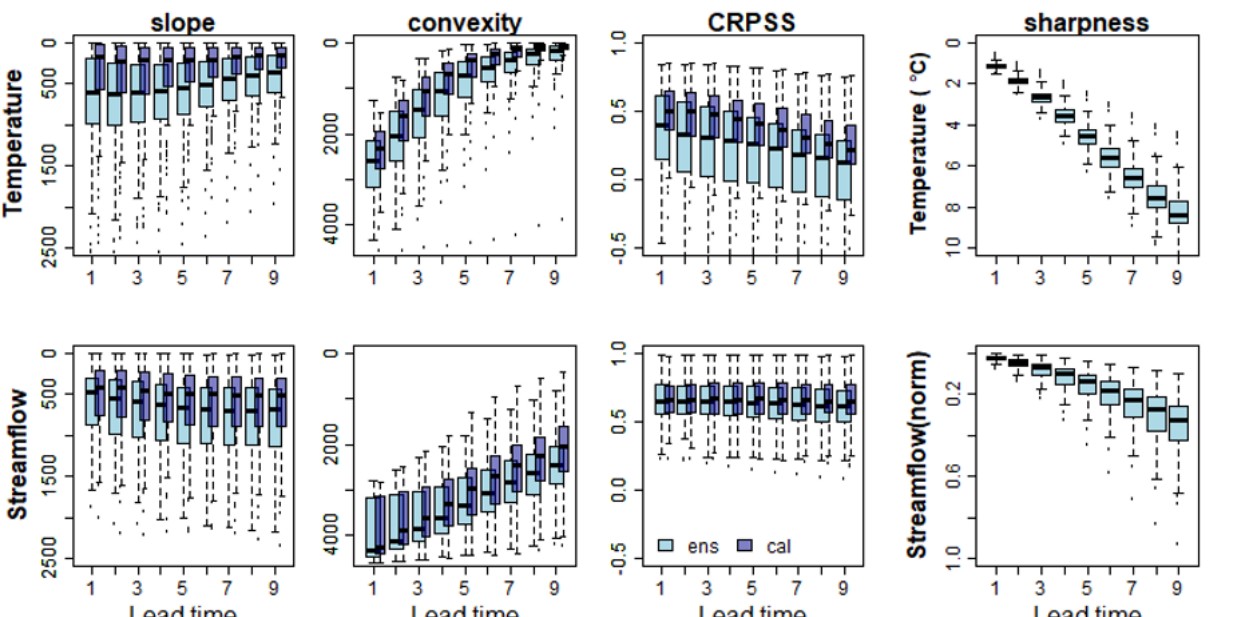

**Figure 3: Summary of temperature and streamflow scores for all lead times. Each box represents the 139 catchments values averaged over all issue dates. Rank-histogram test decomposition for slope and convexity is shown in first and second column respectively and CRPSS in the third column. The forth column shows sharpness for the uncalibrated forecasts. Temperature in the top row and streamflow in the bottom row. Results are based on the full dataset, and are shown for both uncalibrated (light blue) and calibrated (blue) ensembles at lead times 1 to 9 days. For slope and convexity, zero is the optimal value, and the scales are reversed so that the optimal value is on the top, corresponding to CRPSS optimal value at 1.0.**

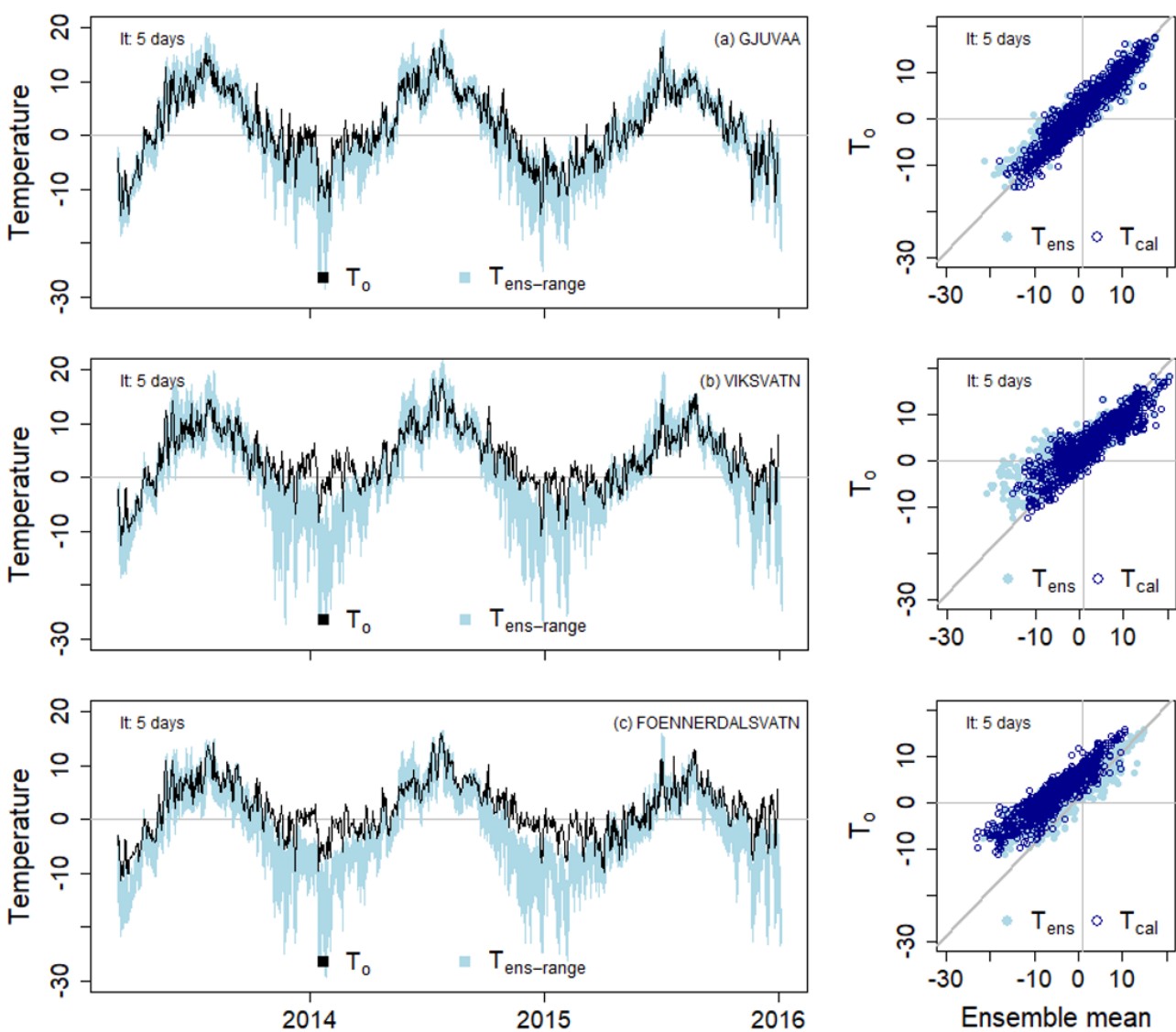

**Figure 4: Timeseries of temperature for Gjuvaa (a), Viksvatn (b) and Foennerdalsvatn (c) showing the range of uncalibrated temperature ensemble forecast ($T_{ens-range}$, lightblue area) for the period 2013-2015, SeNorge observations are shown as black lines. Scatter plots show ensemble mean temperature for both calibrated ($T_{cal}$, blue) and uncalibrated ($T_{ens}$, lightblue) temperature plotted against SeNorge temperature ($T_0$). Lead time is 5 days.**

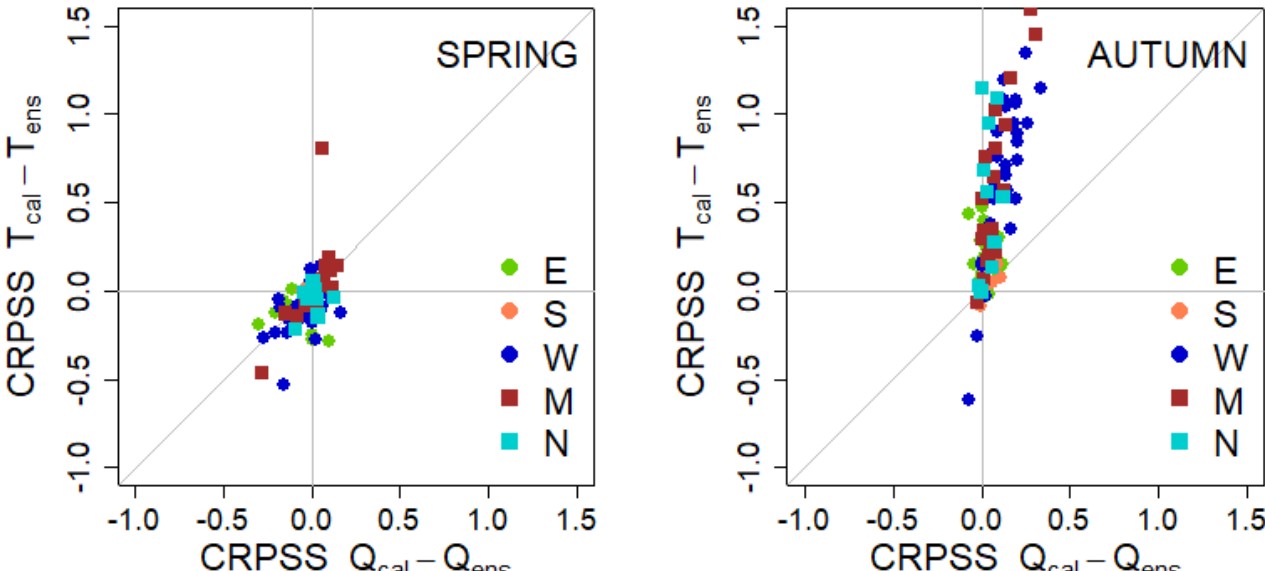

**Figure 5: Difference in CRPSS for uncalibrated and calibrated temperature for spring and autumn. The difference in temperature skill is plotted on the y-axis and the difference in streamflow skill on the x-axis. The grey diagonal represent the 1:1 line. Catchment values are color indexed by region. All plots are presented for lead time 5 days.**

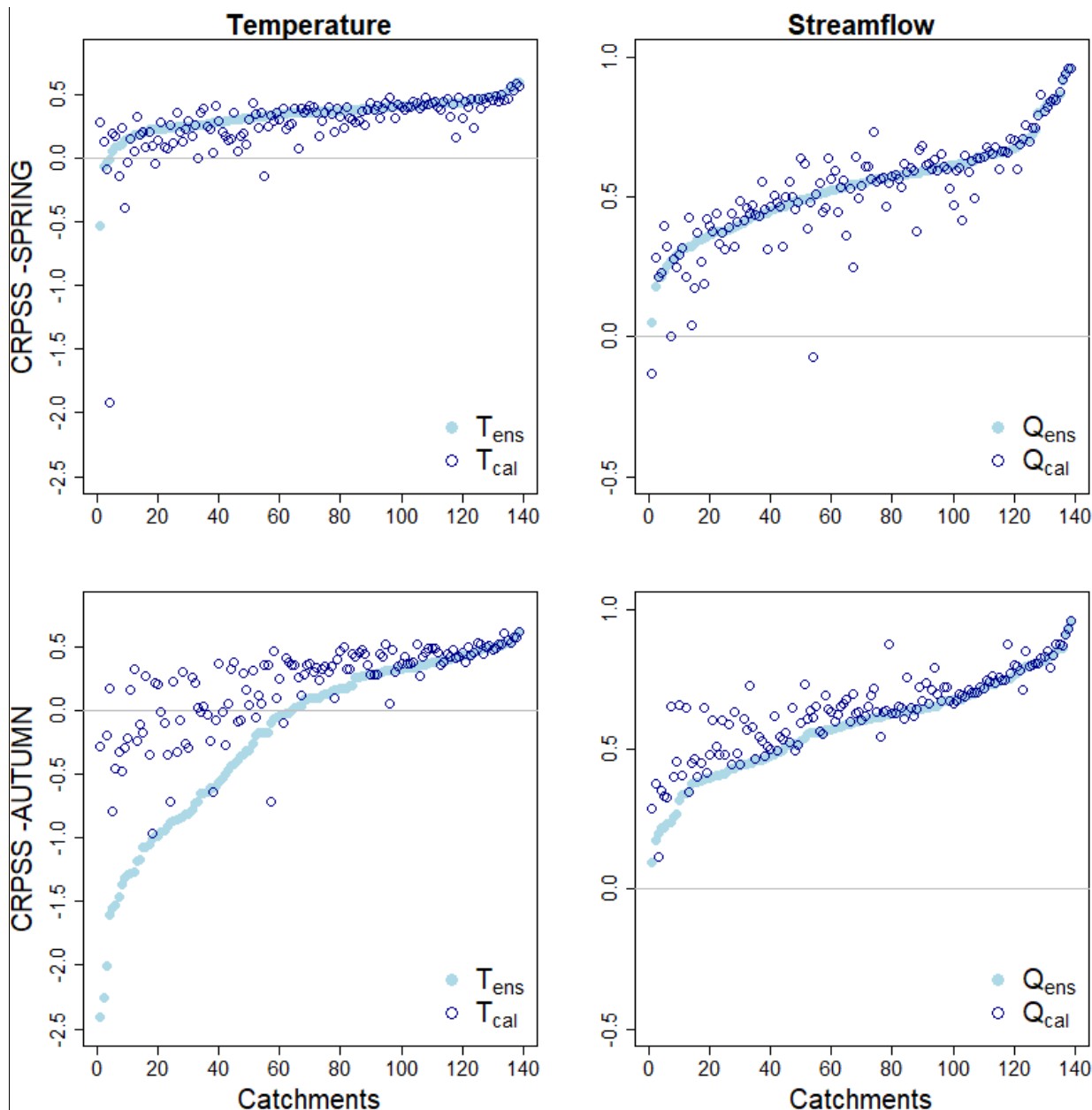

**Figure 6: Temperature (T$_{ens}$ and T$_{cal}$, first column) and streamflow (Q$_{ens}$ and Q$_{cal}$, second column) CRPSS for SPRING (top) and AUTUMN (bottom). The catchments are ordered by increasing CRPSS for T$_{ens}$ and Q$_{ens}$ (light blue dots), the catchment calibrated values (T$_{cal}$ and Q$_{cal}$) are plotted as blue circles. All results are presented for lead time 5 days.**

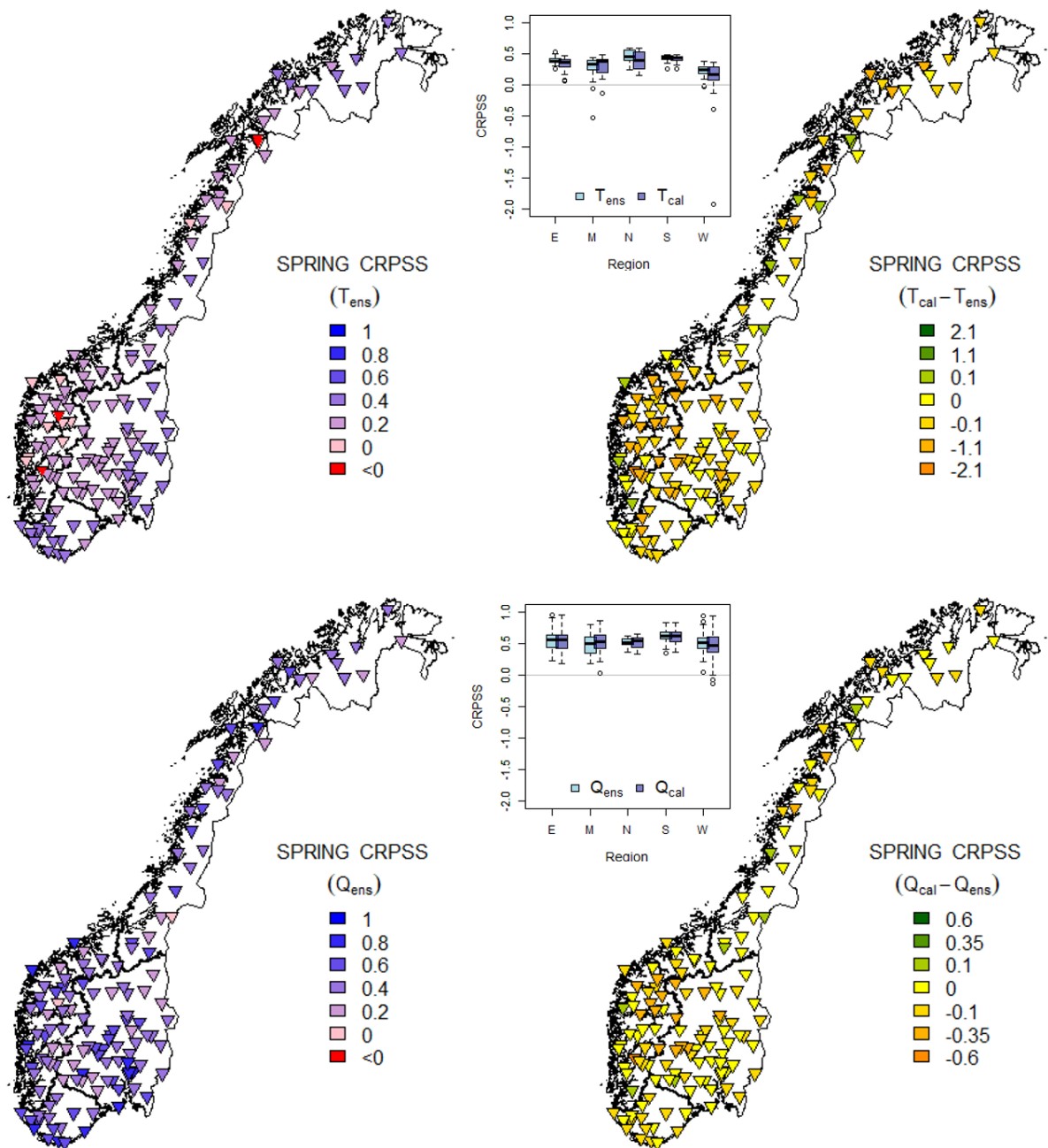

**Figure 7: Spring CRPSS for uncalibrated forecasts (left maps) and CRPSS difference between calibrated and uncalibrated forecasts (right maps) for temperature (upper panel) and streamflow (lower panel). A darker blue color (left maps) indicates an optimal performance (maximum CRPSS=1.0), pink a CRPSS of zero, and red a negative value. A green color (right maps) indicates a positive effect of temperature calibration on the skill, yellow means no effect, and orange color indicates a negative effect. The boxplots show temperature and streamflow CRPSSs grouped by region (Fig. 1). All results are presented for lead time 5 days.**

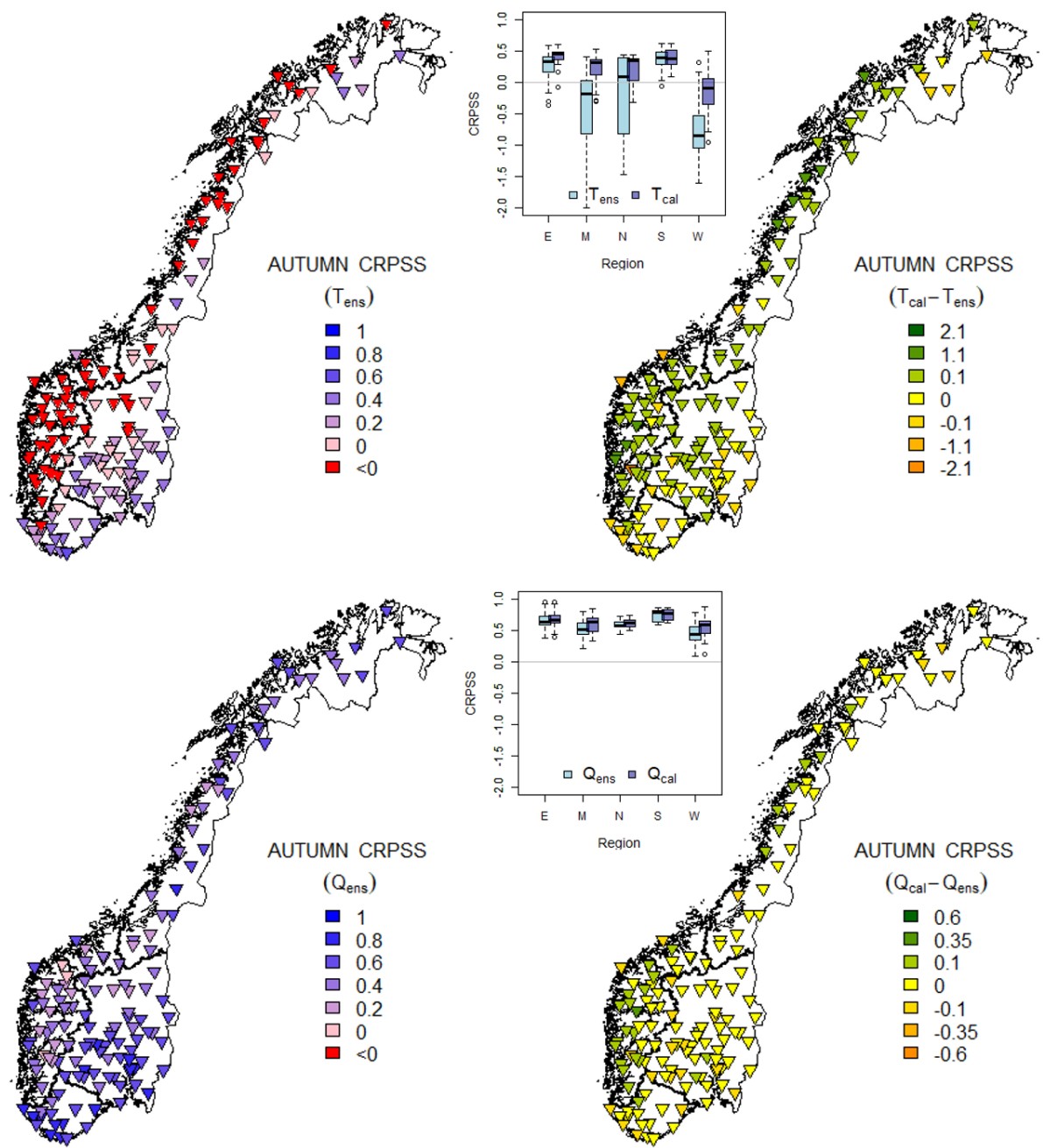

**Figure 8: Autumn CRPSS for uncalibrated forecasts are presented to the left where darker blue color indicates an optimal performance (maximum CRPSS=1.0), pink color represents a CRPSS of zero, and red negative. The differences in CRPSS between calibrated and uncalibrated forecasts are presented to the right, where green color indicates a positive effect of temperature calibration on the skill, yellow zero, and orange color indicates a negative effect. The boxplots of both calibrated and uncalibrated temperature and streamflow CRPSS show catchments grouped by region (Fig. 1). All results are presented for lead time 5 days.**

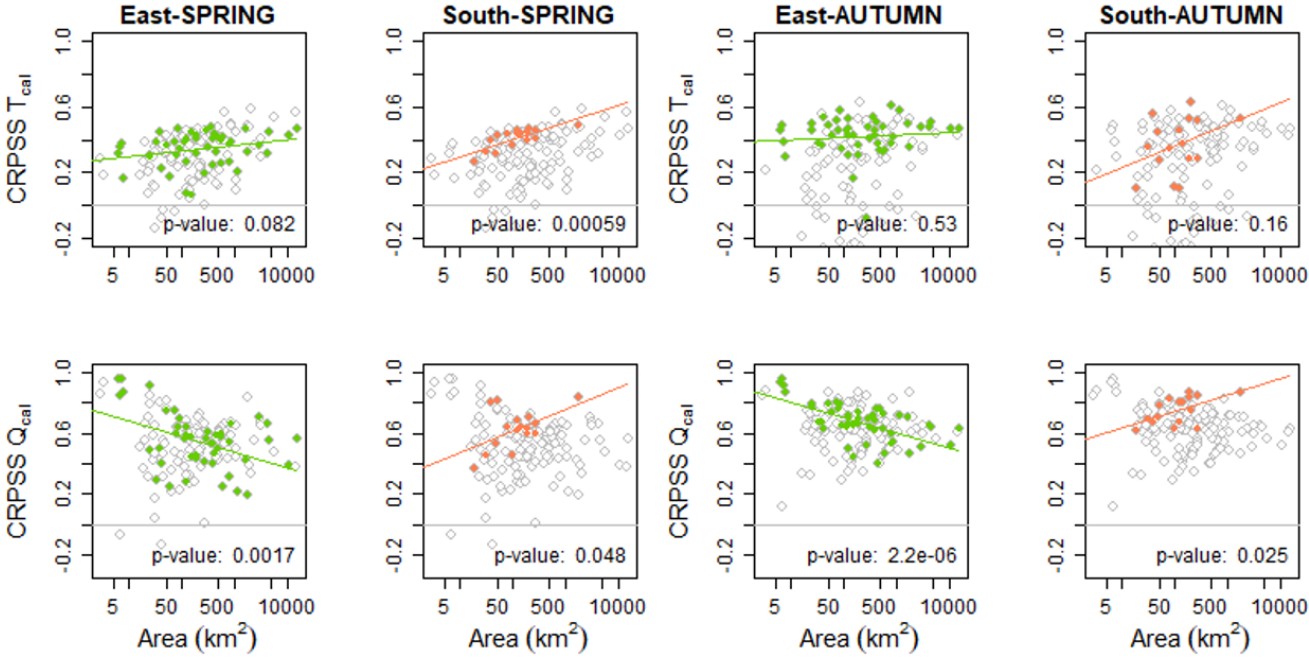

**Figure 9: Temperature (top panels) and streamflow (bottom panels) CRPSS for the two regions East (E) and South (S), plotted as a function of catchments area for both autumn and spring. The colored dots show the CRPSS for the respective regions whereas the grey circles show the CRPSS for all 139 catchments. The linear regression line is plotted along with its p-value (significantly different from zero for p-values < 0.05). All results are presented for lead time 5 days.**

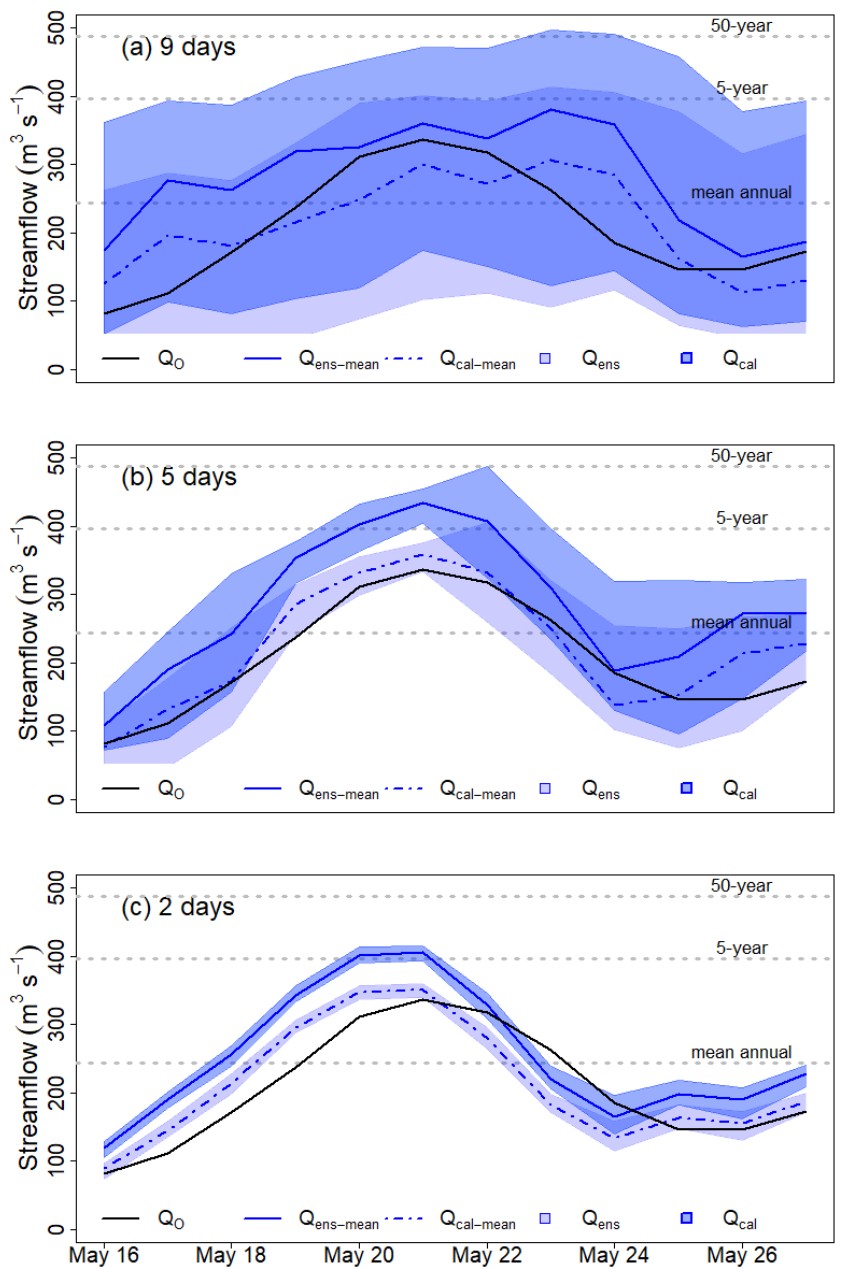

**Figure 10: Forecasted streamflow for the Bulken catchment fort lead times 9, 5 and 2 days. Forecast target dates on the x-axis, and streamflow (m³s⁻¹ ) on y-axis. Reference streamflow with SeNorge observations Qo (black solid line), ensemble mean uncalibrated temperature Qens (blue line), ensemble mean calibrated Qcal (blue dotted line), ensemble range Qens (light violet area) and ensemble range Qcal (light blue area). The grey dotted lines indicate the thresholds for mean annual, 5-year and 50-year floods.**

