# Peer review of "Streamflow forecast sensitivity to air temperature forecast calibration for 139 Norwegian catchments"

_Hydrology and Earth System Sciences, 2018_

## Referee Comment (RC1) · Anonymous Referee #1 · 6 Nov 2018

General comments:

This is a well written paper. It investigates the impact of temperature forecasts on streamflow forecast skill, especially considering the effect of pre-processing of temperature ensemble forecasts. The study is based on forecasts for a large number of catchments in Norway, thus providing a very comprehensive and systematic analysis. The paper provides an important contribution to the research and practical application of ensemble meteorological forecasts for streamflow forecasting.

Detailed comments:

1. Page 2, line 16-17. There are different ways of producing meteorological ensemble

forecasts. Typically, also model physics are perturbed.

2. Page 5, line 18-19. Not clear here how catchment average precipitation and temperature are estimated. Are they based on the SeNorge data sets? If so, is it then necessary to apply elevation corrections for the model calibration, since elevation corrections have been applied for producing the SE Norge data sets?

3. Page 6, line 17-20. Why use a daily time step for the streamflow forecasts? Meteorological forecasts with a 6-hour time step are available.

4. Page 7, line 4-6. For the quantile mapping, a critical issue is the mapping of forecasts outside the range of observed data. How is this done?

5. Page 8, line 12-13. Alternatively, you could use persistent forecast as benchmark. This would be more appropriate for evaluating short-term forecast skill.

6. Page 12, section 5.3. There are a lot of repetitions in this section. I suggest including discussion on spatial patterns in sections 5.1 and 5.2.

Technical corrections:

1. Page 2, line 30. Evensen (2003) not in reference list.

2. Page 4, line 27. "og" -> "and"

3. Page 11, line 20 and 24. Delete "Ivar".

---

## Referee Comment (RC2) · Anonymous Referee #2 · 15 Nov 2018

This manuscript presents analyses of the sensitivity of streamflow forecasts to air temperature forecast calibration. The manuscript is well written, well structured, and I only have a few minor comments to the presentation, most of them just edits.

I find the description of validation scores and evaluation scores in 3.2 somewhat short. The section could give a better description of the rank histograms, and what is actually meant by the different shapes. And what is meant by slope and convexity being "negatively oriented"? Something seems wrong with the last sentence.

P2L5 three main componentS?

P2L14 Langsrud et al, 1998a and 1998b are missing from reference list. What kind of

statistical uncertainty models? (One line, to understand better what is different from the ensemble forecast)

P4 2.1 Is Gjuvaa in the region South or East? Bulken is in the West region?

P5L20 PEST can be generic tools for parameter estimation or a particular software, what is it here?

P6 2.2.4 / 3.1.1 Is the forecast from ECMWF point forecast (centre of the grid cell) or averages for the entire grid cell?

P7L29-30 "In this study, the ensemble range (...) visually assessed the sharpness." Something seems wrong, rephrase.

P9L12-13 since "reliability has improved and some sharpness is maintained". This could be better explained.

P6L29 I guess it should be "atmospheric lapse rate"?

P8L17 remove s from catchments.

P9L9 remove comma after convexity

P10L7 they performs – remove S.

P11L17 Rather than just sensitive, I think QM is unable to correctly map forecasts outside the observation range.

P12L2 temperatureS are?

P14 L29 "elevation correction dependency on lapse rate" – is this correct?

P16L17 No publisher?

Fig1 caption: Most of the catchments on the left are too small to be visible?
* * *
373, 2018.

---

## Referee Comment (RC3) · J.S. Verkade (Referee) · 16 Nov 2018

This manuscript is suitable for publication. The research described in it has a clear objective which is to try and determine if 'calibrated temperature ensemble forecasts' result in better streamflow forecasts compared to the non-calibrated equivalents. The research setting, the approach and the data used is well described and the results are well laid out. I have a few concerns/questions/suggestions but these would require only minor revisions to the manuscript.

Please also note the supplement to this comment:

[Figure]

https://www.hydrol-earth-syst-sci-discuss.net/hess-2018-373/hess-2018-373-RC3-supplement.pdf

[Figure]

**Supplement:**

**Review of 'Streamflow forecast sensitivity to air temperature forecast calibration for 139 Norwegian catchments' by Trine Hegdahl et al.**

Jan Verkade, November 2018

**Overall impression**

This manuscript is suitable for publication. The research described in it has a clear objective which is to try and determine if 'calibrated temperature ensemble forecasts' result in better streamflow forecasts compared to the non-calibrated equivalents. The research setting, the approach and the data used is well described and the results are well laid out. I have a few concerns/questions/suggestions but these would require only minor revisions to the manuscript.

**Minor comments**

**Overall**

- Multiple references are made to seasons in which the effect of temperature forecast calibration on streamflow was negligent. You're right to point out that the reason is that temperature forecasts only matter if/when it affects the simulation of snowmelt processes. You could consider mentioning this in the start of the paper, explain that for this reason, you're looking at only those seasons where temperature affects streamflow through either rain-falling-as-snow or through snowmelt, and then omit reference to the other seasons altogether. I find it a bit distracting from the main points.
- For many hydrologists, the word 'calibration' has a different meaning from how it is used in your paper. I acknowledge that your meaning is consistent with how many meteorologists would interpret it. I would recommend to address this issue by either use a different word (I believe HESSD readers may be more familiar with 'post-processing') or by addressing this in the text somewhere.
- Citations aren't always properly formatted. I think I've seen ((double parentheses)), for example. In S3.1.2, l12, a correct way to refer to the evidence would be (Seierstad, 2017) with the 'personal communication' listed in the bibliography. I think. I've also seen citation in which both first and family names are listed. May be good to verify against Copernicus citation rules.

**Abstract**

- l9-11 These sentences distract from the point you're going to make. While the facts you state may have a place in the introduction, I would omit these from the abstract.
- l20 'the HBV model is used to *calculate* streamflow'. The verb *to calculate* presumes certainty. Pls consider using *estimate* instead.
- l21 'influenced'. My understanding is that 'influences' (and the associated verb) are a thing of the mind ("Who are your main influences?" "Joan Baez"). For physical processes, I think 'affected' is more suitable.
- l26 'however'. I don't think this sentence contradicts anything that was stated before. Hence, the word 'however' may be omitted.

**Section 3.1.2**

- I am not entirely sure who provides the calibration parameters. L5 suggests MetN, but the sentence "To establish the calibration parameters..." (l8) may be interpreted as an explanation of how the authors have done this.

- In the Met Norway procedure, why aren't temperature *observations* used? Are the HIRLAM reanalyses deemed to be sufficiently certain? This may deserve a few informed comments.
- If I am correct in understanding that both the raw and the calibrated ensembles have been provided by Met Norway then maybe this should be stated more clearly. Or is it the case that Met Norway computed the calibration parameters on a data set from 2006-2011 and that you applied these yourself to a data set ranging from March 2013 through Dec 2015? If so, maybe state this more bluntly?
- I am assuming that you used a HIRLAM reanalysis. Is that correct? If not, what lead times are you using and do the HIRLAM forecasts you used have the same max lead time as the ECMWF ensembles? I am only familiar with a few instances of HIRLAM and these all go out to just over 2 days max.
- By off-setting Tens against Tcal, you create the impression that Tcal is not an ensemble forecast. Consider using Traw and Tcal instead.
- l29-30. The 'assessment' was done by you, not by the ensemble range.
- On assessing sharpness: how confident are you that a visual assessment does the job? Pls consider plotting the empirical distribution of sharpness of all your forecasts and comparing those.
- If you're calibrating the temp ensembles on a leadtime by leadtime basis and on a grid cell by grid cell basis, chances are that you'll change the temporal pattern (forecasted temperature as a function of time) as well as the spatial pattern. Does this in any way affect use in streamflow forecasting? I believe there are some techniques that may be helpful in trying to restore spatial-temporal relations (the Schaake shuffle springs to mind). Would these have a use in present study?

**Section 3.2**

- Would it be fair to say that temperature forecasts are only relevant if they can discriminate between freezing and non-freezing situations? If so, would it be justified to focus more on this discrimination? Perhaps by defining an event (T<0, for example) for which one can compute a range of verification scores (false alarms, hits, ROC, Brier's probability score, etc). I acknowledge that this would be feasible for temperature and less obvious for streamflow.

**Section 4**

- " To reduce the amount of presented results, the remaining part of this paper focuses on CRPSS for a lead time of 5 days." This is fine, but temperature forecast at 5-day lead time may not affect streamflow forecasts until a (much) longer lead time. Or conversely, streamflow forecasts at day 5 would have been affected by a day 2 temperature forecast (this is an example). As in some cases you're comparing Q-forecasts with T-forecasts, how have you accounted for this?

**Section 4.1**

- In the text, you refer to observed temp as To. In plots, as Tobs. Pls make this consistent. I recommend using Tobs throughout.
- L23-25. These sentences are better placed in a discussion section, I think.
- L19 'influence' is missing an 's'. Pls consider replacing by 'affects' though.

**Section 4.2**

"Scatter plots of the difference between CRPSS for calibrated and uncalibrated forecasts". CRPSS in itself is a fairly abstract measure. The difference between two CRPSS scores is, I find, even more abstract. What's the meaning of those values? As CRPSS is a skill of a forecast versus a baseline, why not simply calculate the CRPSS of the calibrated forecasts using the CRPS of the uncalibrated forecasts as a baseline?

**Section 5**

L7: 'dispersion' is not an expression of quality but a characteristic of an ensemble. Saying 'dispersion improved' makes little sense then?

**Section 5.1**

- L11 "skill... depends". Consider replacing by "skill... varies with".
- "Quantile mapping is sensitive to forecasts outside the range of calibration values and period". I think it would be good to point out that this is true for any statistical post-processing procedure.
- Immediately following: "and can be a" –> "and *this* can be a"
- On the causes of temperature forecast bias. You go into some detail to explain a situation in which land is colder than sea. Would this be a typical situation for summer/winter? If so, can you more directly link this to some of the results you're showing?

**Section 5.2**

L10 Grammatically, this sentence is awkward if not wrong.

**Figures**

**Overall**

Many figures use a lot of white space between various plots/panels. Consider reducing this or, even better, removing altogether.

**Figure 1**

- Do the grey polygons add up to 139 in total? If so, many must be **really** small?
- Caption: consider using 'boundaries' instead of 'limits'

**Figure 4**

- Why plot the ensemble **mean** and not all five ensemble members, possibly as horizontal lines?
- The axes of the plots in the right-hand column vary. Please consider unifying this. Also: please consider ensuring that horizontal and vertical axes are identical. Maybe they are, but the labeling isn't.

**Figure 5**

- What lead time are these plots for?
- Is the lead time for T identical to that for Q? What is the 'response time' of the catchment to snowmelt? If not zero then shouldn't this be taken into account somehow?

Please consider...

- ... removing data for seasons for which temperature has little or no effect on streamflow levels.
- ... unifying horizontal and vertical axes. it took me a little while longer than I cared to realise that the light grey slanted line is the 1:1 diagonal.

**Figure 6**

- What do you want the reader to compare? CRPSS(T) and CRPSS(Q)? Or CRPSS(spring) v CRPSS(autumn)? Pls ensure panels are ordered accordingly.
- pls ensure that within a row, panels have identical vertical axes so this comparison can indeed be done (i.e. the reader can then easily compare the top left with the top right plot)

**Figure 10**

- The background colours have an effect on the colouring of Qens and Qcal. Please consider removing the background shades. Maybe replace these by threshold lines only?
- Please consider removing the number of lines in the plot, for example by only showing a shaded area with no line at the edges thereof.
- What is the purpose of showing both the 'real' observations and the 'model streamflow with SeNorge observations'? Is this distinction made in the paper, and addressed?
- Consider reversing the order of the graphs. The 9d lead time graph was available before the 2d lead time graph?
- The horizontal axis labeling is not in English.
- As all horizontal axes are identical, pls consider removing white space between plots altogether and only label the axis of the bottom plot.
- The warning levels aren't relevant, are they? On reflection: you're scoring the forecast ensembles using CRPSS and rank histograms. This shows absence of preference for doing well for 'extremes', even though the work appears to be inspired by forecasting for floods. How is this consistent? Maybe omit references to 'floods' altogether?

---

## Referee Comment (RC4) · Anonymous Referee #4 · 22 Nov 2018

**Supplement:**

Especially in hydrometeorological predictions where methods from both the meteorological and the hydrological forecasting community are used, it is of major importance to carefully define the terminology and to coherently use throughout the manuscript.

The current form of the manuscript shows a lack of precise formulations (e.g.: calibration, pre-processing, skill) which should be revised to better communicate the content of the study. Some of the graphics should be enhanced to facilitate the readability and the caption are sometimes incomplete. In addition, more than 15 references mentioned in the text are missing in the reference list and should be added. Furthermore, some additional references could be of interest within discussion to put the findings of the study into a broader picture. Many of the references especially concerning the meteorological forecast are user guides and or technical reports or personal communications, which is fine, but I would appreciate if some more peer-reviewed literature would be cited as there is a large body of existing literature concerning the verification of ECMWF temperature predictions.

In general, the language could be clearer and more concise. To me it is not clear what the authors understand under the term pre-processing, at least in the beginning of the manuscript. E.g. in literature there is a distinction between dynamical and statistical downscaling (see e.g. Li et al. (2017), Yuan et al. (2015)) and statistical downscaling does include a bias-correction. In the present manuscript, the term downscaling does only refer to applying a laps rate correction and interpolation, what is not as downscaling is referred to in the literature. However, I think it would be important for the reader to have a short general overview of what pre-processing is in the introduction. In particular the term calibration, in the present manuscript used as a synonym for bias correction should be introduced more carefully because the term calibration is used by statisticians but in the meteorological, climatological and hydrological communities, the term bias correction is more common.

As you mention the forecasting period used for the study is only two and a half year long which might influence the results. You state this in the discussion but do not explain why it could be critical. I suggest that you discuss this explicitly. Namely, within such a short period, the interannual variability might not be sufficiently covered. In addition, using forecasts from different model Cycles (38r1 to 41r1) might have an influence of the skill as well because the adaption within a new cycle might enhance or decrease the forecast performance making the comparison between seasons difficult as it might not only originate from the particular season but might be influenced by model versions. I suggest including such limitations in the discussion.

To apply Quantile mapping you do need the distribution of the forecast and the distribution of the observations. In section 3.1.2 you state that "MET Norway uses Hirlam temperature forecasts to provide the observational climatology used for parameter estimation". I think here, more information is needed to enable the reader how the calibration is done. Are daily values used for the parameter estimation? Is it empirical or parametric QM used and how are values outside the range treaded (e.g. constant extrapolation)? Is it a member-by-member approach or are the same parameters used for all members? One critical point is that the calibration parameters are interfered from the Hirlam but the hydrological model is run with SeNorge observations. Why are not these observations used? The correction will account for the bias between ECMWF and Hirlam but I would expect that biases with SeNorge will at least slightly differ. Why don't you use the observations from SeNorge to get your calibrations? In the summary it is stated that "The most obvious improvement in the forecasting chain is to use the same temperature information, the SeNorge temperature, for calibrating the temperature forecast that is used for calibrationg the hydrological model, generating …" (P14/L25-27). But if I understand correctly from the manuscript SeNorge and Hrilam are not the same. I have troubles with this procedure as it is known that different forecast models do have different

biases. To bias-correct or calibrate ensembles the observations should be taken into account and not another forecast. In this case the bias between two forecasts will be corrected and not the bias of the forecast with regard to the observations. Furthermore, by the many interpolations used, there is a large uncertainty introduced which will lower the trust in the results. Interpolation of ECMEF and Hirlam to derive correction parameters, another interpolation to meet the hydrological model requirements. I think this limitation should be discussed.

Another point that should be discussed is if seasonal correction parameters are really sufficient or does it introduce artificial jumps between periods. In a climate context, seasonal windows for parameter estimation might be sufficient but in an operational forecasting context a shorter window should be taken into account if possible.

In Section 3.2 where the CRPS is introduced you mention different notations (CRPS, $S_{crp}$) and same for the CRPSS. I think this is confusing, as later in the text only CRPS is used. I suggest only introducing one of the notations and stick to that.

**Specific comments:**
P1
L7-14: You say the flood forecasting system uses deterministic forecasts for temperature and precipitation). But the ECMWF model you reference provides an ensemble of 51 members. Please state how this is used.

L11-12: "An alternative approach is to use meteorological and hydrological ensemble forecasts" is somewhat misleading. Either you used ensemble meteorological forecasts in combination with hydrological models to generate ensemble streamflow forecasts or one uses a different methodology to produce hydrological ensembles forecasts. I suggest rewriting the sentence: "An alternative approach is to combine meteorological ensemble forecast with hydrological models to quantify the uncertainty in the forecasted streamflow".

L14: "for an accurate forecasting of ", or "to accurately forecast streamflows"
L15: Ensemble forecast of temperature from **the** ECMWF "
L16: "to improve **the** skill and reduce bias**es**"
L18: why do you mention precipitation here? If it is not used for the calibration I would avoid it here.
L20: was used to calculate **the** streamflow

P2
L1: Floods **can** damage… and can have a high …
L5: component**S**
L9: The reference "Müller et al." is missing in reference list
L14: Both reference "Langsrud 1998 a and b" are missing.
L16: as a mean to account for uncertainty in **the** forcing.
L21: The Reference Cloke & Pappenberger, 2009 and Wetterhall et al., 2013 are missing
L25: the ensembles **can be** calibrated
L26: Hamill and Colucci, 1997 and Buizza et al, 2005 are both missing
L29: Gneiting et al. 2005 is missing, Wilks and Hamill 2007 is missing, Raftery et al. 2005 is missing
L30: Evens 2003 is missing

L31: Gneiting et al. 2005 is missing, Wilks and Hamill 2007 is missing. The order of the references is different compared to L29. Bremnes, 2007 is missing.

L31-32: This sentence is very general, it is arbitrary clear that different correction methods do correct the biases differently. I suggest either being more specific about single methods, or to summaries different methods to provide a better overview for the reader instead of listing available techniques. Maybe cite some standard books for statistical bias correction and downscaling (Wilks, 2011) and for forecast verification (Jolliffe & Stephenson, 2011).

P3
L1: snow cover without "–"
L2-4: This sentence is unclear to me. Can you elaborate what you mean?
L5: Gragne, 2015 → missing reference
L7-8: Forecasting, downscaling and interpolation are three completely different things and the challenge is connected to much more than laps rate. For interpolation and downscaling a large part can be attributed to temperature height correction which depend to a large degree to laps rates. But forecasting of temperature is far more complex and related to chaos theory. Rephrase please.
L9: Again, missing references: Aguado and Burt, 2010; Pagès and Miro, 2010, Peter et al., 2010.
L13: Alpine (capital A) as the study looks at catchments in the Alps.
L15: ", **found** only modest….",
L17: I think the effect is not marginal, as you later on show with your results.
L26: do you mean from both, the hydrological and the meteorological perspective?
L27: from **the** ECMWF, in addition I would mention the lead time here but maybe not the MET Norway pre-processing setup as you use the QM to pre-process the forecasts which is, if I understood correctly, not yet part of the pre-processing setup at MET Norway.
L28: Are the retrospective forecasts operational forecast for the period within 2013-2015? This could be misleading for readers or misinterpreted as reforecasts (or hindcasts) which are forecasts for the same day as the operational forecast but for the past 20 years using re-analyses for the initialization. Maybe rephrase to avoid any misinterpretation.
L30: again, I think marginal is the wrong word, if the effect is assumed to be marginal, why should you analyse it in such detail.
L31: Not clear to me. Do you mean that the observed precipitation is used to drive the hydrological model? Specify that to make it clearer.
L33-P4L2: Maybe combine this with the preceding paragraph. This would make it less generic.

P4
L5: **spatial** variations
L6: rather high then steep?
L9: delete "flows"
L18: the smallest catchment has an area of only 3 km^2? Or is it a typo?
L21: what are the selection criteria for "data of sufficient quality"
L27: "og" seems to be Norwegian
L31: snowmelt **driven** flood event

P5
L5: write "available at SeNorge.no"
L7: Mention what kind of interpolation is used (bilinear, kriging, …)
Section 2.2.1:

Mention here that you use the precipitation data from this data set as a substitute of the precipitation forecasts (if this is the case).

L15: constitutes **as** the basis
L20: explain what PEST is.
L21: Abbreviation NS (for Nash-Sutcliffe) not introduced before.
Section 2.2.2:
Is the calibration done for each catchment separately? Do the given values for the NS coefficient represent the mean for all catchments? Is this good? Please state how these values translate into performance compared with other hydrological models.
L22: Missing Reference Gusong (2013), In reference list only Gusong 2016 is listed

2.2.3
To make this more coherent I suggest renaming this section into "Reference observations" (or similar) and in the latter part of the study refer to reference observations as well. Otherwise it is difficult to distinguish between the model stream flow and forecasted streamflow. E.g. on P6 L13 you write reference model run, I assume this is the same as model streamflow? This is somewhat confusing if you state it twice in 2 different paragraphs.

P6
L6: write the lead time as well in days 246 hrs (i.e. 10 days). Why is it 246 hrs and not 240?
L7: The Reference "ECMWF (2018a)" does only provide the documentation and support page of the ECMWF. The Specific documentations can be downloaded. The scientific basis of the ENS system has been discussed in multiple publications and it might be worth to reference some of them and point to this documentation for specific points only.
L8: "the ensemble members of ENS are…"
L9: "with different perturbed conditions **to** represent the …"

3.1
See comment to 2.2.3. I don't get the difference between model streamflow and reference model run. If I understand correctly these are the same. If so, only describe it in one section. I think here it would be suitable. Reference run = model streamflow, use the same terminology if it is the same.
Are the ENS forecast temporally aggregated as well?

L25: replace "include" with "referred to as"
L27: Use the same units for both grids. ° or km^2. Best would be use both units for both grids, one of them in brackets.
3.1.1
What is the rationale behind the choice of using a nearest neighbour technique?

P7
L4: Bremnes 2007, 2004 are missing in the reference list.
L8: Can you give a reference for the sentence "gives a higher skill and are less biased"
L20: Ensemble forecast verification does not only focus on reliability and sharpness. Therefore, different measures need to be taken into account (as well biases are important).
L30: "lowest and highest forecasted value" does it mean the minimum and maximum? Why not the $10^{th}$ and $90^{th}$ percentile and the interquartile range. I think this gives a better

estimate of the sharpness of the forecast as it does not only account for the most extreme members.

P8

L12: I would rephrase the sentence. "which a skilful forecast should outperform" and write it in a single sentence.

L18: negative values mean (without s)

L19: "which perform similar to the reference forecast (climatology in this case)"

L20: Do you use here the mean of the daily CRPS? (CRPS with overbar?)

L25-26: This sentence seems to be wrong.

L27: Usually seasons are aggregate in winter = December-February (DJF), spring = mar-may (MAM) and so on. Can you explain your motivation to choose this definition of seasonal aggregation?

P9

L8: as shown in figure…

L9: no comma after "convexity"

The description about the slope and complexity is hard to follow. Could you give an example what the values really tell, e.g. how does a rank histogram look like with a complexity of 2000? I think rank histograms are very useful to be used for visually interpretation and the complexity and slope somehow lead to a reduction of the usefulness of the rank histogram at least to people not familiar with these parameters.

L15: I recommend repeating what $T_O$ and $T_{ens}$ is to enhance the flow in the text.

L27: Same here, mention the abbreviation in brackets in the text to help the reader.

L29: "influenceS" ; Do you mean in streamflow skill or CRPSS?

P10

L4: Do you know why there is no improvement during summer by using calibrated temperatures? Is it due to the absence of snow / snow-melt in summer?

L5: You often mention the Figure number in the last part of the sentence. I personally would prefer this information first what makes it easier to follow the text and figures at once; **It** reveals

L20: What is the significance level you used? I would mention this in the text.

4.3

What are the criteria you used to choose this flood event in May 2016? Mention the motivation for this specific event.

L10: If possible embed this in the floating text and see separate comment to the figure.

P11:

L1: to make it clear I would add: "…increases with lead time (form upper to lower panel)." Linked to my comment on the caption in Figure 10 that it could be misinterpreted as a continuous forecast starting at may 16[th].

L4: "**The** box plots … show" (show without s)

5 Discussion

Here I would again use words instead of the Tens Tcal only: "Both raw (Tens) and calibrated (Tens) temperature forecasts were more skilful with …". I think it makes the text more interesting to read. This could be adapted in different parts of the Manuscript, in the beginning of each section this should be repeated.

L5-9: "Overall, the grid calibration of temperature had a positive effect on both …", but the lines before it states "…, resulted in reduced skill". This is somehow contradictive, could you make this clearer?

L18: missing reference Lafon et al. 2013
L20: wrong citation format Ivar Seierstad et al. (2016)
L24: again, wrong citation format Ivar Seierstad et al. (2016)

Subtitles for 5.1 and 5.2 should be coherent "calibration for …" or "calibration for the…"

L26: forecast**s**

P12
L4: "Hence, calculated … " word at wrong place within sentence.
L7 "indicate" delete additional s

L18: "the bias in $T_{ens}$ is explained by" I think this statement is too strong. It **can** be an explanation, but I think it cannot be reduced to this single causality, as you state in the next sentence.
L21: "The $T_{ens}$ CRPSS is skilful" forecasts have a positive CRPSS and are skilful. The current formulation is not logical, a CRPSS is not skilful.

L28: please state these characteristics very shortly again here.

P13
L1: I don't understand what you mean with "the averaging effect on temperature skill dominates".
If I understand correctly, you could discuss here what the difference would be if you use a spatially distributed hydrological model (e.g. gridded version of the model with high resolution). The effect of temperature downscaling might be higher in this case because you do not average temperature again after the downscaling and the spatial distribution within a catchment would have a much larger effect especially in catchments with high spatial variability of soil properties, altitude and vegetation cover.

L13: "the calibrated temperature reduced the skill of the forecasted streamflow." Please state what skill measure you mean here, did you calculate the CRPSS or bias for that specific event? In the result you only describe the range of the calibrated / uncalibrated ensembles but not a measure of skill.
L15-17: I think you would like to point out that other errors (in the meteorological dataset and the hydrological model) do influence the results. If so, the sentence should be rephrased. Now the reader might think that forecasts are always getting worse if they are calibrated and this would be an argument against your conclusive statement in the summary on Page 14/L19-18.

Figures:

Figure 1: write "grouped" instead of "divided". Something is wrong in the first sentence "this study shown using". Please rephrase.

Figure 3:
Avoid overlap of the boxplots to enhance the readability of the plot. There seem to be two line-artefacts on both sides of the figure.

Figure4:
In the text you write $T_O$ and in the Figure it corresponds to $T_{obs}$. Similarly, $T_{ens}$ and $T_{ens-range}$. It might facilitate the text of the abbrevations are more consistent in the text, captions and the figures.

Figure 6:
Line artefacts on the left of the figure.

Figure 10:
It is hard to see the actual forecast. I suggest removing the background colors for the warning levels and just plot lines instead. The Figure can easily be misinterpreted as the individual plots (e.g. upper panel for lead day 2) look like a continuous forecast. Maybe it would be more suitable to plot boxplots instead.

Captions: Forecast issue date is the date when the forecast was issued, hence the x-axis could be different for each panel in this figure. I recommend adapting the caption to make this clearer, e.g. target day instead of issue date.

"model streamflow with SeNorge observations" this is $Q_O$. I would write it in brackets as you do for $Q_{cal}$.

References:
Jolliffe, I. T., & Stephenson, D. B. (2011). *Forecast Verification*. (I. T. Jolliffe & D. B. Stephenson, Eds.), *Forecast Verification*. Chichester, UK: John Wiley & Sons, Ltd. https://doi.org/10.1002/9781119960003
Li, W., Duan, Q., Miao, C., Ye, A., Gong, W., & Di, Z. (2017). A review on statistical postprocessing methods for hydrometeorological ensemble forecasting. *Wiley Interdisciplinary Reviews: Water*, *4*(December), e1246. https://doi.org/10.1002/wat2.1246
Wilks, D. S. (2011). *Statistical Methods in the Atmospheric Sciences*. (D. S. Wilks, Ed.) (3rd ed.). London: International Geophysics Series, Vol. 100, Academic Press Inc.
Yuan, X., Wood, E. F., & Ma, Z. (2015). A review on climate-model-based seasonal hydrologic forecasting: physical understanding and system development. *Wiley Interdisciplinary Reviews: Water*, *2*(5), 523–536. https://doi.org/10.1002/wat2.1088

---

## Author Comment (AC1) · 20 Dec 2018

General comments:
This is a well written paper. It investigates the impact of temperature forecasts on streamflow forecast skill, especially considering the effect of pre-processing of temperature ensemble forecasts. The study is based on forecasts for a large number of catchments in Norway, thus providing a very comprehensive and systematic analysis. The paper provides an important contribution to the research and practical application of ensemble meteorological forecasts for streamflow forecasting.

Detailed comments:
1. Page 2, line 16-17. There are different ways of producing meteorological ensemble forecasts. Typically, also model physics are perturbed.
AR: You are right. The ECMWF ensemble prediction system includes stochastic perturbation to the model physics. We will add to the sentence to address this aspect.
AC: Suggestion "… are created by perturbing both the initial states of the original deterministic forecast and the physics tendencies of the …."

2. Page 5, line 18-19. Not clear here how catchment average precipitation and temperature are estimated. Are they based on the SeNorge data sets? If so, is it then necessary to apply elevation corrections for the model calibration, since elevation corrections have been applied for producing the SE Norge data sets?
AR: We agree that the description of how temperature is used in the hydrological model is ambiguous and this will be clarified in the text. You are right that elevation correction is applied to the SeNorge dataset. Our set-up for the HBV models uses catchment average temperature as input, calculated from the SeNorge data.  The elevation correction mentioned in l18-19 refers to the internal correction in the HBV model. These are used to adjust catchment average temperature and precipitation, representing the catchment mean elevation, to each elevation zone in the HBV model. A linear elevation adjustment is applied to temperature, whereas an exponential adjustment is applied to the precipitation.
AC: We reformulate line 18-19 as follows:
"The model uses catchment average temperature and precipitation as input. Each catchment is divided into 10 elevation zones, each covering 10% of the total catchment area. The catchment average precipitation and temperature are elevation adjusted to each elevation zone using catchment specific laps rates.

**Author Response to RC#1**

3. Page 6, line 17-20. Why use a daily time step for the streamflow forecasts? Meteorological forecasts with a 6-hour time step are available.

AR: The operational HBV model used for flood forecasting runs on a daily time step. In addition, the SeNorge data that is used for model calibration and updating, provides only daily values.

AC: We make no modifications to the manuscript

4. Page 7, line 4-6. For the quantile mapping, a critical issue is the mapping of forecasts outside the range of observed data. How is this done?

AR: MetNorway use parametric quantile mapping based on the hourly first 24h. When a forecast is outside the observation range, a 1:1 extrapolation is used. Therefore, if a forecast is 2°C higher than the highest percentile of forecasts used for calibration, then the calibrated forecast is 2°C higher than the same percentile for the reference.

AC: Suggestion p7, line 12: "The same coefficients, based on the first 24h mapped, are applied to all lead times and ensemble members individually.  For forecasts outside the observation range, a 1:1 extrapolation is used. I.e. if a forecast is 2°C higher than the highest mapped percentile, then the calibrated forecast is 2°C higher than the same percentile for the reference. "

5. Page 8, line 12-13. Alternatively, you could use persistent forecast as benchmark. This would be more appropriate for evaluating short-term forecast skill.

AR: A persistent forecast will have some predictive skill in the short-range, but less for longer lead times. Engeland and Steinsland (2014; Fig. 4) show that the persistence did not add value after two days for selected Norwegian catchments. Pappenberger et al (2015) suggest using persistence as benchmark, based on a study of catchments larger than 6000km$^2$. However, given our selection of catchments, which are relatively small, quick responding, and with rapid changes in weather, combined with an aim to evaluate at longer lead times, we choice not to use persistence as benchmark. Rather, we used climatology as a benchmark since: (1) it is straightforward to get climatology as an ensemble, and (2) the focus of study is a lead time of five days. The daily climatology represented as daily ensemble (not an average value) gives a good representation of seasonal variations. Moreover, for this lead time persistent forecast has small predictive power due to the relatively short memory of our catchments (e.g.  the streamflow autocorrelation for a time lag 5 days is less than 0.6 for about 80% of our catchments for the 25% highest flows).

AC: We make no modifications.

6. Page 12, section 5.3. There are a lot of repetitions in this section. I suggest including discussion on spatial patterns in sections 5.1 and 5.2.

AR: We will carefully read and revise Section 5.3 to avoid repetitions, and consider rewriting 5.1 and 5.2, to include the discussion from 5.3.

AC: We will revise and rewrite section 5.1 and 5.2 for the new version of the manuscript

Technical corrections:

1. Page 2, line 30. Evensen (2003) not in reference list.

AR: Thanks; will be added

AC: "Evensen, G.: The Ensemble Kalman Filter: theoretical formulation and practical implementation. Ocean Dynamics, 53(4), p343-367, 2003."

2. Page 4, line 27. "og" -> "and"

AR: This will be corrected.

AC: Changed

3. Page 11, line 20 and 24. Delete "Ivar".

**Author Response to RC#1**

AR: OK

AC: We will write: "Seierstad et al. (2016) documented the relatively low skill and cold bias for sub-zero ECMWF temperature forecasts for the Norwegian coastal areas in cold seasons."

**References**

Engeland, K. and Steinsland, I.: Probabilistic postprocessing models for flow forecasts for a system of catchments and several lead times. Water Resources Research 50(1), p182-197, doi:10.1002/2012WR012757, 2014.

Pappenberg, F., et al.: How do I know if my forecasts are better? Using benchmarks in hydrological ensemble prediction. Journal of Hydrology, 522, 697-713. 2015

---

## Author Comment (AC2) · 20 Dec 2018

This manuscript presents analyses of the sensitivity of streamflow forecasts to air temperature forecast calibration. The manuscript is well written, well structured, and I only have a few minor comments to the presentation, most of them just edits.

I find the description of validation scores and evaluation scores in 3.2 somewhat short. The section could give a better description of the rank histograms, and what is actually meant by the different shapes. And what is meant by slope and convexity being "negatively oriented"? Something seems wrong with the last sentence.

AR: Thank you for bringing this to our attention. We will provide some more details in the description of the rank histograms. By "negatively oriented", we mean that lower values (of slope and convexity) are better (i.e. more reliable forecasts). We will revise the sentence to better explain the meaning of "negatively oriented", and rephrase the last sentence to make it clear.
AC: We will apply the following changes:
- "For reliable ensemble forecasts, the rank-histogram will be uniform (horizontal). A bias in the ensemble forecast is recognized as a slope in the rank-histogram, where a negative slope indicates over-estimation by the forecasts (and vice versa). A U-shape indicates that the ensemble forecast is under-dispersed whereas a convex shape indicates over-dispersion (Hamill, 2001).
- Negatively oriented: new "… and convexity are negatively oriented, (i.e. lower values are better), and with an optimum value of zero …."

P2L5 three main componentS?
AR: Thank you, will be corrected.
AC: 'component' replaced by 'components'

P2L14 Langsrud et al, 1998a and 1998b are missing from reference list. What kind of statistical uncertainty models? (One line, to understand better what is different from the ensemble forecast)
AR: The references will be added and the text revised explaining the uncertainty model referred to.

**Author Response to RC#2**

AC: We suggest writing: "the uncertainty model accounts for the strong autocorrelation in forecast errors and estimates an uncertainty band around the deterministic temperature, precipitation and streamflow forecasts."

P4 2.1 Is Gjuvaa in the region South or East? Bulken is in the West region?
AR: We agree that the current manuscript is somewhat unclear on this issue and in the revised manuscript, we suggest adding in parenthesis to which region each catchment belongs.
AC: E.g.: "Bulken (W), Gjuvaa (E)".
We will change the following sentence: "Gjuvaa (E) is non-glaciered and located inland."

P5L20 PEST can be generic tools for parameter estimation or a particular software, what is it here?
AR: We use the PEST software to estimate parameters. We will specify this in the revised manuscript.
AC: We suggest to write: "… we used the operational model setup which has been calibrated using the PEST software to establish model parameters (Doherty, 2015)"

P6 2.2.4 / 3.1.1 Is the forecast from ECMWF point forecast (centre of the grid cell) or averages for the entire grid cell?
AR: The ECMWF forecasts should be considered as average values within the grid box, see Owens (2018, fig 3.2.1) for details.
AC: We will add this to the ECMWF description: "The ECMWF grid temperature, **which represents the average temperature for the grid cell**, was interpolated from a horizontal resolution of 0.25 …."

P7L23-24 "In this study, the ensemble range (…) visually assessed the sharpness." Something seems wrong, rephrase.
AR: Thank you. We will rephrase this sentence. We consider modifying this paragraph according to suggestions by RC#3, evaluate plot of empirical sharpness distribution.
AC: Suggestion: "In this study, the sharpness was visually assessed by looking at the ensemble range (i.e. the interval spanned by the lowest and highest forecasted values)"

P9L12-13 since "reliability has improved and some sharpness is maintained". This could be better explained.
AR: We will modify this part including evaluating plots of the empirical distribution of sharpness, ref. RC#3, and information above.
AC: We will revise this paragraph in the manuscript

P6L24 I guess it should be "atmospheric lapse rate"?
AR: You are quite right; will be corrected.
AC: to "atmospheric**"**

P8L17 remove s from catchments.
P9L9 remove comma after convexity
P10L7 they performs – remove S.
AR: Thank you.
AC: We will correct as suggested.

P11L17 Rather than just sensitive, I think QM is unable to correctly map forecasts outside the observation range.

**Author Response to RC#2**

AR: We will rephrase to enhance the problems of QM mapping outside the observational range. It is important to note that all statistical methods will have problems outside the observational range. (ref RC#3, and discussion in RC#4)

AC: We suggest writing: "Quantile mapping (as most statistical calibration methods) is sensitive to forecasts outside the range of calibration values and period (Lafon et al. 2013), this can be an explanation for too high correction in the highest $T_{ens}$ quantile."

In addition, we add a sentence p7, l19 to clarify the use of quantile mapping: "The same coefficients based on the first 24h mapped, are applied to all lead times and ensemble members individually. For forecasts outside the observation range, a 1:1 extrapolation is used. I.e. if a forecast is 2°C higher than the highest mapped percentile, then the calibrated forecast is 2°C higher than the same percentile for the reference."

P12L2 temperatureS are?

AR: Thank you; will be corrected.

AC: Changed

P14 L29 "elevation correction dependency on lapse rate" – is this correct?

AR: We will rewrite to make this phrasing clearer (ref. Detailed comment no 2 by R#1),

AC: e.g. "… an elevation correction depending on lapse rate"

P16L17 No publisher?

AR: Thank you; will be corrected.

AC: "Engeland, K., Renard, B., Steinsland, I., and Kolberg, S.: Evaluation of statistical models for forecast errors from the HBV model. Journal of Hydrology, 384(1), 142-155, 2010."

Fig1 caption: Most of the catchments on the left are too small to be visible?

AR: We agree. The western catchments are small and thus difficult to distinguish on the map. We will revise the figure accordingly and further suggest adding a note on the fact that catchments on the western coast are small in the figure legend.

AC: We add the following to the caption:" Please note that many catchments are relatively small, for location confer the right map."

Reference

Owens, R G, Hewson, T D: ECMWF Forecast User Guide. Reading: ECMWF. doi: 10.21957/m1cs7h, 2018.

---

## Author Comment (AC3) · 20 Dec 2018

Thank you for the positive and good evaluation of our article. We appreciate the comments that are valuable and helpful in order to improve the manuscript.

We would like to apology for the missing references. The error emerged when we specified the HESS format, and un-intentionally deleted many references from the reference list. The main author should nonetheless have detected this flaw prior to posting.

Replies and corrections are done as follows: the Author responses (AR) are marked with red text, while the author's suggestions to corrections (AC) are marked with blue text.  All Referee comments are kept black; we use page and line number when needed to specify the appropriate location.

**Review of 'Streamflow forecast sensitivity to air temperature forecast calibration for 139 Norwegian catchments' by Trine Hegdahl et al.**
Jan Verkade, November 2018

**Overall impression**
This manuscript is suitable for publication. The research described in it has a clear objective which is to try and determine if 'calibrated temperature ensemble forecasts' result in better streamflow forecasts compared to the non-calibrated equivalents. The research setting, the approach and the data used is well described and the results are well laid out. I have a few concerns/questions/suggestions but these would require only minor revisions to the manuscript.

**Minor comments**
**Overall**
• Multiple references are made to seasons in which the effect of temperature forecast calibration on streamflow was negligent. You're right to point out that the reason is that temperature forecasts only matter if/when it affects the simulation of snowmelt processes. You could consider mentioning this in the start of the paper, explain that for this reason, you're looking at only those seasons where temperature affects streamflow through either rain-falling-as-snow or through snowmelt, and then omit reference to the other seasons altogether. I find it a bit distracting from the main points.
AR: This is a good suggestion. We think, however, that it is useful to include all seasons in the first part of our analyses in order to highlight the differences between seasons, which subsequently provide the motivation for leaving some seasons out of the final analysis.
AC: No changes will be introduced in the manuscript.

• For many hydrologists, the word 'calibration' has a different meaning from how it is used in your paper.
I acknowledge that your meaning is consistent with how many meteorologists would interpret it. I would recommend to address this issue by either use a different word (I believe HESSD readers may be more familiar with 'post-processing') or by addressing this in the text somewhere.
AR: We agree that hydrologists might interpret the term "calibration" to "hydrological model calibration", and we will clarify our use of the terminology as illustrated in Figure 2. Pre-processing is, in our paper, a general term for any modifications applied to a raw meteorological forecast. We distinguish between calibration and downscaling, that both are pre-processing methods. This is consistent with the terminology used by the Norwegian Meteorological Institute (MetNorway) (https://github/metno/gridpp).
AC: We will clarify the distinct use of the term calibration in this paper.

• Citations aren't always properly formatted. I think I've seen ((double parentheses)), for example. In S3.1.2, l12, a correct way to refer to the evidence would be (Seierstad, 2017) with the 'personal communication' listed in the bibliography. I think. I've also seen citation in which both first and family names are listed. May be good to verify against Copernicus citation rules.

AR: Thank you.

AC: The citations and references will be formatted according to the HESS standard.

**Abstract**

• l9-11 These sentences distract from the point you're going to make. While the facts you state may have a place in the introduction, I would omit these from the abstract.

AR: You are right. We will consider rewriting the abstract.

AC: We will change the first sentences as follows:

"In this study, we used meteorological ensemble forecasts with the hydrological models to quantify the uncertainty in forecasted streamflow, with a particular focus on the impact of ensemble temperature forecasts. In catchments with seasonal snow cover, snowmelt is an important flood generating process."

• l20 'the HBV model is used to *calculate* streamflow'. The verb *to calculate* presumes certainty. Pls consider using *estimate* instead.

AR: Thank you, we will change as suggested, i.e. using 'estimate' both in the abstract and in the text.

• l21 'influenced'. My understanding is that 'influences' (and the associated verb) are a thing of the mind

("Who are your main influences?" "Joan Baez"). For physical processes, I think 'affected' is more suitable.

AR: Thank you. We will change 'influence' used as a verb to affect, and to 'effect' where 'influence' is used as a noun.

AC: Change to affects or in some cases effect: p1 l23; p2 l5;p8 l26 (effect); p9 l29; p12 l25(effect); p13 l25 (effect); p14 l7 (effect)

• l26 'however'. I don't think this sentence contradicts anything that was stated before. Hence, the word 'however' may be omitted.

AR: Thank you, we will omit "however".

AC: "Altogether, it is evident that temperature forecasts are important for streamflow forecasts in climates with seasonal snow cover."

**Section 3.1.2**

• I am not entirely sure who provides the calibration parameters. L5 suggests MetN, but the sentence "To establish the calibration parameters. . . " (l8) may be interpreted as an explanation of how the authors have done this.

AR: MetNorway did the quantile mapping, and established the calibration parameters. The calibration parameters were originally used to bias correct the temperature forecasts as provided on yr.no (the Norwegian weather forecasting). We applied the Met-parameters to the raw ENS temperature forecasts of our selected period.

AC: We will change the sentence in l8: "To establish the calibration parameters MetNorway used both ENS re-forecast (Owens, 2018) and Hirlam data from July 2006 to December 2011 interpolated to a 5×5 km$^2$ grid."

In the Met Norway procedure, why aren't temperature *observations* used? Are the HIRLAM reanalyses deemed to be sufficiently certain? This may deserve a few informed comments.

Author Response to RC#3 Jan Verkade

AR: You are right to point out these differences in data sets used for calibration of forecasts and the hydrological model. First, as you mention, SeNorge and Hirlam are not the same data. Hirlam is a short-range regional forecast model (4 km resolution) used in the operational weather forecast for the first 2 days, whereas SeNorge is a dataset where observations are interpolated to a 1 km grid.

In this study, we wanted to use the available operational method from MetNorway, and they use quantile mapping with Hirlam as a reference to calibrate the ECMWF ensemble forecast. Both Hirlam (for the first 2-3 days) and ECMWF (for the following 7-8 days) forecasts are used in the operational weather forecast (yr.no). Using Hirlam data to calibrate ECMWF will improve the transition between the forecasts. Hirlam is available as a sub daily grid and makes it possible for MetNorway to provide different calibration parameters for day and night, whereas SeNorge is only available as a daily grid and would not offer this possibility.

Hirlam have less errors than ECMWF in the temperature forecast for Norway (Engdahl et al. 2015), and as we see from e.g. fig 6 and 7 that the calibration improves especially the cold biases in the ECMWF forecasts. When we evaluated the hydrological model, the temperature calibration improved, in most cases, the hydrological forecasts, providing an indirect confirmation that the HIRLAM temperature is less biased than the ECMWF temperature. Nevertheless, the results suggest that there might be improvements using the SeNorge data instead of Hirlam, but this needs to be tested (beyond the scope of this study).

AC: We will rewrite the following sentence: "MET Norway uses Hirlam (Bengtsson et al., 2017) temperature forecast (on a 4×4 km$^2$) to provide a reference for the parameter estimation (calibration). Hirlam is suitable as a reference since it provides a continuous field covering all of Norway at a sub daily time step. In addition, Hirlam gives a higher skill and are less biased than the ENS (Engdahl et al., 2015).

• If I am correct in understanding that both the raw and the calibrated ensembles have been provided by Met Norway then maybe this should be stated more clearly. Or is it the case that Met Norway computed the calibration parameters on a data set from 2006-2011 and that you applied these yourself to a data set ranging from March 2013 through Dec 2015? If so, maybe state this more bluntly?

AR: Your second suggestion is correct. The raw ensembles from ECMWF (March 2013-Dec 2015) and the calibration parameters (based on data ranging from 2006-2011) were supplied by MetNorway, whereas we did the calibration using the provided calibration parameters and available computer scripts (github/metno/gridpp).

AC: We separate what MetNorway did from what we did. The first paragraph of section 3.1.2 contains the description of calibration parameters from MetNorway, whereas the second paragraphs what we did:

(1) We suggest adding to the first paragraph: "To establish the calibration parameters MetNorway used both ENS re-forecast (Owens, 2018) and Hirlam data from July 2006 to December 2011, both interpolated to a 5×5 km$^2$ grid… "

(2) And, to the second paragraph: "In this study, we applied the calibration coefficients provided by MetNorway to the temperature forecasts for the period 2013-2015. Accordingly, the ENS was interpolated to the 5×5 km$^2$ …."

• I am assuming that you used a HIRLAM reanalysis. Is that correct? If not, what lead times are you using and do the HIRLAM forecasts you used have the same max lead time as the ECMWF ensembles? I am only familiar with a few instances of HIRLAM and these all go out to just over 2 days max.

AR: MetNorway used the operational Hirlam forecasts for the calibration period. It is correct that Hirlam does not cover the same lead times as ENS. Met Norway established the calibration parameters using the 24 first hours of the forecasts as the reference.

AC: We add a sentence page 7, line 12-13 to clarify this: "The same coefficients, based on the first 24h mapped, are applied to all lead times and ensemble members individually. For forecasts outside the observation range, a 1:1 extrapolation is used. That is, if a forecast is 2°C higher than the highest mapped percentile, then the calibrated forecast is 2°C higher than the same percentile for the reference. "

• By off-setting Tens against Tcal, you create the impression that Tcal is not an ensemble forecast. Consider using Traw and Tcal instead.

AR: We chose to use "ens" instead of "raw", since an elevation-correction was applied the forecasts, and hence they are not actually "raw".

AC: We will clarify in the text that the $T_{cal}$ is an ensemble.

• l29-30. The 'assessment' was done by you, not by the ensemble range.

AR: Thank you. We will rephrase this sentence

AC: Suggestion: "In this study, the sharpness was visually assessed by looking at the ensemble range (i.e. the interval spanned by the lowest and highest forecasted values)"

• On assessing sharpness: how confident are you that a visual assessment does the job? Pls consider plotting the empirical distribution of sharpness of all your forecasts and comparing those.

AR: We will plot the empirical distribution of sharpness for all temperature ensembles, and rephrase the sentence concerning sharpness accordingly.

AC: Add a new sentence on sharpness evaluation.

• If you're calibrating the temp ensembles on a leadtime by leadtime basis and on a grid cell by grid cell basis, chances are that you'll change the temporal pattern (forecasted temperature as a function of time) as well as the spatial pattern. Does this in any way affect use in streamflow forecasting? I believe there are some techniques that may be helpful in trying to restore spatial-temporal relations (the Schaake shuffle springs to mind). Would these have a use in present study?

AR: We think that the calibration will not affect the spatial and temporal pattern significantly. The calibration function was applied to each ensemble member individually. We therefore kept the order of the ensemble members, both in space and time, and it was not necessary to use the Schaake shuffle.

AC: We think this will be clearer by adding the following description to quantile mapping page 7, line 12-13. (Response above): "… are applied to all lead times and ensemble members individually…"

**Section 3.2**

• Would it be fair to say that temperature forecasts are only relevant if they can discriminate between freezing and non-freezing situations? If so, would it be justified to focus more on this discrimination? Perhaps by defining an event (T<0, for example) for which one can compute a range of verification scores (false alarms, hits, ROC, Brier's probability score, etc). I acknowledge that this would be feasible for temperature and less obvious for streamflow.

AR: This is a good suggestion. Nonetheless, we think this is beyond the scope of this study. This could be an interesting topic for a future study.

AC: No change

**Section 4**

• " To reduce the amount of presented results, the remaining part of this paper focuses on CRPSS for a lead time of 5 days." This is fine, but temperature forecast at 5-day lead time may not affect streamflow forecasts until a (much) longer lead time. Or conversely, streamflow forecasts at day 5 would have been affected by a day 2 temperature forecast (this is an example). As in some cases you're comparing Q-forecasts with T-forecasts, how have you accounted for this?

AR: This is an interesting question. The streamflow forecast at day 5 will be affected by the temperature forecast the previous 4 days as well as day 5. However, for most catchments in this study, the concentration time is less than one day, and the streamflow will respond the same day as a major water input from rain or snow melt. For specific events, it is not evident which of the T-forecasts at day 1-5 is the most important for the Q-forecast at day 5. The sensitivity depends on the sequence of temperature and precipitation. Nevertheless, we think that using temperature CRPSS for day 5 is a good choice since the streamflow at day 5 is the most sensitive to the temperature at day 5 on average (which applies to all lead times). In addition, we see that the improvement in CRPSS across lead times is highly correlated and our results and conclusions would not change if we used temperature CRPSS for days 2, 3, or 4 instead.

AC: Add a sentence in the discussion: e.g. "The result are robust since most catchment in this study have a concentration time of less than one day."

**Section 4.1**

• In the text, you refer to observed temp as To. In plots, as Tobs. Pls make this consistent. I recommend using Tobs throughout.

AR: Thank you for highlighting the in-consistency in the use of $T_{obs}$ and $T_o$. Since the SeNorge temperature is an interpolated product of the observations, we therefore prefer to use $T_o$.

AC: Changed to $T_o$ in fig 4, and in the text

• L23-25. These sentences are better placed in a discussion section, I think.
AR: OK.
AC: We will move these sentences to the discussion.

• L19 'influence' is missing an 's'. Pls consider replacing by 'affects' though.
AR: Thank you.
AC: We will replacing "influence" with "affect".

**Section 4.2**

"Scatter plots of the difference between CRPSS for calibrated and uncalibrated forecasts". CRPSS in itself is a fairly abstract measure. The difference between two CRPSS scores is, I find, even more abstract. What's the meaning of those values? As CRPSS is a skill of a forecast versus a baseline, why not simply calculate the CRPSS of the calibrated forecasts using the CRPS of the uncalibrated forecasts as a baseline?

AR: We wanted to evaluate the skill of the uncalibrated forecasts as well. If we were to use the uncalibrated as a benchmark, we would not assess the quality of the original forecast, only the change between the uncalibrated and calibrated forecast.

AC: No changes introduced.

**Section 5**

L7: 'dispersion' is not an expression of quality but a characteristic of an ensemble. Saying 'dispersion improved' makes little sense then?

AR: Thank you. What we mean is that dispersion, as measured by rank histogram convexity, improved.

AC: We change to "Even though both bias and dispersion (i.e. reliability) as measured by rank histogram slope and convexity improved with longer lead time, the reduced sharpness and increased uncertainty, resulted in a reduced skill."

**Section 5.1**

• L11 "skill. . . depends". Consider replacing by "skill. . . varies with".

AR: Thank you.

AC: We will change as suggested.

• "Quantile mapping is sensitive to forecasts outside the range of calibration values and period". I think it would be good to point out that this is true for any statistical post-processing procedure.

AR: Good point.

AC: We suggest writing: "Quantile mapping (*as most statistical calibration methods)* is sensitive to forecasts outside the range of calibration values and period (Lafon et al. 2013), this may explain the too high correction in the highest $T_{ens}$ quantile. "

• Immediately following: "and can be a" –> "and *this* can be a"

AR: Noted

AC: Changed as suggested

• On the causes of temperature forecast bias. You go into some detail to explain a situation in which land is colder than sea. Would this be a typical situation for summer/winter? If so, can you more directly link this to some of the results you're showing?

AR: We will clarify that this is a typical situation of winter.  This is to some point already exemplified in the text, and we can underline in the text that the situations are typical for winter. (5.3 will be included in 5.1 and 5.2, and we will ensure to get this information in the revised manuscript):

AC: Add "winter" to the existing text:  "This seasonal cold bias is also clearly seen in the western catchments Viksvatn and Foennerdalsvatn (Fig. 4). The cold bias in $T_{ens}$ along the coast during winter months can be explained by the radiative land heating and cooling in the coarse resolution forecasts (see Sect.5.1) "

**Section 5.2**

L10 Grammatically, this sentence is awkward if not wrong.

AR: Thank you; we will rephrase this sentence.

AC: We suggest rephrasing: "These results show that in order to further improve the skill of streamflow forecasts, improved temperature forecasts during the snowmelt season in spring should be in focus. For spring, the streamflow forecasts are sensitive to temperature forecasts. In this study, however, the temperature forecasts were not, for a majority of the catchments, improved by calibration during spring. Thus, we may expect streamflow forecasts to improve if the temperature forecasts themselves are improved."

**Figures**
**Overall**

Many figures use a lot of white space between various plots/panels. Consider reducing this or, even better, removing altogether.

AR: We will reduce some white space in figure 1 and 3.

AC: New figures provided

**Figure 1**
• Do the grey polygons add up to 139 in total? If so, many must be **really** small?
AR: Yes. Especially on the western coast, the catchments are small. This will be clarified in the caption
AC: New caption text: "Figure 1: The maps for Norway shows the 139 catchments used in this study. The left map show the catchment boundaries including the location of four selected catchments. Please note that many catchments are relatively small and difficult to detect. The location of the catchments' gauging stations are shown in the right map. Norway is grouped into five regions (N=north, M=mid, W=west, S=south, and E=east), and all regions are marked with different colors and regional boundaries."

• Caption: consider using 'boundaries' instead of 'limits'
AR: Thank you, we will use 'boundaries'.
AC: See caption text above.

**Figure 4**
• Why plot the ensemble **mean** and not all five ensemble members, possibly as horizontal lines?
AR: It is not evident to us which modification the reviewer suggests. In this plot, the mean is for the 51 ensemble members not five. If we were to plot all the members, it will be difficult to retain any information. By plotting the mean we show the bias in the forecast and by using the scatter plot, we also show that some biases are dependent on forecasted temperature (a conditional bias).
AC: No changes introduced in the plots.

• The axes of the plots in the right-hand column vary. Please consider unifying this. Also: please consider ensuring that horizontal and vertical axes are identical. Maybe they are, but the labeling isn't.
AR: We will unify the axes.
AC: Changed.

**Figure 5**
• What lead time are these plots for?
AR: Thank you; we will add the lead time in the caption.
AC: Caption update: "All plots are presented for lead time 5 days."

• Is the lead time for T identical to that for Q? What is the 'response time' of the catchment to snowmelt?
If not zero then shouldn't this be taken into account somehow?
AR: We use the same lead time for temperature as for streamflow.  See comment to section 4.
AC: No changes applied.

Please consider. . .
• . . . removing data for seasons for which temperature has little or no effect on streamflow levels.
AR: We would like to keep the plots for all seasons here. By showing the difference between the seasons, we think it is easier to understand the large variations we see.
AC: No changes applied

• . . . unifying horizontal and vertical axes. it took me a little while longer than I cared to realise that the light grey slanted line is the 1:1 diagonal.
AR: We will consider changing the plots. However, unified axes means that we lose information about the regional distribution. An alternative plot with unified axes is presented below.
AC: We prefer to keep the plot as is.

[Figure]

**Figure 6**

• What do you want the reader to compare? CRPSS(T) and CRPSS(Q)? Or CRPSS(spring) v CRPSS(autumn)? Pls ensure panels are ordered accordingly.

AR: We wanted the reader, first of all, to compare CRPSS(T) and CRPSS(Q) Therefore, we placed CRPSS(T) and CRPSS(q) from the spring season on the first line and for the autumn season in the last line. Then the reader can evaluate how the improvements in temperature will affect improvement in streamflow, for both seasons. Secondary, we wanted to show the difference between seasons. Sub-plots for each season are therefore arranged vertically, for both temperature (left) and streamflow (right).

AC: No changes introduced.

• pls ensure that within a row, panels have identical vertical axes so this comparison can indeed be done (i.e. the reader can then easily compare the top left with the top right plot)

AR: We prefer to use different scales on the vertical axes within a row to increase the readability of each sub-plot. In particular, the plots of the CRPSS(Q) would be more difficult to read if we used the same scale as in the plots of CRPSS(T) in the left panel.

AC: No changes

**Figure 10**

• The background colours have an effect on the colouring of Qens and Qcal. Please consider removing the background shades. Maybe replace these by threshold lines only?

AR: Thank you, this will be done

AC: Changed in plot, and in text: p10l28-30: "The horizontal grey dotted lines represent mean annual flood, and the 5-year and 50-year return level for floods in this catchment."

• Please consider removing the number of lines in the plot, for example by only showing a shaded area with no line at the edges thereof.

AR: Thank you

AC: We change as suggested

• What is the purpose of showing both the 'real' observations and the 'model streamflow with SeNorge observations'? Is this distinction made in the paper, and addressed?

AR: We understand that the introduction of real observations in this figure is confusing, and we will therefore remove the real observation from the figure and from the text.
AC: New plot does not include observation. Need also to rewrite some parts of the text.

• Consider reversing the order of the graphs. The 9d lead time graph was available before the 2d lead time graph?
AR: Thank you, we will change the order of the graphs.

• The horizontal axis labeling is not in English.
AR: Thank you, we will change the labeling to English in al plots
AC: Labeling changed to English

• As all horizontal axes are identical, pls consider removing white space between plots altogether and only label the axis of the bottom plot.
AR: Thank you, this will be done
AC: Label axis only on the bottom, and added (a)-(c) to the different panels. Reduced the amount of white space.

• The warning levels aren't relevant, are they? On reflection: you're scoring the forecast ensembles using
CRPSS and rank histograms. This shows absence of preference for doing well for 'extremes', even though the work appears to be inspired by forecasting for floods. How is this consistent? Maybe omit references to 'floods' altogether?
AR: In Norway, we use the mean annual, the 5-year and the 50-year floods as exceedance thresholds to issue flood warnings. This figure connects the theoretical aspects to the operational implementation, and points to the importance of calibrated temperature for a flood warning system.
AC: We prefer to keep reference to flood levels, but remove the warning colors all together.

[Figure]

New Reference:

Engdahl, B. J. K and Homleid, M: Verification of Experimental and Operational Weather Prediction Models December 2014 to February 2015. Norwegian Meteorological Institute, METinfo (18/2015), 2015

---

## Author Comment (AC4) · 20 Dec 2018

Author Response to RC#4

Thank you for the positive and thorough evaluation of our article. We appreciate the comments, which are valuable for us in order to improve the manuscript.

We would like to apology for the missing references. The error emerged when we specified the HESS format, and un-intentionally deleted many references from the reference list. The main author should nonetheless have detected this flaw prior to posting.

Replies and corrections are done as follows: the Author response (AR) is marked with red text, while the author's suggestions to corrections (AC) are marked with blue text; we use page and line number to specify the appropriate location, where this is needed. Referee comments are kept in a black text.

**Review of 'Streamflow forecast sensitivity to air temperature forecast calibration for 139 Norwegian catchments' by Trine Hegdahl et al.**
**Anonym referee#4**

Supplement:

Especially in hydrometeorological predictions where methods from both the meteorological and the hydrological forecasting community are used, it is of major importance to carefully define the terminology and to coherently use throughout the manuscript.

The current form of the manuscript shows a lack of precise formulations (e.g.: calibration, pre-processing, skill) which should be revised to better communicate the content of the study. Some of the graphics should be enhanced to facilitate the readability and the caption are sometimes incomplete. In addition, more than 15 references mentioned in the text are missing in the reference list and should be added.

AR: We thank you for the feedback. We would like to apology for the missing references. It seems to have been an error when we reformatted EndNote, which evidently lead to many references being deleted from the reference list. The main author should nonetheless have detected this flaw prior to submitting the manuscript. We will carefully revise the text to avoid inaccuracies in formulations.

Furthermore, some additional references could be of interest within discussion to put the findings of the study into a broader picture. Many of the references especially concerning the meteorological forecast are user guides and or technical reports or personal communications, which is fine, but I would appreciate if some more peer-reviewed literature would be cited as there is a large body of existing literature concerning the verification of ECMWF temperature predictions.

AR: We agree that it is better to use peer-reviewed literature. We chose to use technical reports and personal communication only when necessary and we found no other alternatives. In particular, there are not much peer-reviewed papers on the verification of the ECMWF temperature forecasts for Norway available. Hence, we chose to implement what is available of technical documentation.

In general, the language could be clearer and more concise. To me it is not clear what the authors understand under the term pre-processing, at least in the beginning of the manuscript. E.g. in literature there is a distinction between dynamical and statistical downscaling (see e.g. Li et al. (2017), Yuan et al. (2015)) and statistical downscaling does include a bias-correction. In the present manuscript, the term downscaling does only refer to applying a laps rate correction and interpolation, what is not as downscaling is referred to in the literature.

However, I think it would be important for the reader to have a short general overview of what pre-processing is in the introduction. In particular the term calibration, in the present manuscript used as a

synonym for bias correction should be introduced more carefully because the term calibration is used by statisticians but in the meteorological, climatological and hydrological communities, the term bias correction is more common.

AR: We acknowledge that the literature is not consistent in terminology, and particular the terminology differs between the forecasting and the climate projection communities.

In our paper, we chose a terminology that is consistent with a large part of the literature, and that facilitates to explain the approaches we used. We use pre (and post)-processing as a general term, which includes all techniques applied to the raw temperature forecasts in order to improve the temperature output from the atmospheric model (i.e. downscaling and calibration are pre-processing techniques). We pre-processed the temperature in two ways: (i) only downscaling, (ii) both downscaling and calibration, with the purpose to reveal the effect of temperature calibration.

We used the term downscaling on the resampling from a low resolution for the ECMWF forecasts to the 1x1km grid used for the SeNorge data, combined with a temperature correction using a temperature lapse rate. This terminology is used by e.g. UK Met office (Sheridan et al 2010, with references therein). Especially for areas with a complex terrain, where the resolution of the NWP poorly resolves the terrain, the correction for the discrepancy between model elevation and terrain are useful. In some literature, the term downscaling includes both bias correction and resampling, (ref Yuan et al 2015), but we did not use this terminology here.

We used the term calibration on the statistical adjustments of bias and dispersion of the ensembles. The aim of calibration is to make the forecasts reliable in a statistical sense, i.e. 90% of the observations are within a 90% uncertainty interval. In particular, in the meteorological forecasting literature, calibration has this specific meaning (e.g. Gneiting, 2006)

We think that to separate the post-processing into downscaling and calibration is useful, but agree that the term downscaling might have a different signification in parts of the literature. Our terminology is also, to a large degree, in accordance to the descriptions in Li et al (2017). Lie et al (2017) describes the main purposes of post-processing to be the following (1) correct bias and dispersion in the forecasts, (2) to preserve the predictive skill of the forecasts, (3) downscale the forecasts to the scale used in the applications, and (4) to generate ensemble members (…). Further, in the conclusion Li et al (2017) writes that their purpose is "… to calibrate the bias …" In the referred article, we hence see the term calibrate used consistently to describe the statistical properties of both the meteorological and the hydrological ensembles. We further think that using calibration, as part of the pre- and post-processing is a well-established term for the hydrological community using ensemble forecasts. Calibrated ensembles and the calibration methods is more specific than only using only the term pre- or post-processing. Calibration strive for the ensemble to describe the mean and spread of the climatology they should represent.

We have not included any description of the dynamical downscaling, as this usually includes a regional climate model with a different approach, and is not the scope for this study.

AC: We would prefer to keep the notations as is, but we will provide a more detailed description of how we define the term pre-processing, and our use of the terms calibration and downscaling.

As you mention the forecasting period used for the study is only two and a half year long which might influence the results. You state this in the discussion but do not explain why it could be critical. I suggest that you discuss this explicitly. Namely, within such a short period, the interannual variability might not be sufficiently covered. In addition, using forecasts from different model Cycles (38r1 to 41r1) might have an influence of the skill as well because the adaption within a new cycle might enhance or

decrease the forecast performance making the comparison between seasons difficult as it might not only originate from the particular season but might be influenced by model versions. I suggest including such limitations in the discussion.

AR: We agree that the inter-annual variability might affect the calibration coefficients, and of course, there are aspects with the different model version that might affect the result. However, the changes applied to the different model-cycles did not remove the biases apparent in temperature forecasts (fig 4).

AC: We will address the above mentioned explicit in the discussion.

To apply Quantile mapping you do need the distribution of the forecast and the distribution of the observations. In section 3.1.2 you state that "MET Norway uses Hirlam temperature forecasts to provide the observational climatology used for parameter estimation". I think here, more information is needed to enable the reader how the calibration is done. Are daily values used for the parameter estimation? Is it empirical or parametric QM used and how are values outside the range treaded (e.g. constant extrapolation)?

Is it a member-by-member approach or are the same parameters used for all members?

AR: MetNorway uses parametric quantile mapping based on the first 24h. When a forecast is outside the observation range, a 1 to 1 extrapolation is used. Therefore, if a forecast is 2°C higher than the highest percentile, then the calibrated forecast is 2°C higher than the same percentile for the reference. The same parameters are applied to all members and lead times.

AC: We add a sentence page 7, line 12-13 to clarify this: "The same coefficients, based on the first 24h mapped, are applied to all lead times and ensemble members individually. For forecasts outside the observation range, a 1:1 extrapolation is used. I.e. if a forecast is 2°C higher than the highest mapped percentile, then the calibrated forecast is 2°C higher than the same percentile for the reference. "

One critical point is that the calibration parameters are interfered from the Hirlam but the hydrological model is run with SeNorge observations. Why are not these observations used? The correction will account for the bias between ECMWF and Hirlam but I would expect that biases with SeNorge will at least slightly differ. Why don't you use the observations from SeNorge to get your calibrations?

In the summary it is stated that "The most obvious improvement in the forecasting chain is to use the same temperature information, the SeNorge temperature, for calibrating the temperature forecast that is used for calibrationg the hydrological model, generating …" (P14/L25-27).

But if I understand correctly from the manuscript SeNorge and Hrilam are not the same. I have troubles with this procedure as it is known that different forecast models do have different biases. To bias-correct or calibrate ensembles the observations should be taken into account and not another forecast. In this case the bias between two forecasts will be corrected and not the bias of the forecast with regard to the observations.

AR: You are right to point out these differences in data sets used for calibration of forecasts and the hydrological model. First, as you mention, SeNorge and Hirlam are not the same data. Hirlam is a short-range regional forecast model (4 km horizontal resolution) used in the operational weather forecast for the first 2 days, whereas SeNorge is a dataset where observations are interpolated to a 1 km grid.

In this study, we wanted to use the available operational method from MetNorway, and they used quantile mapping with Hirlam as a reference to calibrate the ECMWF ensemble forecast. Both Hirlam (for the first 2-3 days) and ECMWF (for the following 7-8 days) forecasts are used in the operational

weather forecast (yr.no). Using Hirlam data to calibrate ECMWF will improve the transition between the forecasts. Hirlam is available as a sub daily grid and makes it possible for MetNorway to provide different calibration parameters for day and night, whereas SeNorge is only available as a daily grid and would not offer this possibility.

Hirlam has less (smaller) errors than ECMWF in the temperature forecast for Norway (Engdahl et al. 2015), and as we see from e.g. fig 6 and 7 in this manuscript, the calibration reduces the cold biases in the ECMWF forecasts. When we evaluated the hydrological model, the temperature calibration improved in most cases the hydrological forecasts, providing an indirect conformation that the Hirlam temperature is less biased than the ECMWF temperature.

Furthermore, by the many interpolations used, there is a large uncertainty introduced which will lower the trust in the results. Interpolation of ECMEF and Hirlam to derive correction parameters, another interpolation to meet the hydrological model requirements.

AR: We agree that there are uncertainties due to interpolation and downscaling. A temperature calibration that is tailored to the needs for the hydrological modelling would solve this challenge.

AC: We will add a sentence on this in the discussion.

Another point that should be discussed is if seasonal correction parameters are really sufficient or does it introduce artificial jumps between periods. In a climate context, seasonal windows for parameter estimation might be sufficient but in an operational forecasting context a shorter window should be taken into account if possible.

AR: MetNorway provided unique parameters for each month. The parameters are based on a window of three months, which smooths the seasonal patterns. A three month window was chosen to ensure enough data for robust calibration parameters.

In Section 3.2 where the CRPS is introduced you mention different notations (CRPS, Scrp) and same for the CRPSS. I think this is confusing, as later in the text only CRPS is used. I suggest only introducing one of the notations and stick to that.

AR: We agree that this notation might introduce confusion. The reason is the formatting standard of HESS where equations should only contain one capital letter with sub or super script. However, we find it appropriate to use CRPSS in the text since this is the abbreviation used in the community, and in the equations, we used an alternative notation according to the HESS standard: ($S_{crp}$ and $S_{crps}$ are only used in the equations). This approach is used in many HESS papers.

AC: No change introduced to the manuscript.

Specific comments:

P1

L7-14: You say the flood forecasting system uses deterministic forecasts for temperature and precipitation). But the ECMWF model you reference provides an ensemble of 51 members. Please state how this is used.

AR: The operational system today, uses one deterministic forecast, not the ensemble forecasts. In our setup, the hydrological system is setup to run the 51 ensemble members.  We make sure that the same

initial states are used for all members. This is explained in details in the main text, and in the abstract we keep the description simple. We think the suggested changes in the following point also covers this point.
AC: We suggest clarifying this in section 3.1. P6, l24: "In the forecasting mode each temperature ensemble member was used as input and run as separate deterministic forecasts."

L11-12: "An alternative approach is to use meteorological and hydrological ensemble forecasts" is somewhat misleading. Either you used ensemble meteorological forecasts in combination with hydrological models to generate ensemble streamflow forecasts or one uses a different methodology to produce hydrological ensembles forecasts. I suggest rewriting the sentence: "An alternative approach is to combine meteorological ensemble forecast with hydrological models to quantify the uncertainty in the forecasted streamflow".

AR: You are right. We apply the suggested rewriting.
AC: Applied the suggested rewriting

L14: "for an accurate forecasting of ", or "to accurately forecast streamflows"
L15: Ensemble forecast of temperature from the ECMWF "
L16: "to improve the skill and reduce biases"

AR: Thank you. We include the suggestion L14, L15, and L16
AC: We will apply the corrections

L18: why do you mention precipitation here? If it is not used for the calibration I would avoid it here.

AR: We mention precipitation since the "observed" precipitation and temperature was used to calculate the initial states of the hydrological model until the forecast issue day. We will consider omitting the sentence about SeNorge in the abstract. Ref RC#3, and discussion on abstract.
AC: We will omit from the abstract: "Estimated observed daily temperature and precipitation were obtained from the SeNorge-dataset, which is station data interpolated to a 1×1 km2 grid covering all of Norway."

L20: was used to calculate the streamflow

AR: Thank you. We include the suggestion
AC: The sentence will be corrected

P2

L1: Floods can damage… and can have a high …

L5: componentS

AR: Thank you.
AC: We will change to the above as suggested.

L9: The reference "Müller et al." is missing in reference list

L14: Both reference "Langsrud 1998 a and b" are missing.

AR: Thank you.

Author Response to RC#4

AC: We will update the reference list

L16: as a means to account for uncertainty in the forcing.

AR: Thank you.
AC: The sentence will be corrected

L21: The Reference Cloke & Pappenberger, 2009 and Wetterhall et al., 2013 are missing

L25: the ensembles can be calibrated

L26: Hamill and Colucci, 1997 and Buizza et al, 2005 are both missing

L29: Gneiting et al. 2005 is missing, Wilks and Hamill 2007 is missing, Raftery et al. 2005 is missing

L30: Evens 2003 is missing

L31: Gneiting et al. 2005 is missing, Wilks and Hamill 2007 is missing. The order of the references is different compared to L29. Bremnes, 2007 is missing.

AR: All references are now included.
AC: Thank you. We will update the reference list. In addition "Wang and Bishop"

L31-32: This sentence is very general, it is arbitrary clear that different correction methods do correct the biases differently. I suggest either being more specific about single methods, or to summaries different methods to provide a better overview for the reader instead of listing available techniques. Maybe cite some standard books for statistical bias correction and downscaling (Wilks, 2011) and for forecast verification (Jolliffe & Stephenson, 2011).

AR: We will cite some standard books and papers that provides reviews of forecast calibration methods.
AC: We will add the following sentence at the end of the paragraph: "A recent review of calibration methods are given in Li et al (2017) and the text book edited by Vanniitsem et al (2018)
- Vannitsem,S. Daniel S. Wilks, Jakob W. Messner, Editor(s): (2018) Statistical Postprocessing of Ensemble Forecasts, Elsevier, ISBN 9780128123720, doi: 10.1016/B978-0-12-812372-0.09988-X.
- Li, W., Duan, Q., Miao, C., Ye, A., Gong, W., & Di, Z. (2017). A review on statistical postprocessing methods for hydrometeorological ensemble forecasting. Wiley Interdisciplinary Reviews: Water, 4(December), e1246.  https://doi.org/10.1002/wat2.1246

P3

L1: snow cover without "−"

AR: Thank you.
AC: We will correct as suggested.

L2-4: This sentence is unclear to me. Can you elaborate what you mean?

AR: We mean that an improvement in temperature forecast will not necessarily translate directly into an improvement of streamflow forecast. If temperatures are well below zero, an improvement in temperature forecasts has no effect on the streamflow forecasts, whereas for temperatures around zero degrees, the streamflow is very sensitive to temperature, in particular when it might turn on or of rain and/or snow melt.

AC: We will rewrite as follows: "The sensitivity of daily streamflow to temperature is non-linear since streamflow depends on temperature thresholds for rain/snow partitioning and for snow melt/freeze processes. The latter depends on the state of the system, i.e. snow is needed to generate snowmelt. For temperatures well below 0$^o$C, the streamflow is not sensitive to temperature, whereas for temperatures around 0$^o$C relatively small changes in temperature might control if the precipitation falls as rain or snow, and consequently, whether streamflow is generated or not."

L5: Gragne, 2015 . missing reference

AR: We will not use this reference in the modified manuscript
AC: The reference will not be used.

L7-8: Forecasting, downscaling and interpolation are three completely different things and the challenge is connected to much more than laps rate. For interpolation and downscaling a large part can be attributed to temperature height correction which depend to a large degree to laps rates. But forecasting of temperature is far more complex and related to chaos theory.

Rephrase please.

AR: You are quite right. We should not have included forecasting in this sentence. We are addressing the downscaling and interpolation of forecasts.
AC: We will remove the word "forecasting" from the sentence.

L9: Again, missing references: Aguado and Burt, 2010; Pagès and Miro, 2010, Peter et al., 2010.

AR: Thanks. We see that in the case of Peter, this is the first name, it should have been Sheridan et al.
AC: All references are corrected and added.

L13: Alpine (capital A) as the study looks at catchments in the Alps.

L15: ", found only modest….",

AR: Thank you.
AC: We will correct the above as suggested.

L17: I think the effect is not marginal, as you later on show with your results.

AR: We used marginal to separate the effect of temperature from that of precipitation. We will change the sentence to 'the isolated effect of…'
AC: We will change to 'the isolated effect of'

L26: do you mean from both, the hydrological and the meteorological perspective?

AR: Yes, we do. This will be clarified in the manuscript.

AC: We change to "Are there spatial patterns in the temperature and streamflow ensemble forecast skill and if so, can these be related to catchment characteristics?"

L27: from the ECMWF, in addition I would mention the lead time here but maybe not the MET Norway pre-processing setup as you use the QM to pre-process the forecasts which is, if I understood correctly, not yet part of the pre-processing setup at MET Norway.

AR: The information in line 27 is correct. The QM is (was) a part the operational pre-processing chain at MET Norway and is used at the forecast published at yr.no. We chose to not mention lead time here since the choice to focus on lead time 5 days was based on preliminary results.
AC: In section 3.1.2 we add one sentence to clarify: "MET Norway provided temperature grid calibration parameters used in this study. This grid calibration was used in the operational post-processing chain for meteorological forecast including the forecasts published on yr.no."

L28: Are the retrospective forecasts operational forecast for the period within 2013-2015? This could be misleading for readers or misinterpreted as reforecasts (or hindcasts) which are forecasts for the same day as the operational forecast but for the past 20 years using re-analyses for the initialization. Maybe rephrase to avoid any misinterpretation.

AR: We chose retrospective to underline that we used the operational forecasts in retrospect. Nevertheless, we understand that this can be misinterpreted. We will rephrase the sentence.
AC: We will rewrite the sentence as follows: «Three years of operational ECMWF forecasts from 2013-2015 were used to re-generate streamflow forecasts, and the skill of temperature and streamflow forecasts were systematically evaluated for these catchments.

L30: again, I think marginal is the wrong word, if the effect is assumed to be marginal, why should you analyze it in such detail.

AR: OK
AC: We change "marginal" to "isolated"

L31: Not clear to me. Do you mean that the observed precipitation is used to drive the hydrological model? Specify that to make it clearer.

AR: Yes. The observed precipitation is used to drive the hydrological model.We will rephrase to make clearer
AC: We will rewrite the sentence as follows:
"To investigate the isolated effect of the temperature ensembles on the streamflow forecasts, the observed SeNorge precipitation (Tveito et al., 2005) was used instead of the precipitation ensemble forecasts when we re-generated streamflow forecasts."

L33-P4L2: Maybe combine this with the preceding paragraph. This would make it less generic.

AR: We will join the two paragraphs as suggested.
AC: The two paragraphs will be joined.

P4

Author Response to RC#4

L5: spatial variations

AR: Thank you.
AC: We will change the sentence as suggested.

L6: rather high then steep?

AR: The Mountains are both high and steep. However, we think that steep is the most important description of the high elevation gradients in the area.
AC: We make no changes in the manuscript.

L9: delete "flows"

AR: Thank you.
AC: The word will be corrected in the manuscript.

L18: the smallest catchment has an area of only 3 km^2? Or is it a typo?

AR: This is not a typo. There are several small catchments in our dataset, but only one of this size.
AC: there will be no changes in the manuscript

L21: what are the selection criteria for "data of sufficient quality"
AR: This was inaccurate description since the catchments disregarded from the study was due to different reasons, both data retrieving and technical problems. For three catchments, we had problems running the model with the reference data, one catchments there was an issue with the elevation correction, and for two catchments, there were technical problems during the regional analysis. We have a large dataset, so the exclusion of the six catchments will not change our conclusions.
AC: We suggest writing: "Of the 145 flood forecasting catchments, 139 were chosen as the basis for the study (Fig. 1)."

L27: "og" seems to be Norwegian

AR: Thank you.
AC: The word will be corrected.

L31: snowmelt driven flood event

AR: Thank you.
AC: The sentence will be corrected as suggested.

P5

L5: write "available at SeNorge.no"

AR: Thank you.
AC: the sentence will be corrected.

L7: Mention what kind of interpolation is used (bilinear, kriging, …)

AR: The SeNorge temperature is interpolated using kriging on de-trended temperature using standard temperature lapse rates.
AC: We rewrite the sentence to "For this version, gridded temperature is calculated by kriging, where both the elevation and location of temperature stations are accounted for."

Section 2.2.1:

Mention here that you use the precipitation data from this data set as a substitute of the precipitation forecasts (if this is the case).

AR: Thank you. That is a good suggestion.
AC:  We will add a sentence at the end of the paragraph: page 5, line 12
"The SeNorge precipitation substitutes the precipitation forecasts in the ensemble forecasting chain, to reveal the isolated effect of temperature calibration on streamflow forecasts. (see section 3.1 for more details)"

L15: constitutes as the basis

AR: We prefer to keep the sentence as it is.
AC: No changes will be introduced in the manuscript,

L20: explain what PEST is.

AR: We will modify the sentence and explain what PEST is.
AC: The sentence will be rewritten as follows:  "... which has been calibrated using the PEST software to establish model parameters (Doherty, 2015) …"

L21: Abbreviation NS (for Nash-Sutcliffe) not introduced before.

AR: Thank you. We will be corrected in the manuscript.
AC: The abbreviation 'NS' will be changed to Nash-Sutcliffe

Section 2.2.2:

Is the calibration done for each catchment separately? Do the given values for the NS coefficient represent the mean for all catchments? Is this good? Please state how these values translate into performance compared with other hydrological models.

AR: The calibration is done for each catchment separately. The mean is presented to give an impression of the performance, and of course, there is a great difference in  the NS-score between the catchments. We think that NS between 0.73 and 0.77 is ok. Within the range of NS-scores there are of course catchments where the models performs less optimal. Other models applied to the same catchments has a very similar performance, indicating that the quality of data (precipiptation, temperature and streamflow) is an important contribution to model uncertainty. Since we in this paper use the model streamflow in stead of the observed streamflows for evaluation of forecast, we think it is not necessary to provide more details on the calibration of the hydrological model   .
AC: No changes introduced in the manuscript.

Author Response to RC#4

L22: Missing Reference Gusong (2013), In reference list only Gusong 2016 is listed

AR: Thank you.
AC: The reference will be corrected to Gusong (2016).

2.2.3

To make this more coherent I suggest renaming this section into "Reference observations" (or similar) and in the latter part of the study refer to reference observations as well. Otherwise it is difficult to distinguish between the model stream flow and forecasted streamflow. E.g. on P6 L13 you write reference model run, I assume this is the same as model streamflow? This is somewhat confusing if you state it twice in 2 different paragraphs.

AR: Thank you. We will change "model streamflow" to "reference streamflow" in the section title and in the text.
AC: We will changed all "model streamflow" to "reference streamflow".

P6

L6: write the lead time as well in days 246 hrs (i.e. 10 days). Why is it 246 hrs and not 240?

AR: We used lead time 246 hours since we have used the forecast issued at 00:00 aggregated to daily values for the time period 06-06. We can change this to days.
AC: We add one sentence on line 7 to clarify this: "We used the forecast issued at 00:00 and aggregated daily values for the meteorological 24-hour period defined as 06:00-06:00 to provide forecasts for lead times up to nine days."

L7: The Reference "ECMWF (2018a)" does only provide the documentation and support page of the ECMWF. The Specific documentations can be downloaded. The scientific basis of the ENS system has been discussed in multiple publications and it might be worth to reference some of them and point to this documentation for specific points only.

AR: We would like to keep the sentence and reference as it is, since this is provides a detailed overview of the model cycles. We provide an additional sentence, including references, to the description of ECMWF.
AC: We add the following sentence to the end of this paragraph: "A more detailed description of the ECMWF ENS system is provided in e.g. Buizza et al. (1999) and Persson (2015)."

L8: "the ensemble members of ENS are…"

L9: "with different perturbed conditions to represent the …"

AR: Thank you.
AC: The sentences will be changed as suggested

Author Response to RC#4

3.1

See comment to 2.2.3. I don't get the difference between model streamflow and reference model run. If I understand correctly these are the same. If so, only describe it in one section. I think here it would be suitable. Reference run = model streamflow, use the same terminology if it is the same.

AR: We will change to 'model streamflow' to 'reference streamflow', but be prefer to keep section 2.2.3- since we in section 2 describes the data and models, whereas in section 3 we describe how we used the data.
AC: "model streamflow" will be changed to "reference streamflow"

Are the ENS forecast temporally aggregated as well?

AR: The ENS are also temporally aggregated. Ref p7 l1-2 (3.1.1) and l15-16 (3.1.2), and fig 2.
AC: We will add one sentence in line 20: "all temperature forecasts were aggregated to daily values"

L25: replace "include" with "referred to as"

AR: Thank you.
AC: We will change "include" to "refers to"

L27: Use the same units for both grids. ° or km^2. Best would be use both units for both grids, one of them in brackets.

AR: We think it is more accurate to use use degress for the ECMWF grid, but we will add a parenthesis with the grid resolution in km. Hence, we use degrees and km for EC, only km for SeNorge
AC: We will change as follows: " ... resolution of 0.25° ( ~ 30km)"

3.1.1

What is the rationale behind the choice of using a nearest neighbour technique?

AR: We tested also other techniques, e.g. bilinear interpolation, which has a higher computational demand and creates larger output files, than the nearest neighbor interpolation. Since the quality of the forecasts temperature was almost similar, the reduced computing time and smaller storage requirements made the nearest neighbor method more useful.
AC: We introduce no changes in the manuscript.

P7

L4: Bremnes 2007, 2004 are missing in the reference list.

AR: Thank you.
AC: The references will be added.

L8: Can you give a reference for the sentence "gives a higher skill and are less biased"

AR: The reference is Engdahl et al 2015
AC: We include this in the text and in the reference list.

Author Response to RC#4

L20: Ensemble forecast verification does not only focus on reliability and sharpness. Therefore, different measures need to be taken into account (as well biases are important).

AR: In this sentence we refer to a specific paper where the reliability and sharpness is used for evaluation of forecasts. We also think the bias is a part of the evaluation according to reliability. If the forecast is biased it will not be reliable.  In the rank-histogram decomposition slope will identify bias in the forecasts.
AC: We introduce no changes in the manuscript.

L30: "lowest and highest forecasted value" does it mean the minimum and maximum? Why not the 10th and 90th percentile and the interquartile range. I think this gives a better estimate of the sharpness of the forecast as it does not only account for the most extreme members.

AR: we agree that specific interquartile range might be a more robust measure for sharpness. Nevertheless, using inter-quantile range does not change the choice we made, and we introduce therefore no changes in the manuscript.
AC: No changes introduced in the manuscript.

P8

L12: I would rephrase the sentence. "which a skilful forecast should outperform" and write it in a single sentence.

AR: We think the sentence is fine as it is.
AC: We introduce no changes in the manuscript.

L18: negative values mean (without s)

AR: Thank you.
AC: The word will be corrected as suggested.

L19: "which perform similar to the reference forecast (climatology in this case)"

AR: Thank you.
AC: We will change to "implies that it performs similar to the benchmark (climatology in this case)".

L20: Do you use here the mean of the daily CRPS? (CRPS with overbar?)

AR: Yes, in this case it refers to calculating the average ($\overline{CRPS}$) over all daily CRPS (without an overbar), for the months in question.
AC: No changes will be introduced in the manuscript.

L25-26: This sentence seems to be wrong.

AR: We will reformulate the sentence.
AC: We rephrase: "Finally, we used linear regression to identify relationships between catchment characteristics (here elevation difference and catchment area) and the skill score ($T_{cal}$ and $Q_{cal}$ CRPSS)"

L27: Usually seasons are aggregate in winter = December-February (DJF), spring = mar-may (MAM) and so on. Can you explain your motivation to choose this definition of seasonal aggregation?

AR: You are right about the usual definition of seasons. We used a different definition since we wanted to isolate a snow melt season, that for most catchment most catchments is in the period April to June. . We think this better seasonal description for streamflow in Norway.

AC: We will add one sentence: "We used this definition of season to better capture a snow melt season that for most catchments in our case study is in the period April to June."

P9

L8: as shown in figure…

L9: no comma after "convexity"

AR: Thank you.

AC: We will change the sentences as suggested.

The description about the slope and complexity is hard to follow. Could you give an example what the values really tell, e.g. how does a rank histogram look like with a complexity of 2000? I think rank histograms are very useful to be used for visually interpretation and the complexity and slope somehow lead to a reduction of the usefulness of the rank histogram at least to people not familiar with these parameters.

AR: We used the convexity and slope since then it is much easier to provide aggregated information of forecast performance. In or results, we do not focus on the values in themselves; the change of the values is the important information. We find that Jolliffe and Primo, 2007 provide detailed information.

AC: We will add one sentence to explain this better:
"As shown in Fig. 3, temperature slope and convexity, improve with increasing lead time, whereas CRPSS gets poorer. The improvement in slope reflects that the under-estimation in the raw ensemble is improved, whereas the improvement in convexity reflects that the under-dispersion is the original forecast is improved. For streamflow, slope gets poorer, convexity improves, whereas CRPSS shows small changes with lead time. "

L15: I recommend repeating what TO and Tens is to enhance the flow in the text.

L27: Same here, mention the abbreviation in brackets in the text to help the reader.

AR: For both comments above, we will repeat the meaning of abbreviations in the beginning of each section.

AC: The sentences will be corrected as suggested.

Author Response to RC#4

L29: "influenceS" ; Do you mean in streamflow skill or CRPSS?

AR: All skill is measured by CRPSS.
AC: Changed to " affects" Ref RC#3

P10

L4: Do you know why there is no improvement during summer by using calibrated temperatures? Is it due to the absence of snow / snow-melt in summer?

AR: There are two reasons for the small changes during summer (i) the skill of uncalibrated temperature forecasts are higher in summer and (ii) there is less or no snow in summer, and  that will reduce the streamflow sensitivity to temperature.
AC: We will add one sentence: "Two explanations for the small changes in CRPSS during summer are (i) the skill of uncalibrated temperature forecasts are higher and the potential for improvement is lower, and (ii) there is less or no snow in summer, resulting in a reduced streamflow sensitivity to temperature."

L5: You often mention the Figure number in the last part of the sentence. I personally would prefer this information first what makes it easier to follow the text and figures at once; It reveals

AR: We try to vary the placing of the figure number in a sentence and it is a question of style / preference.
AC: We make no specific changes in the manuscript.

L20: What is the significance level you used? I would mention this in the text.

AR: For the slope of the regression lines being different from zero we used a significance level of p-values < 0,05. This information is available in the caption text for fig. 9. We will consider including this in the text.
AC: We will include the following sentence: "By indicating the significance and sign of the relationships, significant relationships were found for 12 out of 40 regression equations (5% significance level)."

4.3

What are the criteria you used to choose this flood event in May 2016? Mention the motivation for this specific event.

AR: We wanted to present a snowmelt flood event during spring and the selected event in  May 2013 in Bulken was a snowmelt flood.
AC: p10, l27: changed from "2014" to "2013".

Author Response to RC#4

L10: If possible embed this in the floating text and see separate comment to the figure.

AR: We are not certain to which line this comment refer.
AC: We added "target days", to ensure a consistency to figure 10.

P11:

L1: to make it clear I would add: "…increases with lead time (form upper to lower panel)." Linked to my comment on the caption in Figure 10 that it could be misinterpreted as a continuous forecast starting at may 16th.

AR: Thank you. We will modify the sentence as suggested.
AC: We will change the sentence and add "from lower to upper panel". Since we have changed the order of panels as suggested in RC#3.

L4: "The box plots … show" (show without s)

AR: Thank you.
AC: The word will be corrected.

5 Discussion

Here I would again use words instead of the Tens Tcal only: "Both raw (Tens) and calibrated (Tens) temperature forecasts were more skilful with …". I think it makes the text more interesting to read. This could be adapted in different parts of the Manuscript, in the beginning of each section this should be repeated.

AR: We will introduce the abbreviations in the beginning of the sections
AC: We will changed according to suggestion.

L5-9: "Overall, the grid calibration of temperature had a positive effect on both …", but the lines before it states "…, resulted in reduced skill". This is somehow contradictive, could you make this clearer?

AR: The last sentence refer to the difference between raw and calibrated ensembles for all lead times, and we see that the grid calibration improves the performance for most scores and lead times. The previous statements are related to the development of performance for increased lead times. In short, the CRPSS is reduced for increased lead time, it is better for calibrated than raw ensembles.
AC: We will change the sentence: "Overall, the grid calibration of temperature had a positive effect on both temperature and streamflow for most validation scores and lead times."

L18: missing reference Lafon et al. 2013

L20: L24: wrong citation format Ivar Seierstad et al. (2016)

AR: Thank you.
AC: The reference will be added and the citation will be corrected.

Author Response to RC#4

Subtitles for 5.1 and 5.2 should be coherent "calibration for …" or "calibration for the…"

AR: Thank you.
AC: The subtitles will be corrected as suggested

L26: forecasts

AR: Thank you.
AC: The word will be corrected.

P12

L4: "Hence, calculated … " word at wrong place within sentence.

AR: Thank you.
AC: The sentence will be corrected to "Hence, estimated streamflow has a high…"

L7 "indicate" delete additional s

AR: Thank you.
AC: The word will be corrected in the manuscript.

L18: "the bias in Tens is explained by" I think this statement is too strong. It can be an explanation, but I think it cannot be reduced to this single causality, as you state in the next sentence.

AR: Thank you.
AC: We suggest to replace "is" with "is partly explained by"

L21: "The Tens CRPSS is skilful" forecasts have a positive CRPSS and are skilful. The current formulation is not logical, a CRPSS is not skilful.

AR: We will rephrase to clarify that skillful refers to the forecast.
AC: We will change to: "…, CRPSS show that Tens is skillful for both…"

L28: please state these characteristics very shortly again here.

AR: We will modify as suggested.
AC: We will change the text as follows: " … Only a few significant relationships between catchment charactersitcs, e.g. catchment area and elevation gradient, were found"

P13

L1: I don't understand what you mean with "the averaging effect on temperature skill dominates".

If I understand correctly, you could discuss here what the difference would be if you use a spatially distributed hydrological model (e.g. gridded version of the model with high resolution). The effect of temperature downscaling might be higher in this case because you do not average temperature again

after the downscaling and the spatial distribution within a catchment would have a much larger effect especially in catchments with high spatial variability of soil properties, altitude and vegetation cover.

AR: What discuss in this paragraph is the effect of catchment size on the performance of the forecasts. We think that a forecast for small catchments are more sensitive than large catchments to the spatial pattern of forecasted temperature. The reasons are that (i) the smallest catchment are smaller than the grid size of the ECMWF model and (ii) it is more challenging to forecast weather on small spatial scales than large spatial scales.
AC: We will replace the last sentence in the discussion: This result is not conclusive, but indicates that (i) the smallest catchment are smaller than the grid size of the ECMWF model and therefore very sensitive to the pre-processing (ii) it is more challenging to forecast weather on small spatial scales than large spatial scales.

L13: "the calibrated temperature reduced the skill of the forecasted streamflow." Please state what skill measure you mean here, did you calculate the CRPSS or bias for that specific event? In the result you only describe the range of the calibrated / uncalibrated ensembles but not a measure of skill.

AR: You are right. In this sentence, the use of skill is misleading. We did not calculate a specific measure of skill, but merely point to fact that compared to the reference streamflow, the calibrated T forecast induce too high streamflow, and the error becomes larger. A better word would might be performance.
AC: We will changed "skill" to "performance"

L15-17: I think you would like to point out that other errors (in the meteorological dataset and the hydrological model) do influence the results. If so, the sentence should be rephrased. Now the reader might think that forecasts are always getting worse if they are calibrated and this would be an argument against your conclusive statement in the summary on Page 14/L19-18.

AR: We agree, and will add a sentence to clarify this. We will also remove streamflow observations from the figure and consequently from the discussion.
AC: We suggest rewriting: "Firstly, deterioration in the forecast performance using calibrated temperature is particular for this event. Other results provided in this study shows clearly that the calibrated temperature ensembles improve the streamflow ensemble forecasts on average. This discrepancy reveals the other sources of errors; such as the uncertainty of the observed SeNorge precipitation and temperature, and the ability of the hydrological model to capture the highest flood peaks. These points are outside the scope of this study and will not be followed up further here, but are of course important for the performance of a flood forecasting system."

Figures:

Figure 1: write "grouped" instead of "divided". Something is wrong in the first sentence "this study shown using". Please rephrase.

AR: We will rephrase the caption.
AC: The caption will be rephrased:  "The maps for Norway indicates the 139 catchments used in this study. The left map shows the catchment boundaries including the location of four selected catchments.

The right map presents the location of the gauging stations grouped into five regions (N=north, M=mid, W=west, S=south, and E=east), and marked with colors and region boundaries."

Figure 3:

Avoid overlap of the boxplots to enhance the readability of the plot. There seem to be two line-artefacts on both sides of the figure.

AR: We will have a look at the box-plots, the artefacts in the figures will probably disappear in the finishing stage, as all figures will be provided separately. We used partly overlapping boxes for each lead time to increase the readability of the figure, since it is easy to see to which boxes that belongs to the same lead time. We tried without, but found it then more difficult to read the plot.
AC: No changes will be introduced.

Figure4:

In the text you write TO and in the Figure it corresponds to Tobs. Similarly, Tens and Tens-range. It might facilitate the text of the abbrevations are more consistent in the text, captions and the figures.

AR: Thank you. This will be corrected a suggested
AC: We changed the figures, and correct "$T_{obs}$ to $T_o$" in both plots.

Figure 6:

Line artefacts on the left of the figure.

AR: The artefacts in the figures will probably disappear in the finishing stage; all figures will be provided separately.
AC: We will check that the line artefacts are not present in the final manuscript.

Figure 10:

It is hard to see the actual forecast. I suggest removing the background colors for the warning levels and just plot lines instead. The Figure can easily be misinterpreted as the individual plots (e.g. upper panel for lead day 2) look like a continuous forecast. Maybe it would be more suitable to plot boxplots instead.

AR: OK. We will do some changes to this figure. Ref A#3. We prefer- however, to not use box-plots. We think that the use of lines and shaded areas increase the readability of the figures.
AC: We have changed the figure. The background removed and the streamflow observation are removed.

Captions: Forecast issue date is the date when the forecast was issued, hence the x-axis could be different for each panel in this figure. I recommend adapting the caption to make this clearer, e.g. target day instead of issue date.

AR: .Thank you. We will follow the suggestion.
AC: We will change to "target day" in the caption, and in the text p10-l28

"model streamflow with SeNorge observations" this is QO. I would write it in brackets as you do for Qcal.

AR: Thank you, we will follow the suggestion.
AC: OK. Add both "Obs" and "$Q_o$" in the text.

References:

Engdahl, B. J. K and Homleid, M: Verification of Experimental and Operational Weather Prediction Models December 2014 to February 2015. Norwegian Meteorological Institute, METinfo (18/2015), 2015

References:

Jolliffe, I. T., & Stephenson, D. B. (2011). Forecast Verification. (I. T. Jolliffe & D. B. Stephenson, Eds.), Forecast Verification. Chichester, UK: John Wiley & Sons, Ltd. https://doi.org/10.1002/9781119960003

Li, W., Duan, Q., Miao, C., Ye, A., Gong, W., & Di, Z. (2017). A review on statistical postprocessing methods for hydrometeorological ensemble forecasting. Wiley Interdisciplinary Reviews: Water, 4(December), e1246. https://doi.org/10.1002/wat2.1246

Wilks, D. S. (2011). Statistical Methods in the Atmospheric Sciences. (D. S. Wilks, Ed.) (3rd ed.). London: International Geophysics Series, Vol. 100, Academic Press Inc.

Yuan, X., Wood, E. F., & Ma, Z. (2015). A review on climate-model-based seasonal hydrologic forecasting: physical understanding and system development. Wiley Interdisciplinary Reviews: Water, 2(5), 523–536. https://doi.org/10.1002/wat2.1088

---

## Author Response (AR1)

[revised manuscript text omitted]
 colorsgrey dotted lines indicate the warning level green, yellow, and orangethresholds for mean annual, 5-year and 50-year floods.

---

## Author Response (AR2)

**Author Response to Editor**

Dear Dr. Jan Seibert.

Thank you for your comments to our author responses and corrections.
To the Editor comments (grey text) in this first section, we provide author corrections (AC, blue text). The author corrections show the page and line to which the implemented corrections and changes are found in the provided track changes version of the revised manuscript. Hence, **P6L3-5** indicates changes on page 6 and lines 3 to 5. This document further includes all author response (AR) and corrections (AC) made to the referees and the revised manuscript with tracked changes.

We first address the editor comments thereafter the author response to each referee:

Looking through your responses in the discussion phase, I have the following minor comments:

**1: Reviewer 1: Your motivation for using a daily time step could be included in the text**

AC: **P7L13-15** We have added "All temperature forecasts were aggregated to daily time steps since the operational HBV model runs on a daily time step, and the SeNorge data used as a reference provides only daily values."

**2: Reviewer 3: Please reconsider addressing the first comment. I agree with the reviewer that discussing seasons where temperature obviously has much less of an effect could be distracting.**

AC: We omitted the reference to summer and winter. We further removed these seasons from figure 5 and updated the figure caption accordingly.
Moved from section 4.2 to 3.2 **P10L15-19**: "For this paper, we chose to focus on the results for autumn and spring. Summer (July to September) was excluded due to the relatively small changes in CRPSS explained by (i) the skill of uncalibrated temperature forecasts are higher and the potential for improvement is lower, and (ii) there is less or no snow in summer, resulting in a reduced streamflow sensitivity to temperature. Winter (October to December) was excluded since it performs similarly as the autumn season."
Removing references to summer and winter in the following place: **P10L13-14, P11L21-24, P13L10-11, P17L1+3+8**

**3: Reviewer 4: CRPS and the subscript version, I agree with the reviewer's concern and would suggest to more clearly state that these refer to the same thing"**

AR: Firstly, we would like to emphasize that there are many recent examples in HESS where CRPSS is used as a multi-letter variable in equations: Sadri et al. (2018), Woldemeskel et al (2018), Lucatero et al (2018), Rogelis et al. (2018), Bazile et al., (2017). Since CRPSS is a well established abbreviation, we want to keep it in the text. In order to follow the HESS standard for notation in equations, and use single letter variables we did the following updates in the revised manuscript.

AC: **P10L5-7** We provided a sentence to clarify sec 3.2: "For readability, the abbreviation Scrp and Scrps used in the equation will be substituted with CRPS and CRPSS in the text hereafter"

**P9L11+20+21:** We added explanations similar to "CRPS denoted as $S_{CRP}$ in Eq. 1"

**Author Response to Editor**

Sadri, S., Wood, E. F., and Pan, M.: Developing a drought-monitoring index for the contiguous US using SMAP, Hydrol. Earth Syst. Sci., 22, 6611-6626, https://doi.org/10.5194/hess-22-6611-2018, 2018.
https://www.hydrol-earth-syst-sci.net/22/6611/2018/hess-22-6611-2018.pdf

Woldemeskel, F., McInerney, D., Lerat, J., Thyer, M., Kavetski, D., Shin, D., Tuteja, N., and Kuczera, G.: Evaluating post-processing approaches for monthly and seasonal streamflow forecasts, Hydrol. Earth Syst. Sci., 22, 6257-6278, https://doi.org/10.5194/hess-22-6257-2018, 2018.
https://www.hydrol-earth-syst-sci.net/22/6257/2018/hess-22-6257-2018.pdf

Lucatero, D., Madsen, H., Refsgaard, J. C., Kidmose, J., and Jensen, K. H.: On the skill of raw and post-processed ensemble seasonal meteorological forecasts in Denmark, Hydrol. Earth Syst. Sci., 22, 6591-6609, https://doi.org/10.5194/hess-22-6591-2018, 2018.
https://www.hydrol-earth-syst-sci.net/22/6591/2018/hess-22-6591-2018.pdf

Bazile, R., Boucher, M.-A., Perreault, L., and Leconte, R.: Verification of ECMWF System 4 for seasonal hydrological forecasting in a northern climate, Hydrol. Earth Syst. Sci., 21, 5747-5762, https://doi.org/10.5194/hess-21-5747-2017, 2017.
https://www.hydrol-earth-syst-sci.net/21/5747/2017/hess-21-5747-2017.pdf

Rogelis, M. C. and Werner, M.: Streamflow forecasts from WRF precipitation for flood early warning in mountain tropical areas, Hydrol. Earth Syst. Sci., 22, 853-870, https://doi.org/10.5194/hess-22-853-2018, 2018.
https://www.hydrol-earth-syst-sci.net/22/853/2018/hess-22-853-2018.pdf

**Author Response to RC#1**

Thank you for a very positive feedback on our article. We appreciate the valuable comments that are helpful in order to improve the manuscript.

We would like to apology for the missing references. The error emerged when we specified the HESS format, and un-intentionally deleted many references from the reference list. The main author should nonetheless have detected this flaw prior to posting.

Replies and corrections are done as follows: the Author response (AR) is marked with red text, while the author's suggestions to corrections (AC) are marked with blue text. All Referee comments are kept in a black; we use page and line number when needed to specify the appropriate location. All page and line references from the author are to the provided track changes version of the revised manuscript. Hence, **P6L3-5** indicates changes on page 6 and lines 3 to 5.
This is a well written paper. It investigates the impact of temperature forecasts on streamflow forecast skill, especially considering the effect of pre-processing of temperature ensemble forecasts. The study is based on forecasts for a large number of catchments in Norway, thus providing a very comprehensive and systematic analysis. The paper provides an important contribution to the research and practical application of ensemble meteorological forecasts for streamflow forecasting.

Detailed comments:
1. Page 2, line 16-17. There are different ways of producing meteorological ensemble forecasts. Typically, also model physics are perturbed.
AR: You are right. The ECMWF ensemble prediction system includes stochastic perturbation to the model physics. We will add to the sentence to address this aspect.
AC: **P2L22** We changed to "… are created by perturbing both the initial states of the original deterministic forecast and the physics tendencies of the …."

2. Page 5, line 18-19. Not clear here how catchment average precipitation and temperature are estimated. Are they based on the SeNorge data sets? If so, is it then necessary to apply elevation corrections for the model calibration, since elevation corrections have been applied for producing the SE Norge data sets?
AR: We agree that the description of how temperature is used in the hydrological model is ambiguous and this will be clarified in the text. You are right that elevation correction is applied to the SeNorge dataset. Our set-up for the HBV models uses catchment average temperature as input, calculated from the SeNorge data. The elevation correction mentioned in l18-19 refers to the internal correction in the HBV model. These are used to adjust catchment average temperature and precipitation, representing the catchment mean elevation, to each elevation zone in the HBV model. A linear elevation adjustment is applied to temperature, whereas an exponential adjustment is applied to the precipitation.
AC: **P6L6-9** We reformulated
"The model uses catchment average temperature and precipitation as input. Each catchment is divided into 10 elevation zones, each covering 10% of the total catchment area. The catchment average precipitation and temperature are elevation adjusted to each elevation zone using catchment specific laps rates.

**Author Response to RC#1**

3. Page 6, line 17-20. Why use a daily time step for the streamflow forecasts? Meteorological forecasts with a 6-hour time step are available.

AR: The operational HBV model used for flood forecasting runs on a daily time step. In addition, the SeNorge data that is used for model calibration and updating, provides only daily values.

AC: **P7L13-15** We have added "All temperature forecasts were aggregated to daily time steps since the operational HBV model runs on a daily time step, and the SeNorge data used as a reference provides only daily values."

4. Page 7, line 4-6. For the quantile mapping, a critical issue is the mapping of forecasts outside the range of observed data. How is this done?

AR: MetNorway use parametric quantile mapping based on the hourly first 24h. When a forecast is outside the observation range, a 1:1 extrapolation is used. Therefore, if a forecast is 2°C higher than the highest percentile of forecasts used for calibration, then the calibrated forecast is 2°C higher than the same percentile for the reference.

AC: **P8L13-15** Rewritten "The same coefficients, based on mapping the first 24 hours, are applied to all lead times and ensemble members individually. For forecasts outside the observation range, a 1:1 extrapolation is used. I.e. if a forecast is 2°C higher than the highest mapped forecasted temperature, then the calibrated forecast is 2°C higher than the highest mapped reference temperature. "

5. Page 8, line 12-13. Alternatively, you could use persistent forecast as benchmark. This would be more appropriate for evaluating short-term forecast skill.

AR: A persistent forecast will have some predictive skill in the short-range, but less for longer lead times. Engeland and Steinsland (2014; Fig. 4) show that the persistence did not add value after two days for selected Norwegian catchments. Pappenberger et al (2015) suggest using persistence as benchmark, based on a study of catchments larger than 6000km$^2$. However, given our selection of catchments, which are relatively small, quick responding, and with rapid changes in weather, combined with an aim to evaluate at longer lead times, we choice not to use persistence as benchmark. Rather, we used climatology as a benchmark since: (1) it is straightforward to get climatology as an ensemble, and (2) the focus of study is a lead time of five days. The daily climatology represented as daily ensemble (not an average value) gives a good representation of seasonal variations. Moreover, for this lead time persistent forecast has small predictive power due to the relatively short memory of our catchments (e.g. the streamflow autocorrelation for a time lag 5 days is less than 0.6 for about 80% of our catchments for the 25% highest flows).

AC: We made no modifications.

6. Page 12, section 5.3. There are a lot of repetitions in this section. I suggest including discussion on spatial patterns in sections 5.1 and 5.2.

AR: We will carefully read and revise Section 5.3 to avoid repetitions, and consider rewriting 5.1 and 5.2, to include the discussion from 5.3.

AC: **P13L9-P15L15** We have revised and rewritten section 5.1 and 5.2 to include spatial patterns previously presented in section 5.3. The original section 5.3 (Spatial patterns) is in total removed. For both 5.1 and 5.2 the Section headings are updated.

Technical corrections:
1. Page 2, line 30. Evensen (2003) not in reference list.

AR: Thanks; will be added

AC: The reference is added to the reference list: "Evensen, G.: The Ensemble Kalman Filter: theoretical formulation and practical implementation. Ocean Dynamics, 53(4), p343-367, 2003."

**Author Response to RC#1**

2. Page 4, line 27. "og" -> "and"
AR: This will be corrected.
AC: **P5L11** Changed

3. Page 11, line 20 and 24. Delete "Ivar".
AR: OK
AC: **P14L2+L6** Ivar is deleted in the citation

**References**

Engeland, K. and Steinsland, I.: Probabilistic postprocessing models for flow forecasts for a system of catchments and several lead times. Water Resources Research 50(1), p182-197, doi:10.1002/2012WR012757, 2014.

Pappenberg, F., et al.: How do I know if my forecasts are better? Using benchmarks in hydrological ensemble prediction. Journal of Hydrology, 522, 697-713. 2015

**Author Response to RC#2**

Thank you for the positive evaluation of our article. We appreciate the feedback that will contribute to improving the manuscript.

We would like to apology for the missing references. The error emerged when we specified the HESS format, and un-intentionally deleted many references from the reference list. The main author should nonetheless have detected this flaw prior to posting.

Replies and corrections are done as follows: the Author response (AR) is marked with red text, while the author's suggestions to corrections (AC) are marked with blue text.  All Referee comments are kept in a black; we use page and line number when needed to specify the appropriate location. All page and line references from the author are to the provided track changes version of the revised manuscript. Hence, **P6L3-5** indicates changes on page 6 and lines 3 to 5.

**Interactive comment on "Streamflow forecast sensitivity to air temperature forecast calibration for 139 Norwegian catchments"**
By Trine J. Hegdahl et al.

**Anonymous Referee #2**

This manuscript presents analyses of the sensitivity of streamflow forecasts to air temperature forecast calibration. The manuscript is well written, well structured, and I only have a few minor comments to the presentation, most of them just edits.

I find the description of validation scores and evaluation scores in 3.2 somewhat short. The section could give a better description of the rank histograms, and what is actually meant by the different shapes. And what is meant by slope and convexity being "negatively oriented"? Something seems wrong with the last sentence.

AR: Thank you for bringing this to our attention. We will provide some more details in the description of the rank histograms. By "negatively oriented", we mean that lower values (of slope and convexity) are better (i.e. more reliable forecasts). We will revise the sentence to better explain the meaning of "negatively oriented", and rephrase the last sentence to make it clear.
AC: We applied the following changes:
>    **P8L28-31** "For reliable ensemble forecasts, the rank-histogram will be uniform (horizontal). A bias in the ensemble forecast is recognized as a slope in the rank-histogram, where a negative slope indicates over-estimation by the forecasts (and vice versa). A U-shape indicates that the ensemble forecast is under-dispersed whereas a convex shape indicates over-dispersion (Hamill, 2001).
>    **P9L2-3** "… and convexity are negatively oriented, i.e. lower values are better, and with an optimum value of zero …."

P2L5 three main componentS?
AR: Thank you, will be corrected.
AC: **P2L8**: "component" replaced by "components"

P2L14 Langsrud et al, 1998a and 1998b are missing from reference list. What kind of statistical uncertainty models? (One line, to understand better what is different from the ensemble forecast)
AR: The references will be added and the text revised explaining the uncertainty model referred to.

**Author Response to RC#2**

AC: **P2L17-19** Rewritten "the uncertainty model accounts for the strong autocorrelation in forecast errors and estimates an uncertainty band around the deterministic temperature, precipitation and streamflow forecasts."

P4 2.1 Is Gjuvaa in the region South or East? Bulken is in the West region?
AR: We agree that the current manuscript is somewhat unclear on this issue and in the revised manuscript, we suggest adding in parenthesis to which region each catchment belongs.
AC: **P5L11+13** "Bulken (W), Gjuvaa (E) … " and for all later references to these catchments
**P5L14** We changed the following sentence: "Gjuvaa (E) is non-glaciered and located inland."

P5L20 PEST can be generic tools for parameter estimation or a particular software, what is it here?
AR: We use the PEST software to estimate parameters. We will specify this in the revised manuscript.
AC: **P6L9-10** Rewritten "… we used the operational model setup which has been calibrated using the PEST software to establish model parameters (Doherty, 2015)"

P6 2.2.4 / 3.1.1 Is the forecast from ECMWF point forecast (centre of the grid cell) or averages for the entire grid cell?
AR: The ECMWF forecasts should be considered as average values within the grid box, see Owens (2018, fig 3.2.1) for details.
AC: **P7L23-24** We added to the ECMWF description: "The ECMWF grid temperature, which represents the average temperature for the grid cell, was interpolated from a horizontal resolution of 0.25 (~30 km) …."

P7L23-24 "In this study, the ensemble range (…) visually assessed the sharpness." Something seems wrong, rephrase.
AR: Thank you. We will rephrase this sentence. We consider modifying this paragraph according to suggestions by RC#3, evaluate plot of empirical sharpness distribution.
AC: **P9L5-8** Rewritten "In this study, the temperature sharpness was assessed by first estimating the range between the 5$^{th}$ and the 95$^{th}$ percentile of the ordered ensemble forecasts for all issue dates, lead times and catchments. For streamflow, we estimated a relative sharpness by dividing the 5$^{th}$ to 95$^{th}$ percentile range by the ensemble mean. Thereafter, sharpness was determined for each catchment and lead time as the average range of all issue dates. "

P9L12-13 since "reliability has improved and some sharpness is maintained". This could be better explained.
AR: We will modify this part including evaluating plots of the empirical distribution of sharpness, ref. RC#3, and information above.
AC: **P11L1-2** Rewritten "A lead time of 5 days was chosen since reliability (convexity and slope) has improved and some sharpness is maintained, i.e. too large ensemble spread will increase the reliability but the forecast value will be reduced."

P6L24 I guess it should be "atmospheric lapse rate"?
AR: You are quite right; will be corrected.
AC: **P7L27** "atmospheric**"**

P8L17 remove s from catchments.
P9L9 remove comma after convexity
P10L7 they performs – remove S.
AR: Thank you.

**Author Response to RC#2**

AC: **P9L27, P10L29, P11L26** We corrected as suggested.

P11L17 Rather than just sensitive, I think QM is unable to correctly map forecasts outside the observation range.
AR: We will rephrase to enhance the problems of QM mapping outside the observational range. It is important to note that all statistical methods will have problems outside the observational range. (ref RC#3, and discussion in RC#4)
AC: **P13L16** Rewritten "Quantile mapping, as most statistical calibration methods, is sensitive to forecasts outside the range of calibration values and period (Lafon et al. 2013), this can be an explanation for too high correction in the highest $T_{ens}$ quantile."
In addition, we added a sentence **P8L13-15** to clarify the use of quantile mapping: "The same coefficients based on mapping the first 24 hours, were applied to all lead times and members.  For forecasts outside the observation range, a 1:1 extrapolation was used.  I.e. if a forecast is 2°C higher than the highest mapped forecasted temperature, then the calibrated forecast is 2°C higher than the highest mapped reference temperature."

P12L2 temperatureS are?
AR: Thank you; will be corrected.
AC: **P14L18** Changed

P14 L29 "elevation correction dependency on lapse rate" – is this correct?
AR: We will rewrite to make this phrasing clearer
AC: **P17L29** Rewritten "… an elevation correction depending on lapse rate"

P16L17 No publisher?
AR: Thank you; will be corrected.
AC: Added to the reference list. "Engeland, K., Renard, B., Steinsland, I., and Kolberg, S.: Evaluation of statistical models for forecast errors from the HBV model. Journal of Hydrology, 384(1), 142-155, 2010."

Fig1 caption: Most of the catchments on the left are too small to be visible?
AR: We agree. The western catchments are small and thus difficult to distinguish on the map. We will revise the figure accordingly and further suggest adding a note on the fact that catchments on the western coast are small in the figure legend.
AC: **P24L5-10** Rewritten Fig 1 caption:" The maps for Norway indicates the 139 catchments used in this study. The left map show the catchment boundaries including the location of four selected catchments. Please note that many catchments are relatively small and difficult to detect. The location of the gauging station for all catchments are shown in the right map.  Norway was grouped into five regions (N=North, M=Mid, W=West, S=South, and E=East), all regions are marked with different colors and regional boundaries."

Reference
Owens, R G, Hewson, T D: ECMWF Forecast User Guide. Reading: ECMWF. doi: 10.21957/m1cs7h, 2018.

**Author Response to RC#3**

Thank you for the positive and good evaluation of our article. We appreciate the comments that are valuable and helpful in order to improve the manuscript.

We would like to apology for the missing references. The error emerged when we specified the HESS format, and un-intentionally deleted many references from the reference list. The main author should nonetheless have detected this flaw prior to posting.

Replies and corrections are done as follows: the Author responses (AR) are marked with red text, while the author's suggestions to corrections (AC) are marked with blue text. . All Referee comments are kept in a black; we use page and line number when needed to specify the appropriate location. All page and line references from the author are to the provided track changes version of the revised manuscript. Hence, **P6L3-5** indicates changes on page 6 and lines 3 to 5.

**Review of 'Streamflow forecast sensitivity to air temperature forecast calibration for 139 Norwegian catchments' by Trine Hegdahl et al.**
Jan Verkade, November 2018

**Overall impression**
This manuscript is suitable for publication. The research described in it has a clear objective which is to try and determine if 'calibrated temperature ensemble forecasts' result in better streamflow forecasts compared to the non-calibrated equivalents. The research setting, the approach and the data used is well described and the results are well laid out. I have a few concerns/questions/suggestions but these would require only minor revisions to the manuscript.

**Minor comments**
**Overall**
• Multiple references are made to seasons in which the effect of temperature forecast calibration on streamflow was negligent. You're right to point out that the reason is that temperature forecasts only matter if/when it affects the simulation of snowmelt processes. You could consider mentioning this in the start of the paper, explain that for this reason, you're looking at only those seasons where temperature affects streamflow through either rain-falling-as-snow or through snowmelt, and then omit reference to the other seasons altogether. I find it a bit distracting from the main points.
AR: This is a good suggestion. We found it useful to include all seasons in the first part of our analyses in order to highlight the differences between seasons, which subsequently provide the motivation for leaving some seasons out of the final analysis.
AC: We omitted the reference to summer and winter. We further removed these seasons from figure 5 and updated the figure caption accordingly.
Moved from section 4.2 to 3.2 **P10L14-19**: "For this paper, we chose to focus on the results for autumn and spring. Summer (July to September) was excluded due to the relatively small changes in CRPSS explained by (i) the skill of uncalibrated temperature forecasts are higher and the potential for improvement is lower, and (ii) there is less or no snow in summer, resulting in a reduced streamflow sensitivity to temperature. Winter (October to December) was excluded since it performs similarly as the autumn season."
Removing references to summer and winter in the following place: **P10L13-14, P11L20-24, P13L10-11, P17L1+3+8**

• For many hydrologists, the word 'calibration' has a different meaning from how it is used in your paper.

**Author Response to RC#3**

I acknowledge that your meaning is consistent with how many meteorologists would interpret it. I would recommend to address this issue by either use a different word (I believe HESSD readers may be more familiar with 'post-processing') or by addressing this in the text somewhere.

AR: We agree that hydrologists might interpret the term "calibration" to "hydrological model calibration", and we will clarify our use of the terminology as illustrated in Figure 2. Post-processing is, in our paper, a general term for any modifications applied to a raw meteorological forecast. We distinguish between calibration and downscaling, that both are post-processing methods. This is consistent with the terminology used by the Norwegian Meteorological Institute (MetNorway) (https://github/metno/gridpp).

AC: **P2L7-11** We rewrote to clarify the terminology used "Post-processing refers to all techniques used to change the output from a meteorological model, and includes calibration (described above) and downscaling. Downscaling implies resampling from the original forecast grid size to a grid of higher resolution, and both statistical (e.g. interpolation) and dynamical (e.g. a regional weather forecast model) techniques, can be used (Schaake et. al., 2010). A recent review of post-processing methods are given in Li et al (2017) and the textbook edited by Vannitsem et al (2018)"

• Citations aren't always properly formatted. I think I've seen ((double parentheses)), for example. In S3.1.2, l12, a correct way to refer to the evidence would be (Seierstad, 2017) with the 'personal communication' listed in the bibliography. I think. I've also seen citation in which both first and family names are listed. May be good to verify against Copernicus citation rules.

AR: Thank you.

AC: The citations and references have been formatted according to the HESS standard.

**Abstract**

• l9-11 These sentences distract from the point you're going to make. While the facts you state may have a place in the introduction, I would omit these from the abstract.

AR: You are right. We will consider rewriting the abstract.

AC: **P1L9-14** We changed the first sentences as follows:

"In this study, we used meteorological ensemble forecasts with the hydrological models to quantify the uncertainty in forecasted streamflow, with a particular focus on the impact of ensemble temperature forecasts. In catchments with seasonal snow cover, snowmelt is an important flood generating process."

• l20 'the HBV model is used to *calculate* streamflow'. The verb *to calculate* presumes certainty. Pls consider using *estimate* instead.

AR: Thank you, we will change as suggested, i.e. using 'estimate' both in the abstract and in the text.

AC: We changed as suggested **P1L23, P2L20, P7L8+11+16, P10L25, P14L21.**

• l21 'influenced'. My understanding is that 'influences' (and the associated verb) are a thing of the mind ("Who are your main influences?" "Joan Baez"). For physical processes, I think 'affected' is more suitable.

AR: Thank you. We will change 'influence' used as a verb to affect, and to 'effect' where 'influence' is used as a noun.

AC: Changed to affects or in some cases effect: **P1L24+27(affected), P2L8(affected), P10L20 (effect), P11L19 (affects) P16L8 (affects) + L25 (effect), P17L7 (effect)**

• l26 'however'. I don't think this sentence contradicts anything that was stated before. Hence, the word 'however' may be omitted.

AR: Thank you, we will omit "however".

**Author Response to RC#3**

AC: **P1L31** Rephrased to "Overall, it is evident that temperature forecasts are important for streamflow forecasts in climates with seasonal snow cover."

**Section 3.1.2**

• I am not entirely sure who provides the calibration parameters. L5 suggests MetN, but the sentence "To establish the calibration parameters. . . " (l8) may be interpreted as an explanation of how the authors have done this.

AR: MetNorway did the quantile mapping, and established the calibration parameters. The calibration parameters were originally used to bias correct the temperature forecasts as provided on yr.no (the Norwegian weather forecasting). We applied the Met-parameters to the raw ENS temperature forecasts of our selected period.

AC: **P8L8** We rephrased the sentence "To establish the calibration parameters MET Norway used both ENS re-forecast (Owens, 2018) and Hirlam data from July 2006 to December 2011 interpolated to a 5×5 km$^2$ grid."

In the Met Norway procedure, why aren't temperature *observations* used? Are the HIRLAM reanalyses deemed to be sufficiently certain? This may deserve a few informed comments.

AR: You are right to point out these differences in data sets used for calibration of forecasts and the hydrological model. First, as you mention, SeNorge and Hirlam are not the same data. Hirlam is a short-range regional forecast model (4 km resolution) used in the operational weather forecast for the first 2 days, whereas SeNorge is a dataset where observations are interpolated to a 1 km grid.

In this study, we wanted to use the available operational method from MET Norway, and they use quantile mapping with Hirlam as a reference to calibrate the ECMWF ensemble forecast. Both Hirlam (for the first 2-3 days) and ECMWF (for the following 7-8 days) forecasts are used in the operational weather forecast (yr.no). Using Hirlam data to calibrate ECMWF will improve the transition between the forecasts. Hirlam is available as a sub daily grid and makes it possible for MET Norway to provide different calibration parameters for day and night, whereas SeNorge is only available as a daily grid and would not offer this possibility.

Hirlam have less errors than ECMWF in the temperature forecast for Norway (Engdahl et al. 2015), and as we see from e.g. fig 6 and 7 that the calibration improves especially the cold biases in the ECMWF forecasts. When we evaluated the hydrological model, the temperature calibration improved, in most cases, the hydrological forecasts, providing an indirect confirmation that the HIRLAM temperature is less biased than the ECMWF temperature. Nevertheless, the results suggest that there might be improvements using the SeNorge data instead of Hirlam, but this needs to be tested (beyond the scope of this study).

AC: **P8L5-8** Rewritten "MET Norway uses Hirlam (Bengtsson et al., 2017) temperature forecast (on a 4×4 km$^2$) to provide a reference for the parameter estimation (calibration). Hirlam is suitable as a reference since it provides a continuous field covering all of Norway at a sub daily time step. In addition, Hirlam gives a higher skill and are less biased than the ENS (Engdahl et al., 2015).

• If I am correct in understanding that both the raw and the calibrated ensembles have been provided by Met Norway then maybe this should be stated more clearly. Or is it the case that Met Norway computed the calibration parameters on a data set from 2006-2011 and that you applied these yourself to a data set ranging from March 2013 through Dec 2015? If so, maybe state this more bluntly?

AR: Your second suggestion is correct. The raw ensembles from ECMWF (March 2013-Dec 2015) and the calibration parameters (based on data ranging from 2006-2011) were supplied by MET Norway, whereas

**Author Response to RC#3**

we did the calibration using the provided calibration parameters and available computer scripts (github/metno/gridpp).
AC: We separate what MET Norway did from what we did. The first paragraph of section 3.1.2 contains the description of calibration parameters from MET Norway **P8L2-15**, whereas the second paragraph **P8L16-21** what we did:
(1) **P8L8-9** added to the first paragraph: "To establish the calibration parameters MET Norway used both ENS re-forecast (Owens, 2018) and Hirlam data from July 2006 to December 2011, both interpolated to a 5×5 km² grid… "
(2) **P8L16-17** added to the second paragraph: "In this study, we applied the calibration coefficients provided by MET Norway to the temperature forecasts for the period 2013-2015. Accordingly, the ENS was interpolated to the 5×5 km² …."

• I am assuming that you used a HIRLAM reanalysis. Is that correct? If not, what lead times are you using and do the HIRLAM forecasts you used have the same max lead time as the ECMWF ensembles? I am only familiar with a few instances of HIRLAM and these all go out to just over 2 days max.
AR: MetNorway used the operational Hirlam forecasts for the calibration period. It is correct that Hirlam does not cover the same lead times as ENS. Met Norway established the calibration parameters using the 24 first hours of the forecasts as the reference.
AC: **P8L13-15** We added a sentence to clarify this "The same coefficients, based on the first 24h mapped, are applied to all lead times and ensemble members individually.  For forecasts outside the observation range, a 1:1 extrapolation is used.  That is, if a forecast is 2°C higher than the highest mapped percentile, then the calibrated forecast is 2°C higher than the same percentile for the reference. "

• By off-setting Tens against Tcal, you create the impression that Tcal is not an ensemble forecast. Consider using Traw and Tcal instead.
AR: We chose to use "ens" instead of "raw", since an elevation-correction was applied the forecasts, and hence they are not actually "raw".
AC:  We added to existing text to underline that $T_{cal}$ (and $Q_{cal}$) is an ensemble.  **P11L4-5+22-23, P12L29-30, P13L9-10, P14L8**

• l29-30. The 'assessment' was done by you, not by the ensemble range.
• On assessing sharpness: how confident are you that a visual assessment does the job? Pls consider plotting the empirical distribution of sharpness of all your forecasts and comparing those.
AR: We will plot the empirical distribution of sharpness for all temperature ensembles, and rephrase the sentence concerning sharpness accordingly.
AC: **P9L5-8** Rewritten "In this study, the temperature sharpness was assessed by first estimating the range between the 5th and the 95th percentile of the ordered ensemble forecasts for all issue dates, lead times and catchments. For streamflow, we estimated a relative sharpness by dividing the 5th to 95th percentile range by the ensemble mean. Thereafter, sharpness was determined for each catchment and lead time as the average range of all issue dates."

• If you're calibrating the temp ensembles on a leadtime by leadtime basis and on a grid cell by grid cell basis, chances are that you'll change the temporal pattern (forecasted temperature as a function of time) as well as the spatial pattern. Does this in any way affect use in streamflow forecasting? I believe there are some techniques that may be helpful in trying to restore spatial-temporal relations (the Schaake shuffle springs to mind). Would these have a use in present study?

**Author Response to RC#3**

AR: We think that the calibration will not affect the spatial and temporal pattern significantly. The calibration function was applied to each ensemble member individually. We therefore kept the order of the ensemble members, both in space and time, and it was not necessary to use the Schaake shuffle.

AC: We think this will be clearer by adding the following description to quantile mapping page 7, line 12-13. (Response above): "… are applied to all lead times and ensemble members individually…"

**Section 3.2**

• Would it be fair to say that temperature forecasts are only relevant if they can discriminate between freezing and non-freezing situations? If so, would it be justified to focus more on this discrimination? Perhaps by defining an event (T<0, for example) for which one can compute a range of verification scores (false alarms, hits, ROC, Brier's probability score, etc). I acknowledge that this would be feasible for temperature and less obvious for streamflow.

AR: This is a good suggestion. Nonetheless, we think this is beyond the scope of this study. This could be an interesting topic for a future study.

AC: No change

**Section 4**

• " To reduce the amount of presented results, the remaining part of this paper focuses on CRPSS for a lead time of 5 days." This is fine, but temperature forecast at 5-day lead time may not affect streamflow forecasts until a (much) longer lead time. Or conversely, streamflow forecasts at day 5 would have been affected by a day 2 temperature forecast (this is an example). As in some cases you're comparing Q-forecasts with T-forecasts, how have you accounted for this?

AR: This is an interesting question. The streamflow forecast at day 5 will be affected by the temperature forecast the previous 4 days as well as day 5. However, for most catchments in this study, the concentration time is less than one day, and the streamflow will respond the same day as a major water input from rain or snow melt. For specific events, it is not evident which of the T-forecasts at day 1-5 is the most important for the Q-forecast at day 5. The sensitivity depends on the sequence of temperature and precipitation. Nevertheless, we think that using temperature CRPSS for day 5 is a good choice since the streamflow at day 5 is the most sensitive to the temperature at day 5 on average (which applies to all lead times). In addition, we see that the improvement in CRPSS across lead times is highly correlated and our results and conclusions would not change if we used temperature CRPSS for days 2, 3, or 4 instead.

AC: **P14L26-28** Added "The same lead time was used to relate improvement in streamflow to temperature, we consider this robust since most catchments in this study have a concentration time of less than a day."

**Section 4.1**

• In the text, you refer to observed temp as To. In plots, as Tobs. Pls make this consistent. I recommend using Tobs throughout.

AR: Thank you for highlighting the in-consistency in the use of $T_{obs}$ and $T_o$. Since the SeNorge temperature is an interpolated product of the observations, we therefore prefer to use $T_o$.

AC: **P30L1-5** Changed to $T_o$ in fig 4 and caption

• L23-25. These sentences are better placed in a discussion section, I think.

AR: OK.

AC: Sentences were deleted **P11L13-15** and rewritten for sec 5.3 **P16L1-4**

**Author Response to RC#3**

• L19 'influence' is missing an 's'. Pls consider replacing by 'affects' though.
AR: Thank you.
AC: **P11L19** We replaced "influence" with "affect".

**Section 4.2**

"Scatter plots of the difference between CRPSS for calibrated and uncalibrated forecasts". CRPSS in itself is a fairly abstract measure. The difference between two CRPSS scores is, I find, even more abstract. What's the meaning of those values? As CRPSS is a skill of a forecast versus a baseline, why not simply calculate the CRPSS of the calibrated forecasts using the CRPS of the uncalibrated forecasts as a baseline?
AR: We wanted to evaluate the skill of the uncalibrated forecasts as well. If we were to use the uncalibrated as a benchmark, we would not assess the quality of the original forecast, only the change between the uncalibrated and calibrated forecast.
AC: No changes introduced.

**Section 5**

L7: 'dispersion' is not an expression of quality but a characteristic of an ensemble. Saying 'dispersion improved' makes little sense then?
AR: Thank you. What we mean is that dispersion, as measured by rank histogram convexity, improved.
AC: **P13L1-3** We changed to "Even though both bias and dispersion (i.e. reliability) as measured by rank histogram slope and convexity improved with longer lead time, the reduced sharpness and increased uncertainty, resulted in a reduced skill (CRPSS)."

**Section 5.1**
• L11 "skill. . . depends". Consider replacing by "skill. . . varies with".
AR: Thank you.
AC: **P13L9** We applied as suggested "The skill for both raw (uncalibrated) $T_{ens}$ and calibrated $T_{cal}$ temperature ensembles varies with season."

• "Quantile mapping is sensitive to forecasts outside the range of calibration values and period". I think it would be good to point out that this is true for any statistical post-processing procedure.
AR: Good point.
AC: **P13L16** "Quantile mapping, as most statistical techniques, is sensitive to forecasts outside the range of calibration values and period (Lafon et al. 2013), this may explain the too high correction in the highest $T_{ens}$ quantile. "

• Immediately following: "and can be a" –> "and *this* can be a"
AR: Noted
AC: **P13L17** Changed as suggested

• On the causes of temperature forecast bias. You go into some detail to explain a situation in which land is colder than sea. Would this be a typical situation for summer/winter? If so, can you more directly link this to some of the results you're showing?
AR: We will clarify that this is a typical situation of winter. This is to some point already exemplified in the text, and we can underline in the text that the situations are typical for winter. (5.3 will be included in 5.1 and 5.2, and we will ensure to get this information in the revised manuscript):

**Author Response to RC#3**

AC: **P13L20-28** Implemented and rewritten for the revised sec 5.1 "The most pronounced spatial pattern is the low autumn CRPSS for uncalibrated ensembles $T_{ens}$ in the coastal areas. This is seen from the boxplots for the regions West, Mid and North (Fig. 8) and in the plots of the western catchments Viksvatn and Foennerdalsvatn during winter months (Fig. 4). This cold bias is documented for the Norwegian coastal areas in the cold seasons by Seierstad et al (2016), and is mainly caused by the radiation calculations in the ECMWF model (Hogan et al., 2017). The coarse radiation grid results in warmer sea points being used to compute longwave fluxes applied over colder land points, causing too much cooling. This effect is seen for the temperature forecast for winter 2014 and 2015 for the coastal catchments in fig 4 (b) and (c), in contrast to the inland catchment (a) which is less biased.  The radiation resolution is improved in later model cycles (Hogan et al., 2017; Seierstad et al., 2016). In addition, the challenging steep coastal topography is not well represented by the spatial resolution in the ECMWF model (Seierstad et al., 2016). For inland catchments, and the regions "

**Section 5.2**
L10 Grammatically, this sentence is awkward if not wrong.
AR: Thank you; we will rephrase this sentence.
AC: **P14L29-P15L1** We rephrased "In summary, it can be concluded that to further improve streamflow forecasts during the snowmelt season, improved temperature forecasts are essential.  Streamflow forecasts during spring have the highest potential for improvements since the temperature forecasts were not, for a majority of the catchments, improved by the applied calibration."

**Figures**
**Overall**
Many figures use a lot of white space between various plots/panels. Consider reducing this or, even better, removing altogether.
AR: We will reduce some white space in figure 1 and 3.
AC: New figures provided P

**Figure 1**
• Do the grey polygons add up to 139 in total? If so, many must be **really** small?
• Caption: consider using 'boundaries' instead of 'limits'
AR: Yes. Especially on the western coast, the catchments are small. This will be clarified in the caption
AC: **P24L5-10** New caption text Fig 1: "Figure 1: The maps for Norway shows the 139 catchments used in this study. The left map show the catchment boundaries including the location of four selected catchments. Please note that many catchments are relatively small and difficult to detect. The location of the catchments' gauging stations are shown in the right map. Norway is grouped into five regions (N=north, M=mid, W=west, S=south, and E=east), and all regions are marked with different colors and regional boundaries."

**Figure 4**
• Why plot the ensemble **mean** and not all five ensemble members, possibly as horizontal lines?
AR: It is not evident to us which modification the reviewer suggests. In this plot, the mean is for the 51 ensemble members not five. If we were to plot all the members, it will be difficult to retain any information. By plotting the mean we show the bias in the forecast and by using the scatter plot, we also show that some biases are dependent on forecasted temperature (a conditional bias).
AC: No changes introduced in the plots.

• The axes of the plots in the right-hand column vary. Please consider unifying this. Also: please consider ensuring that horizontal and vertical axes are identical. Maybe they are, but the labeling isn't.
AR: We will unify the axes.
AC: **P30** The axis are unified in Fig 4.

**Author Response to RC#3**

**Figure 5**
• What lead time are these plots for?
AR: Thank you; we will add the lead time in the caption.
AC: **P32L5** Caption updated "All plots are presented for lead time 5 days."

• Is the lead time for T identical to that for Q? What is the 'response time' of the catchment to snowmelt?
If not zero then shouldn't this be taken into account somehow?
AR: We use the same lead time for temperature as for streamflow.  See comment to section 4.
AC: No changes applied.

Please consider. . .
• . . . removing data for seasons for which temperature has little or no effect on streamflow levels.
AR: We would like to keep the plots for all seasons here. By showing the difference between the seasons, we think it is easier to understand the large variations we see.
AC: No changes applied

• . . . unifying horizontal and vertical axes. it took me a little while longer than I cared to realise that the light grey slanted line is the 1:1 diagonal.
AR: We will consider changing the plots. However, unified axes means that we lose information about the regional distribution..
AC: **P32** We unified the axis in Fig 5, and omitted summer and winter.

**Figure 6**
• What do you want the reader to compare? CRPSS(T) and CRPSS(Q)? Or CRPSS(spring) v CRPSS(autumn)? Pls ensure panels are ordered accordingly.
AR: We wanted the reader, first of all, to compare CRPSS(T) and CRPSS(Q) Therefore, we placed CRPSS(T) and CRPSS(q) from the spring season on the first line and for the autumn season in the last line. Then the reader can evaluate how the improvements in temperature will affect improvement in streamflow, for both seasons. Secondary, we wanted to show the difference between seasons. Sub-plots for each season are therefore arranged vertically, for both temperature (left) and streamflow (right).
AC: No changes introduced.

• pls ensure that within a row, panels have identical vertical axes so this comparison can indeed be done (i.e. the reader can then easily compare the top left with the top right plot)
AR: We prefer to use different scales on the vertical axes within a row to increase the readability of each sub-plot. In particular, the plots of the CRPSS(Q) would be more difficult to read if we used the same scale as in the plots of CRPSS(T) in the left panel.
AC: No changes

**Figure 10**
• The background colours have an effect on the colouring of Qens and Qcal. Please consider removing the background shades. Maybe replace these by threshold lines only?
• Please consider removing the number of lines in the plot, for example by only showing a shaded area with no line at the edges thereof.
• What is the purpose of showing both the 'real' observations and the 'model streamflow with SeNorge observations'? Is this distinction made in the paper, and addressed?

**Author Response to RC#3**

• Consider reversing the order of the graphs. The 9d lead time graph was available before the 2d lead time graph?

• The horizontal axis labeling is not in English.

• As all horizontal axes are identical, pls consider removing white space between plots altogether and only label the axis of the bottom plot.

AR: We will change the plots as suggested. We understand that the introduction of real observations in this figure is confusing, and we will therefore remove the real observation from the figure and from the text.

AC: **P38** New Fig 10 and updated caption. Updated the text **P12L21-23** "The horizontal grey dotted lines represent mean annual flood, the 5-year and the 50-year floods (i.e. the operational flood warning levels) in this catchment."

• The warning levels aren't relevant, are they? On reflection: you're scoring the forecast ensembles using
CRPSS and rank histograms. This shows absence of preference for doing well for 'extremes', even though the work appears to be inspired by forecasting for floods. How is this consistent? Maybe omit references to 'floods' altogether?

AR: In Norway, we use the mean annual, the 5-year and the 50-year floods as exceedance thresholds to issue flood warnings. This figure connects the theoretical aspects to the operational implementation, and points to the importance of calibrated temperature for a flood warning system.

AC: We kept the reference to flood levels, but removed the warning colors all together.

New Reference:

Engdahl, B. J. K and Homleid, M: Verification of Experimental and Operational Weather Prediction Models December 2014 to February 2015. Norwegian Meteorological Institute, METinfo (18/2015), 2015

Author Response to RC#4

Thank you for the positive and thorough evaluation of our article. We appreciate the comments, which are valuable for us in order to improve the manuscript.

We would like to apology for the missing references. The error emerged when we specified the HESS format, and un-intentionally deleted many references from the reference list. The main author should nonetheless have detected this flaw prior to posting.

Replies and corrections are done as follows: the Author response (AR) is marked with red text, while the author's suggestions to corrections (AC) are marked with blue text; we use page and line number to specify the appropriate location, where this is needed. All Referee comments are kept in a black; we use page and line number when needed to specify the appropriate location. All page and line references from the author are to the provided track changes version of the revised manuscript. Hence, **P6L3-5** indicates changes on page 6 and lines 3 to 5.

**Review of 'Streamflow forecast sensitivity to air temperature forecast calibration for 139 Norwegian catchments' by Trine Hegdahl et al.**
**Anonym referee#4**

Supplement:

Especially in hydrometeorological predictions where methods from both the meteorological and the hydrological forecasting community are used, it is of major importance to carefully define the terminology and to coherently use throughout the manuscript.

The current form of the manuscript shows a lack of precise formulations (e.g.: calibration, pre-processing, skill) which should be revised to better communicate the content of the study. Some of the graphics should be enhanced to facilitate the readability and the caption are sometimes incomplete. In addition, more than 15 references mentioned in the text are missing in the reference list and should be added.

AR: We thank you for the feedback. We would like to apology for the missing references. It seems to have been an error when we reformatted EndNote, which evidently lead to many references being deleted from the reference list. The main author should nonetheless have detected this flaw prior to submitting the manuscript. We will carefully revise the text to avoid inaccuracies in formulations.

Furthermore, some additional references could be of interest within discussion to put the findings of the study into a broader picture. Many of the references especially concerning the meteorological forecast are user guides and or technical reports or personal communications, which is fine, but I would appreciate if some more peer-reviewed literature would be cited as there is a large body of existing literature concerning the verification of ECMWF temperature predictions.

AR: We agree that it is better to use peer-reviewed literature. We chose to use technical reports and personal communication only when necessary and we found no other alternatives. In particular, there are not much peer-reviewed papers on the verification of the ECMWF temperature forecasts for Norway available. Hence, we chose to implement what is available of technical documentation.

In general, the language could be clearer and more concise. To me it is not clear what the authors understand under the term pre-processing, at least in the beginning of the manuscript. E.g. in literature there is a distinction between dynamical and statistical downscaling (see e.g. Li et al. (2017), Yuan et al. (2015)) and statistical downscaling does include a bias-correction. In the present manuscript, the term downscaling does only refer to applying a laps rate correction and interpolation, what is not as downscaling is referred to in the literature.

However, I think it would be important for the reader to have a short general overview of what pre-processing is in the introduction. In particular the term calibration, in the present manuscript used as a synonym for bias correction should be introduced more carefully because the term calibration is used by statisticians but in the meteorological, climatological and hydrological communities, the term bias correction is more common.

AR: We acknowledge that the literature is not consistent in terminology, and particular the terminology differs between the forecasting and the climate projection communities.

In our paper, we chose a terminology that is consistent with a large part of the literature, and that facilitates to explain the approaches we used. We use pre (and post)-processing as a general term, which includes all techniques applied to the raw temperature forecasts in order to improve the temperature output from the atmospheric model (i.e. downscaling and calibration are pre-processing techniques). We pre-processed the temperature in two ways: (i) only downscaling, (ii) both downscaling and calibration, with the purpose to reveal the effect of temperature calibration.

We used the term downscaling on the resampling from a low resolution for the ECMWF forecasts to the 1x1km grid used for the SeNorge data, combined with a temperature correction using a temperature lapse rate. This terminology is used by e.g. UK Met office (Sheridan et al 2010, with references therein). Especially for areas with a complex terrain, where the resolution of the NWP poorly resolves the terrain, the correction for the discrepancy between model elevation and terrain are useful. In some literature, the term downscaling includes both bias correction and resampling, (ref Yuan et al 2015), but we did not use this terminology here.

We used the term calibration on the statistical adjustments of bias and dispersion of the ensembles. The aim of calibration is to make the forecasts reliable in a statistical sense, i.e. 90% of the observations are within a 90% uncertainty interval. In particular, in the meteorological forecasting literature, calibration has this specific meaning (e.g. Gneiting, 2006)

We think that to separate the pre-processing into downscaling and calibration is useful, but agree that the term downscaling might have a different signification in parts of the literature. Our terminology is also, to a large degree, in accordance to the descriptions in Li et al (2017). Lie et al (2017) describes the main purposes of post-processing to be the following (1) correct bias and dispersion in the forecasts, (2) to preserve the predictive skill of the forecasts, (3) downscale the forecasts to the scale used in the applications, and (4) to generate ensemble members (…). Further, in the conclusion Li et al (2017) writes that their purpose is "… to calibrate the bias …" In the referred article, we hence see the term calibrate used consistently to describe the statistical properties of both the meteorological and the hydrological ensembles. We further think that using calibration, as part of the pre- and post-processing is a well-established term for the hydrological community using ensemble forecasts. Calibrated ensembles and the calibration methods is more specific than only using only the term pre- or post-processing. Calibration strive for the ensemble to describe the mean and spread of the climatology they should represent.

We have not included any description of the dynamical downscaling, as this usually includes a regional climate model with a different approach, and is not the scope for this study.

AC: We added a description to clarify the use of pre-processing, calibration and downscaling. We further omitted the reference to post-processing (**P2L32-P3L2)** since in this study we focus on the calibration and downscaling of the meteorological forecasts, which from a hydrological perspective is pre-processing.

**P3L7-11** "Pre-processing (from a hydrological perspective) refers to all techniques used to change the output from a meteorological model, and includes calibration (described above) and downscaling. Downscaling implies resampling from the original forecast grid size to a grid of higher resolution, and both statistical (e.g. interpolation) and dynamical (e.g. a regional weather forecast model) techniques, can be used (Schaake et. al., 2010). A recent review of post-processing methods are given in Li et al (2017) and the textbook edited by Vannitsem et al (2018)."

As you mention the forecasting period used for the study is only two and a half year long which might influence the results. You state this in the discussion but do not explain why it could be critical. I suggest that you discuss this explicitly. Namely, within such a short period, the interannual variability might not be sufficiently covered. In addition, using forecasts from different model Cycles (38r1 to 41r1) might have an influence of the skill as well because the adaption within a new cycle might enhance or decrease the forecast performance making the comparison between seasons difficult as it might not only originate from the particular season but might be influenced by model versions. I suggest including such limitations in the discussion.

AR: We agree that the inter-annual variability might affect the calibration coefficients, and of course, there are aspects with the different model version that might affect the result. However, the changes applied to the different model-cycles did not remove the biases apparent in temperature forecasts (fig 4).

AC: **P13L17-19** "The use of forecasts from different model cycles might affect the consistency in the forecasts. Moreover, the calibration parameters are sensitive to the representativeness of the calibration period"

To apply Quantile mapping you do need the distribution of the forecast and the distribution of the observations. In section 3.1.2 you state that "MET Norway uses Hirlam temperature forecasts to provide the observational climatology used for parameter estimation". I think here, more information is needed to enable the reader how the calibration is done. Are daily values used for the parameter estimation? Is it empirical or parametric QM used and how are values outside the range treaded (e.g. constant extrapolation)?

Is it a member-by-member approach or are the same parameters used for all members?

AR: MetNorway uses parametric quantile mapping based on the first 24h. When a forecast is outside the observation range, a 1 to 1 extrapolation is used. Therefore, if a forecast is 2°C higher than the highest percentile, then the calibrated forecast is 2°C higher than the same percentile for the reference. The same parameters are applied to all members and lead times.

AC: **P8L13-17** We added "The same coefficients, based on mapping the first 24 hours, were applied to all lead times and members.  For forecasts outside the observation range, a 1:1 extrapolation was used. I.e. if a forecast is 2°C higher than the highest mapped forecasted temperature, then the calibrated forecast is 2°C higher than the highest mapped reference temperature."

One critical point is that the calibration parameters are interfered from the Hirlam but the hydrological model is run with SeNorge observations. Why are not these observations used? The correction will account for the bias between ECMWF and Hirlam but I would expect that biases with SeNorge will at least slightly differ. Why don't you use the observations from SeNorge to get your calibrations?

In the summary it is stated that "The most obvious improvement in the forecasting chain is to use the same temperature information, the SeNorge temperature, for calibrating the temperature forecast that is used for calibrationg the hydrological model, generating …" (P14/L25-27).

But if I understand correctly from the manuscript SeNorge and Hrilam are not the same. I have troubles with this procedure as it is known that different forecast models do have different biases. To bias-correct or calibrate ensembles the observations should be taken into account and not another forecast. In this case the bias between two forecasts will be corrected and not the bias of the forecast with regard to the observations.

AR: You are right to point out these differences in data sets used for calibration of forecasts and the hydrological model. First, as you mention, SeNorge and Hirlam are not the same data. Hirlam is a short-range regional forecast model (4 km horizontal resolution) used in the operational weather forecast for the first 2 days, whereas SeNorge is a dataset where observations are interpolated to a 1 km grid.

In this study, we wanted to use the available operational method from MetNorway, and they used quantile mapping with Hirlam as a reference to calibrate the ECMWF ensemble forecast. Both Hirlam (for the first 2-3 days) and ECMWF (for the following 7-8 days) forecasts are used in the operational weather forecast (yr.no). Using Hirlam data to calibrate ECMWF will improve the transition between the forecasts. Hirlam is available as a sub daily grid and makes it possible for MetNorway to provide different calibration parameters for day and night, whereas SeNorge is only available as a daily grid and would not offer this possibility.

Hirlam has less (smaller) errors than ECMWF in the temperature forecast for Norway (Engdahl et al. 2015), and as we see from e.g. fig 6 and 7 in this manuscript, the calibration reduces the cold biases in the ECMWF forecasts. When we evaluated the hydrological model, the temperature calibration improved in most cases the hydrological forecasts, providing an indirect conformation that the Hirlam temperature is less biased than the ECMWF temperature.

Furthermore, by the many interpolations used, there is a large uncertainty introduced which will lower the trust in the results. Interpolation of ECMEF and Hirlam to derive correction parameters, another interpolation to meet the hydrological model requirements.

AR: We agree that there are uncertainties due to interpolation and downscaling. A temperature calibration that is tailored to the needs for the hydrological modelling would solve this challenge.

AC: P13L5-7 We added "The calibration procedure applied in this study involves many interpolations and downscaling steps that increases the uncertainty in temperature forecasts. We believe that a catchment specific temperature calibration, tailored to the needs for hydrological forecasting, would solve this challenge."

Another point that should be discussed is if seasonal correction parameters are really sufficient or does it introduce artificial jumps between periods. In a climate context, seasonal windows for parameter estimation might be sufficient but in an operational forecasting context a shorter window should be taken into account if possible.

AR: MetNorway provided unique parameters for each month. The parameters are based on a window of three months, which smooths the seasonal patterns. A three month window was chosen to ensure enough data for robust calibration parameters.

In Section 3.2 where the CRPS is introduced you mention different notations (CRPS, Scrp) and same for the CRPSS. I think this is confusing, as later in the text only CRPS is used. I suggest only introducing one of the notations and stick to that.

AR: We agree that this notation might introduce confusion. The reason is the formatting standard of HESS where equations should only contain one capital letter with sub or super script. However, we find it appropriate to use CRPSS in the text since this is the abbreviation used in the community, and in the

equations, we used an alternative notation according to the HESS standard: (S$_{crp}$ and S$_{crps}$ are only used in the equations). This approach is used in many HESS papers.

AC: **P10L5-7** We provided a sentence to clarify sec 3.2: "For readability, the abbreviation Scrp and Scrps used in the equation will be substituted with CRPS and CRPSS in the text hereafter"

**P9L11+14+20-21+27:** We added explanations similar to "CRPS denoted as $S_{CRP}$ in Eq. 1"

Specific comments:

P1

L7-14: You say the flood forecasting system uses deterministic forecasts for temperature and precipitation). But the ECMWF model you reference provides an ensemble of 51 members. Please state how this is used.

AR: The operational system today, uses one deterministic forecast, not the ensemble forecasts. In our setup, the hydrological system is setup to run the 51 ensemble members. We make sure that the same initial states are used for all members. This is explained in details in the main text, and in the abstract we keep the description simple. We think the suggested changes in the following point also covers this point.
AC: **P7L15-16** We added "In the forecasting mode each temperature ensemble member was used as input and run as separate deterministic forecasts."

L11-12: "An alternative approach is to use meteorological and hydrological ensemble forecasts" is somewhat misleading. Either you used ensemble meteorological forecasts in combination with hydrological models to generate ensemble streamflow forecasts or one uses a different methodology to produce hydrological ensembles forecasts. I suggest rewriting the sentence: "An alternative approach is to combine meteorological ensemble forecast with hydrological models to quantify the uncertainty in the forecasted streamflow".

AR: You are right. We apply the suggested rewriting.
AC: **P1L9-14** Rewritten "In this study, we used meteorological ensemble forecast as input to hydrological models to quantify the uncertainty in forecasted streamflow, with a particular focus on the effect of temperature forecast calibration on the streamflow ensemble forecast skill."

L14: "for an accurate forecasting of ", or "to accurately forecast streamflows"
L15: Ensemble forecast of temperature from the ECMWF "
L16: "to improve the skill and reduce biases"

AR: Thank you. We include the suggestion L14, L15, and L16
AC: **P1L15+17+18** Changed accordingly

L18: why do you mention precipitation here? If it is not used for the calibration I would avoid it here.

Author Response to RC#4

AR: We mention precipitation since the "observed" precipitation and temperature was used to calculate the initial states of the hydrological model until the forecast issue day. We will consider omitting the sentence about SeNorge in the abstract. Ref RC#3, and discussion on abstract.
AC:  **P1L20-22** We omitted "Estimated observed daily temperature and precipitation were obtained from the SeNorge-dataset, which is station data interpolated to a 1×1 km2 grid covering all of Norway."

L20: was used to calculate the streamflow

AR: Thank you. We include the suggestion
AC:  **P1L23**  Included

P2

L1: Floods can damage… and can have a high …

L5: componentS

AR: Thank you.
AC: **P2L5** Changed " Floods can severely …"  + **P2L8**

L9: The reference "Müller et al." is missing in reference list

L14: Both reference "Langsrud 1998 a and b" are missing.

AR: Thank you.
AC: Uppdated in the **Reference list**

L16: as a means to account for uncertainty in the forcing.

AR: Thank you.
AC: **P2L22** Corrected

L21: The Reference Cloke & Pappenberger, 2009 and Wetterhall et al., 2013 are missing

L25: the ensembles can be calibrated

L26: Hamill and Colucci, 1997 and Buizza et al, 2005 are both missing

L29: Gneiting et al. 2005 is missing, Wilks and Hamill 2007 is missing, Raftery et al. 2005 is missing

L30: Evens 2003 is missing

L31: Gneiting et al. 2005 is missing, Wilks and Hamill 2007 is missing. The order of the references is different compared to L29. Bremnes, 2007 is missing.

AR: All references are now included.
AC: We updated the **Reference list** and added "Wang and Bishop"

L31-32: This sentence is very general, it is arbitrary clear that different correction methods do correct the biases differently. I suggest either being more specific about single methods, or to summaries different methods to provide a better overview for the reader instead of listing available techniques.

Author Response to RC#4

Maybe cite some standard books for statistical bias correction and downscaling (Wilks, 2011) and for forecast verification (Jolliffe & Stephenson, 2011).

AR: We will cite some standard books and papers that provides reviews of forecast calibration methods.
AC: **P3L10-11** We added the following sentence at the end of the paragraph: "A recent review of calibration methods are given in Li et al (2017) and the text book edited by Vanniitsem et al (2018)

- Vannitsem,S. Daniel S. Wilks, Jakob W. Messner, Editor(s): (2018) Statistical Postprocessing of Ensemble Forecasts, Elsevier, ISBN 9780128123720, doi: 10.1016/B978-0-12-812372-0.09988-X.

- Li, W., Duan, Q., Miao, C., Ye, A., Gong, W., & Di, Z. (2017). A review on statistical postprocessing methods for hydrometeorological ensemble forecasting. Wiley Interdisciplinary Reviews: Water, 4(December), e1246.  https://doi.org/10.1002/wat2.1246

P3

L1: snow cover without "–"

AR: Thank you.
AC: **P3L12** Changed

L2-4: This sentence is unclear to me. Can you elaborate what you mean?

AR: We mean that an improvement in temperature forecast will not necessarily translate directly into an improvement of streamflow forecast. If temperatures are well below zero, an improvement in temperature forecasts has no effect on the streamflow forecasts, whereas for temperatures around zero degrees, the streamflow is very sensitive to temperature, in particular when it might turn on or of rain and/or snow melt.

AC: **P3L13-18** Rewritten "The sensitivity of daily streamflow to temperature is non-linear since streamflow depends on temperature thresholds for rain/snow partitioning and for snow melt/freeze processes. The latter depends on the state of the system, i.e. snow is needed to generate snowmelt. For temperatures well below 0$^{\circ}$C, the streamflow is not sensitive to temperature, whereas for temperatures around 0$^{\circ}$C relatively small changes in temperature might control if the precipitation falls as rain or snow, and consequently, whether streamflow is generated or not."

L5: Gragne, 2015 . missing reference

AR: We will not use this reference in the modified manuscript
AC: The reference will not be used.

L7-8: Forecasting, downscaling and interpolation are three completely different things and the challenge is connected to much more than laps rate. For interpolation and downscaling a large part can be attributed to temperature height correction which depend to a large degree to laps rates. But forecasting of temperature is far more complex and related to chaos theory.

Rephrase please.

AR: You are quite right. We should not have included forecasting in this sentence. We are addressing the downscaling and interpolation of forecasts.

Author Response to RC#4

**AC: P3L21** We removed "forecasting" from the sentence.

L9: Again, missing references: Aguado and Burt, 2010; Pagès and Miro, 2010, Peter et al., 2010.

AR: Thanks. We see that in the case of Peter, this is the first name, it should have been Sheridan et al.
**AC: P3L23** Updated references in text and in **Reference list**.

L13: Alpine (capital A) as the study looks at catchments in the Alps.

L15: ", found only modest….",

AR: Thank you.
**AC: P3L27** Corrected

L17: I think the effect is not marginal, as you later on show with your results.

AR: We used marginal to separate the effect of temperature from that of precipitation. We will change the sentence to 'the isolated effect of…'
**AC: P3L31** Changed to "isolated"

L26: do you mean from both, the hydrological and the meteorological perspective?

AR: Yes, we do. This will be clarified in the manuscript.
**AC: P4L6** Changed "Are there spatial patterns in the temperature and streamflow ensemble forecast skill and if so, can these be related to catchment characteristics?"

L27: from the ECMWF, in addition I would mention the lead time here but maybe not the MET Norway pre-processing setup as you use the QM to pre-process the forecasts which is, if I understood correctly, not yet part of the pre-processing setup at MET Norway.

AR: The information in line 27 is correct. The QM was (new techniques has been implemented recently) a part the operational pre-processing chain at MET Norway and used on the forecast published at yr.no. We chose to not mention lead time here since the choice to focus on lead time 5 days was based on preliminary results.
**AC: P8L4-5** In section 3.1.2 we add one sentence to clarify: "This grid calibration was used in the operational post-processing chain for meteorological forecast including the forecasts published on yr.no."

L28: Are the retrospective forecasts operational forecast for the period within 2013-2015? This could be misleading for readers or misinterpreted as reforecasts (or hindcasts) which are forecasts for the same day as the operational forecast but for the past 20 years using re-analyses for the initialization. Maybe rephrase to avoid any misinterpretation.

AR: We chose retrospective to underline that we used the operational forecasts in retrospect. Nevertheless, we understand that this can be misinterpreted. We will rephrase the sentence.
**AC: P4L9-10** Rephrased "Three years of operational ECMWF forecasts from 2013-2015 were used to re-generate streamflow forecasts, and the skill of temperature and streamflow forecasts were systematically evaluated for these catchments."

Author Response to RC#4

L30: again, I think marginal is the wrong word, if the effect is assumed to be marginal, why should you analyze it in such detail.

AR: OK
AC: **P4L11** Changed to "isolated"

L31: Not clear to me. Do you mean that the observed precipitation is used to drive the hydrological model? Specify that to make it clearer.

AR: Yes. The observed precipitation is used to drive the hydrological model.We will rephrase to make clearer
AC: **P4L11-13** Rewritten "To investigate the isolated effect of the temperature ensembles on the streamflow forecasts, the observed SeNorge precipitation (Tveito et al., 2005) was used instead of the precipitation ensemble forecasts when we re-generated streamflow forecasts, to run the hydrological model."

L33-P4L2: Maybe combine this with the preceding paragraph. This would make it less generic.

AR: We will join the two paragraphs as suggested.
AC: **P4L14** Combined

P4

L5: spatial variations

AR: Thank you.
AC: **P4L20** Changed.

L6: rather high then steep?

AR: The Mountains are both high and steep. However, we think that steep is the most important description of the high elevation gradients in the area.
AC: We made no changes in the manuscript.

L9: delete "flows"

AR: Thank you.
AC: **P4L24** Deleted.

L18: the smallest catchment has an area of only 3 km^2? Or is it a typo?

AR: This is not a typo. There are several small catchments in our dataset, but only one of this size.
AC: There will be no changes in the manuscript

L21: what are the selection criteria for "data of sufficient quality"
AR: This was inaccurate description since the catchments disregarded from the study was due to different reasons, both data retrieving and technical problems. For three catchments, we had problems

running the model with the reference data, one catchments there was an issue with the elevation correction, and for two catchments, there were technical problems during the regional analysis. We have a large dataset, so the exclusion of the six catchments will not change our conclusions.
AC: **P5L5-6** Rewritten "Of the 145 flood forecasting catchments, 139 were chosen as the basis for the study (Fig. 1)."

L27: "og" seems to be Norwegian

AR: Thank you.
AC: **P5L11** Corrected.

L31: snowmelt driven flood event

AR: Thank you.
AC: **P5L16** Corrected as suggested.

P5

L5: write "available at SeNorge.no"

AR: Thank you.
AC: **P5L21** Corrected.
L7: Mention what kind of interpolation is used (bilinear, kriging, …)

AR: The SeNorge temperature is interpolated using kriging on de-trended temperature using standard temperature lapse rates.
AC: **P5L21-22** Rewritten "For this version, gridded temperature is calculated by kriging, where both the elevation and location of temperature stations are accounted for."

Section 2.2.1:

Mention here that you use the precipitation data from this data set as a substitute of the precipitation forecasts (if this is the case).

AR: Thank you. That is a good suggestion.
AC: **P5L28-29** We added a sentence at the end of the paragraph "The SeNorge precipitation substitutes the precipitation forecasts in the ensemble forecasting chain, and hence the isolated effect of temperature calibration on streamflow forecastswas obtained"

L15: constitutes as the basis

AR: We prefer to keep the sentence as it is.
AC: No changes will be introduced in the manuscript,

L20: explain what PEST is.

AR: We will modify the sentence and explain what PEST is.
AC: **P6L9-10** Modified "... which has been calibrated using the PEST software for parameter estimation (Doherty, 2015), …"

Author Response to RC#4

L21: Abbreviation NS (for Nash-Sutcliffe) not introduced before.

AR: Thank you. We will be corrected in the manuscript.
AC: **P6L11-12** Changed to "Nash-Sutcliffe"

Section 2.2.2:

Is the calibration done for each catchment separately? Do the given values for the NS coefficient represent the mean for all catchments? Is this good? Please state how these values translate into performance compared with other hydrological models.

AR: The calibration is done for each catchment separately. The mean is presented to give an impression of the performance, and of course, there is a great difference in  the NS-score between the catchments. We think that NS between 0.73 and 0.77 is ok. Within the range of NS-scores there are of course catchments where the models performs less optimal. Other models applied to the same catchments has a very similar performance, indicating that the quality of data (precipiptation, temperature and streamflow) is an important contribution to model uncertainty. Since we in this paper use the model streamflow in stead of the observed streamflows for evaluation of forecast, we think it is not necessary to provide more details on the calibration of the hydrological model   .
AC: No changes introduced in the manuscript.

L22: Missing Reference Gusong (2013), In reference list only Gusong 2016 is listed

AR: Thank you.
AC: **P6L12** Corrected to "Gusong (2016)".

2.2.3

To make this more coherent I suggest renaming this section into "Reference observations" (or similar) and in the latter part of the study refer to reference observations as well. Otherwise it is difficult to distinguish between the model stream flow and forecasted streamflow. E.g. on P6 L13 you write reference model run, I assume this is the same as model streamflow? This is somewhat confusing if you state it twice in 2 different paragraphs.

AR: Thank you. We will change "model streamflow" to "reference streamflow" in the section title and in the text.
AC: **P6L13+14+17+19** We changed to "reference streamflow", throughout the text

P6

L6: write the lead time as well in days 246 hrs (i.e. 10 days). Why is it 246 hrs and not 240?

AR: We used lead time 246 hours since we have used the forecast issued at 00:00 aggregated to daily values for the time period 06-06. We can change this to days.
AC: **P6L29-30** We added one sentence to clarify this "In this study, we used the forecasts issued at 00:00 and aggregated daily values for the meteorological 24-hour period defined as 06:00-06:00 to provide forecasts for lead times up to nine days."

L7: The Reference "ECMWF (2018a)" does only provide the documentation and support page of the ECMWF. The Specific documentations can be downloaded. The scientific basis of the ENS system has

been discussed in multiple publications and it might be worth to reference some of them and point to this documentation for specific points only.

AR: We would like to keep the sentence and reference as it is, since this is provides a detailed overview of the model cycles. We provide an additional sentence, including references, to the description of ECMWF.

AC: **P6L24-27** We moved and rephrased "In short, 50 ensemble members of ENS are generated by adding small perturbations to the forecast initial conditions and model physics schemes, subsequently running the model with different perturbed conditions. The ensemble represent the temperature forecast uncertainty. A more detailed description of the ECMWF ENS system is provided in e.g. Buizza et al. (1999) and Persson (2015)."

L8: "the ensemble members of ENS are…"

L9: "with different perturbed conditions to represent the …"

AR: Thank you.

AC: The sentences are rephrased, see **P6L24-27**

3.1

See comment to 2.2.3. I don't get the difference between model streamflow and reference model run. If I understand correctly these are the same. If so, only describe it in one section. I think here it would be suitable. Reference run = model streamflow, use the same terminology if it is the same.

AR: We will change 'model streamflow' to 'reference streamflow', but be prefer to keep section 2.2.3- since we in section 2 describes the data and models, whereas in section 3 we describe how we used the data.

AC: We changed to "reference streamflow" throughout the text.

Are the ENS forecast temporally aggregated as well?

AR: The ENS are also temporally aggregated. Ref p7 l1-2 (3.1.1) and l15-16 (3.1.2), and fig 2.

AC: **P7L13-15** We added "All temperature forecasts were aggregated to daily time steps since the operational HBV model runs on a daily time step and the SeNorge data used as a reference provides only daily values."

L25: replace "include" with "referred to as"

AR: Thank you.

AC: **P7L22** Changed "refers to"

L27: Use the same units for both grids. ° or km^2. Best would be use both units for both grids, one of them in brackets.

AR: We think it is more accurate to use use degress for the ECMWF grid, but we will add a parenthesis with the grid resolution in km. Hence, we use degrees and km for EC, only km for SeNorge

AC: **P7L25** We changed " ...  resolution of 0.25° ( ~ 30km)"

Author Response to RC#4

3.1.1

What is the rationale behind the choice of using a nearest neighbour technique?

AR: We tested also other techniques, e.g. bilinear interpolation, which has a higher computational demand and creates larger output files, than the nearest neighbor interpolation. Since the quality of the forecasts temperature was almost similar, the reduced computing time and smaller storage requirements made the nearest neighbor method more useful.
AC: We introduced no changes in the manuscript.

P7

L4: Bremnes 2007, 2004 are missing in the reference list.

AR: Thank you.
AC: We updated the **Reference list**.

L8: Can you give a reference for the sentence "gives a higher skill and are less biased"

AR: The reference is Engdahl et al 2015
AC: **P8L8** We included (Engdahl et al, 2015) in the text and to the **Reference list.**

L20: Ensemble forecast verification does not only focus on reliability and sharpness. Therefore, different measures need to be taken into account (as well biases are important).

AR: In this sentence we refer to a specific paper (Gneiting et al., 2007) where the reliability and sharpness is used for evaluation of forecasts. We also think the bias is a part of the evaluation according to reliability. If the forecast is biased it will not be reliable.  In the rank-histogram decomposition slope will identify bias in the forecasts.
AC: We introduced no changes in the manuscript.

L30: "lowest and highest forecasted value" does it mean the minimum and maximum? Why not the 10th and 90th percentile and the interquartile range. I think this gives a better estimate of the sharpness of the forecast as it does not only account for the most extreme members.

AR: we agree that specific interquartile range might be a more robust measure for sharpness, and used the range between the $5^{th}$ and the $95^{th}$ percentile to evaluate the spread.
AC: **P9L5-8** We changed to "In this study, the temperature sharpness was assessed by first estimating the range between the $5_{th}$ and the $95_{th}$ 5 percentile of the ordered ensemble forecasts for all issue dates, lead times and catchments. For streamflow, we estimated a relative sharpness by dividing the $5_{th}$ to $95_{th}$ percentile range by the ensemble mean. Thereafter, sharpness was determined for each catchment and lead time as the average range of all issue dates."

P8

Author Response to RC#4

L12: I would rephrase the sentence. "which a skilful forecast should outperform" and write it in a single sentence.

AR: We think the sentence is fine as it is.
AC: We introduced no changes in the manuscript.

L18: negative values mean (without s)

AR: Thank you.
AC: **P10L1** Corrected

L19: "which perform similar to the reference forecast (climatology in this case)"

AR: Thank you.
AC: **P10L2** Changed to "implies that it performs similar to the benchmark (climatology in this case)".

L20: Do you use here the mean of the daily CRPS? (CRPS with overbar?)

AR: Yes, in this case it refers to calculating the average ($\overline{CRPS}$) over all daily CRPS (without an overbar), for the months in question.
AC: No changes introduced in the manuscript.

L25-26: This sentence seems to be wrong.

AR: We will reformulate the sentence.
AC: **P10L10-11** We rephrased "Finally, we used linear regression to identify relationships between catchment characteristics (elevation difference and catchment area) and the skill score ($T_{cal}$ and $Q_{cal}$ CRPSS)"

L27: Usually seasons are aggregate in winter = December-February (DJF), spring = mar-may (MAM) and so on. Can you explain your motivation to choose this definition of seasonal aggregation?

AR: You are right about the usual definition of seasons. We used a different definition since we wanted to isolate a snowmelt season, that for most catchment most catchments is in the period April to June. . We think this better seasonal description for streamflow in Norway.

AC: **P10L14-15** We added one sentence "This definition of season is used to better capture a snowmelt season that for most Norwegian catchments is in the period April to June."

P9

L8: as shown in figure…

L9: no comma after "convexity"

AR: Thank you.

AC: **P10L29-30** Corrected as suggested

The description about the slope and complexity is hard to follow. Could you give an example what the values really tell, e.g. how does a rank histogram look like with a complexity of 2000? I think rank histograms are very useful to be used for visually interpretation and the complexity and slope somehow lead to a reduction of the usefulness of the rank histogram at least to people not familiar with these parameters.

AR: We used the convexity and slope since then it is much easier to provide aggregated information of forecast performance. In our results, we do not focus on the values in themselves; the change of the values is the important information. We find that Jolliffe and Primo, 2007 provide detailed information.

AC: **P8L28-31** We rephrased and elaborated more on the rank-histogram evaluation "A bias in the ensemble forecast is recognized as a slope in the rank-histogram, where a negative slope indicates too warm temperature forecasts, and positive slope too cold forecasts. A U-shape indicates that the ensemble forecast is under-dispersed, whereas a convex shape indicates over-dispersed (Hamill, 2001)."

L15: I recommend repeating what TO and Tens is to enhance the flow in the text.

L27: Same here, mention the abbreviation in brackets in the text to help the reader.

AR: For both comments above, we will repeat the meaning of abbreviations in the beginning of each section.

AC: **P11L4-5+17-18, P12L29-30, P13L9, P14L8** Changed as suggested for all sections.

L29: "influenceS" ; Do you mean in streamflow skill or CRPSS?

AR: All skill is measured by CRPSS.
AC: **P11L19-20** We rephrased " … Fig 5 shows how the change in temperature CRPSS affects the change in streamflow CRPSS for spring and autumn."

P10

L4: Do you know why there is no improvement during summer by using calibrated temperatures? Is it due to the absence of snow / snow-melt in summer?

AR: There are two reasons for the small changes during summer (i) the skill of uncalibrated temperature forecasts are higher in summer and (ii) there is less or no snow in summer, and that will reduce the streamflow sensitivity to temperature. Ref comments RC#3 and editor, we omitted the results for summer and winter.
AC: **P10L16-19** We added "Summer (July to September) was excluded due to the relatively small changes in CRPSS explained by (i) the skill of uncalibrated temperature forecasts are higher and the potential for improvement is lower, and (ii) there is less or no snow in summer, resulting in a reduced streamflow sensitivity to temperature. Winter (January to March) was excluded since it performs similarly as the autumn."

Author Response to RC#4

L5: You often mention the Figure number in the last part of the sentence. I personally would prefer this information first what makes it easier to follow the text and figures at once; It reveals

AR: We try to vary the placing of the figure number in a sentence and it is a question of style / preference.
AC: We made no specific changes in the manuscript.

L20: What is the significance level you used? I would mention this in the text.

AR: For the slope of the regression lines being different from zero we used a significance level of p-values < 0,05. This information is available in the caption text for fig. 9. We will include this in the text.
AC: **P12L12** We included "By indicating the significance and sign of the relationships, significant relationships were found for 12 out of 40 regression equations (5% significance level)."

4.3

What are the criteria you used to choose this flood event in May 2016? Mention the motivation for this specific event.

AR: We wanted to present a snowmelt flood event during spring and the selected event in May 2013 in Bulken was a snowmelt flood.
AC: **P12L20** Changed to "2013".

L10: If possible embed this in the floating text and see separate comment to the figure.

AR: We are not certain to which line this comment refer.
AC: **P12L19** We added "target days", to ensure a consistency to figure 10.

P11:

L1: to make it clear I would add: "…increases with lead time (form upper to lower panel)." Linked to my comment on the caption in Figure 10 that it could be misinterpreted as a continuous forecast starting at may 16th.

AR: Thank you. We will modify the sentence as suggested.
AC: **P12L26** Changed and we included the reordering of the panels (ref RC#3) "… with lead time (from lower to upper panel), … "

L4: "The box plots … show" (show without s)

AR: Thank you.
AC: **P12L29** Corrected.

5 Discussion

Here I would again use words instead of the Tens Tcal only: "Both raw (Tens) and calibrated (Tens) temperature forecasts were more skilful with …". I think it makes the text more interesting to read. This could be adapted in different parts of the Manuscript, in the beginning of each section this should be repeated.

AR: We will introduce the abbreviations in the beginning of the sections
AC: **P12L29-30** We changed according to suggestion.

L5-9: "Overall, the grid calibration of temperature had a positive effect on both …", but the lines before it states "…, resulted in reduced skill". This is somehow contradictive, could you make this clearer?

AR: The last sentence refer to the difference between raw and calibrated ensembles for all lead times, and we see that the grid calibration improves the performance for most scores and lead times. The previous statements are related to the development of performance for increased lead times. In short, the CRPSS is reduced for increased lead time, it is better for calibrated than raw ensembles.
AC: **P13L4-5** We changed the sentence "Overall, the grid calibration of temperature had a positive effect on both temperature and streamflow for most validation scores and lead times."

L18: missing reference Lafon et al. 2013

L20: L24: wrong citation format Ivar Seierstad et al. (2016)

AR: Thank you.
AC: **P13L23+27-28** we updated the citation (Seierstad et al, 2016) and the **Reference list**.

Subtitles for 5.1 and 5.2 should be coherent "calibration for …" or "calibration for the…"

AR: Thank you.
AC: **P13L8, P14L7** Subtitles are corrected

L26: forecasts

AR: Thank you.
AC: **P14L8** Corrected.

P12

L4: "Hence, calculated … " word at wrong place within sentence.

AR: Thank you.
AC: **P14L2**1 corrected to "Hence, estimated streamflow has a high…"

L7 "indicate" delete additional s

AR:  Thank you.
AC: **P14L24** Corrected.

L18: "the bias in Tens is explained by" I think this statement is too strong. It can be an explanation, but I think it cannot be reduced to this single causality, as you state in the next sentence.

AR: Thank you.

Author Response to RC#4

AC: This sentence is removed from sec 5.3, and content rewritten in sec 5.1

L21: "The Tens CRPSS is skilful" forecasts have a positive CRPSS and are skilful. The current formulation is not logical, a CRPSS is not skilful.

AR: We will rephrase to clarify that skillful refers to the forecast.
AC: **P13L29** We changed to "…, CRPSS show that the uncalibrated Tens is skillful for both…"

L28: please state these characteristics very shortly again here.

AR: We will modify as suggested.
AC: **P15L17-18** We changed the text as follows "Only a few significant relationships between the catchment characteristics, e.g. catchment area and elevation gradient, and skill were found"

P13

L1: I don't understand what you mean with "the averaging effect on temperature skill dominates".

If I understand correctly, you could discuss here what the difference would be if you use a spatially distributed hydrological model (e.g. gridded version of the model with high resolution). The effect of temperature downscaling might be higher in this case because you do not average temperature again after the downscaling and the spatial distribution within a catchment would have a much larger effect especially in catchments with high spatial variability of soil properties, altitude and vegetation cover.

AR: What we discuss in this paragraph is the effect of catchment size on the performance of the forecasts. We think that a forecast for small catchments are more sensitive than large catchments to the spatial pattern of forecasted temperature. The reasons are that (i) the smallest catchment are smaller than the grid size of the ECMWF model and (ii) it is more challenging to forecast weather on small spatial scales than large spatial scales.
AC: **P15L21-24** We rephrased "This result is not conclusive, but indicates that (i) the smallest catchment are smaller than the grid size of the ECMWF model and therefore very sensitive to the pre-processing (ii) it is more challenging to forecast weather on small spatial scales than large spatial scales.

L13: "the calibrated temperature reduced the skill of the forecasted streamflow." Please state what skill measure you mean here, did you calculate the CRPSS or bias for that specific event? In the result you only describe the range of the calibrated / uncalibrated ensembles but not a measure of skill.

AR: You are right. In this sentence, the use of skill is misleading. We did not calculate a specific measure of skill, but merely point to fact that compared to the reference streamflow, the calibrated T forecast induce too high streamflow, and the error becomes larger. A better word would might be performance.
AC: **P16L12** Changed to "performance"

L15-17: I think you would like to point out that other errors (in the meteorological dataset and the hydrological model) do influence the results. If so, the sentence should be rephrased. Now the reader

Author Response to RC#4

might think that forecasts are always getting worse if they are calibrated and this would be an argument against your conclusive statement in the summary on Page 14/L19-18.

AR: We agree, and will add a sentence to clarify this. We will also remove streamflow observations from the figure and consequently from the discussion.
AC: **P16L13-15** Rewritten "Deterioration in the forecast performance using calibrated temperature is particular for this event. Other results provided in this study shows clearly that the calibrated temperature ensembles improve the streamflow ensemble forecasts on average."

Figures:

Figure 1: write "grouped" instead of "divided". Something is wrong in the first sentence "this study shown using". Please rephrase.

AR: We will rephrase the caption.
AC: **P24** Rephrased caption "The maps for Norway indicates the 139 catchments used in this study. The left map show the catchment boundaries including the location of four selected catchments. Please note that many catchments are small and difficult to detect. The location of the catchments gauging stations are shown in the right map. Norway was grouped into five regions (N=north, M=mid, W=west, S=south, and E=east), all regions are marked with colors and regional boundaries."

Figure 3:

Avoid overlap of the boxplots to enhance the readability of the plot. There seem to be two line-artefacts on both sides of the figure.

AR: We will have a look at the box-plots, the artefacts in the figures will probably disappear in the finishing stage, as all figures will be provided separately. We used partly overlapping boxes for each lead time to increase the readability of the figure, since it is easy to see to which boxes that belongs to the same lead time. We tried without, but found it then more difficult to read the plot.
AC: No changes introduced.

Figure4:

In the text you write TO and in the Figure it corresponds to Tobs. Similarly, Tens and Tens-range. It might facilitate the text of the abbrevations are more consistent in the text, captions and the figures.

AR: Thank you. This will be corrected a suggested
AC: **P30** We changed the figures, and correct "$T_{obs}$ to $T_o$" in both plots.

Figure 6:

Line artefacts on the left of the figure.

AR: The artefacts in the figures will probably disappear in the finishing stage; all figures will be provided separately.
AC: We will check that the line artefacts are not present in the final manuscript.

Figure 10:

It is hard to see the actual forecast. I suggest removing the background colors for the warning levels and just plot lines instead. The Figure can easily be misinterpreted as the individual plots (e.g. upper panel for lead day 2) look like a continuous forecast. Maybe it would be more suitable to plot boxplots instead.

AR: OK. We will do some changes to this figure. Ref A#3. We prefer- however, to not use box-plots. We think that the use of lines and shaded areas increase the readability of the figures.
AC: **P38** We have changed the figure. The background colors and the streamflow observation are removed.

Captions: Forecast issue date is the date when the forecast was issued, hence the x-axis could be different for each panel in this figure. I recommend adapting the caption to make this clearer, e.g. target day instead of issue date.

AR: .Thank you. We will follow the suggestion.
AC: **P38** Changed to "target day" in the caption Fig 10, and in the text **P12L19**

"model streamflow with SeNorge observations" this is QO. I would write it in brackets as you do for Qcal.

AR: Thank you, we will follow the suggestion.
AC: **P38** Rewritten caption "Forecasted streamflow for the Bulken catchment fort lead times 9, 5 and 2 days. Forecast target dates on the x-axis, and streamflow ($m^3s^{-1}$) on y-axis. Reference streamflow with SeNorge observations $Q_o$ (black solid line) , ensemble mean uncalibrated temperature $Q_{ens}$ (blue line), ensemble mean calibrated $Q_{cal}$ (blue dotted line), ensemble range $Q_{ens}$ (light violet area)  and ensemble range $Q_{cal}$ (light blue area). The grey dotted lines indicate the thresholds for mean annual, 5-year and 50-year floods."

[revised manuscript text omitted]

---

## Author Response (AR3)

Dear Dr. Jan Seibert.

We thank you for your latest comments, and hereby present the corrections to the manuscript.
To the Editor comments (grey text) in this first section, we provide author corrections (AC, blue text). The author corrections show the page and line to which the implemented corrections and changes are found in the provided track changes version of the revised manuscript. Hence, **P6L3-5** indicates changes on page 6 and lines 3 to 5.

Clarify how the mean NSE, p5l26 has been computed. For how many catchments? Related to this, it would also be useful to mention that there probably is quite some parameter uncertainty and that, thus, using multiple parameter sets might be useful, even if this has not been done here for computational reasons.

AC: To clarify the computation of mean NS we added the following:

P5L26-27: " … , for each catchment individually, … "

P5L26: "… for the 139 catchments..."

P5L30-31: " We used one optimal parameter set for each catchment and ignored the therefore the uncertainty arising from parameter estimation and the hydrological model."

P15L32-P16L3: "In this study, we have investigated the isolated effect of uncertainties in temperature forecasts. For a more complete assessment of forecast uncertainties, error in initial conditions, hydrological model parameters and structure need to be accounted for. In particular, we might expect a strong interaction between uncertainties in temperature forecasts and model parameters controlling snow accumulation and snow melt processes."

Check your references, e.g., bergström instead of Bergstrom.

AC: We changed to Bergström, in text P2L8/P5L21 and in reference

Please check the language once more. Not being a native speaker myself, I will not try to give advice, but some places the English could be improved, I think.

AC: We applied the following corrections after consulting a native English-speaking colleague:

P1L3 Changed affiliation from "Trine J. Hegdahl[1,2]" to "Trine J. Hegdahl[1]"

P1L9 and P1L13 - changed "forecast" to "forecasts"

P1L17 - added a comma after HBV model

P1L23 - changed "high improvement" to "large improvement"

P2L26 - changed "ensembles are ensemble…" to "ensembles include ensemble…"

P2L29/30 - changed "sensitivity to length of training data and ensemble size, and how spread and bias are corrected" to "sensitivity to the length of the training data and ensemble size, and how the spread and bias are corrected"

P3L5 and L6 - changed "degree" to "degrees"

P3L11 - changed "are challenging"... to "are both challenging…"

P3L21 - changed "is not yet"… to "has not yet been"

P3L27 - changed "question" to "questions"

P3L33 - changed "the MET Norway" to "MET" Norway

P4L10 - changed "overview over…" to "overview of…"

P5L19 - changed "constitutes as the" to "constitutes the"

P5L24 - changed "laps" to "lapse"

P6L13 - changed "represent" to "represents"

P6L15/18 – changed the * for l* is superscript in L15 but not L18

P6L23/24 - added comma before and after "aggregated to each catchment c"

P6L29 - changed "provides" to "provide"

P7L1 - changed "as separate" to "as a separate"

P7L11 changed "elevation difference" to "elevation differences"

P7L18 changed "forecast including" to "forecasts including"

P7L19 changed "temperature forecast" and to "temperature forecasts"

P7L21 - changed "Hirlam gives a higher skill and are less biased than the ENS" to "Hirlam gives higher skill and is less biased than ENS"

P7L22/23 - changed "re-forecast" to "reforecasts"

P7L24 - changed "as the ENS" to "as ENS"

P7L25 - changed "May analysis" to "the May analysis"

P7L31 - changed "the ENS" to "ENS"

P8L13 - changed "positive" to "a positive"

P8L14 - changed "over-dispersed" to "over-dispersion"

P9L10 - changed "forecasts" to "forecast"

P9L11 - changed "similar" to "similarly"

P9L15 – removed comma after "equation

P9L21 - added a comma between seasons and spring

P9L22 - changed "snowmelt season that for most Norwegian catchments is in the period April to June" to "snowmelt season, which for most Norwegian catchments is the period April to June"

P9L24 - changed "are higher" to "is higher"

P9L26 - changed "similarly as the autumn" to "similarly to autumn"

P10L6 and 7 - changed "gets" to "get"

P10L24 - changed "shows" to "show"

P10L27 - changed "Catchment CRPSS" to "Catchment CRPSSs"

P10L28 - changed "reveal" to "reveals"

P10L29 - changed "performs" to "perform"

P11L16/17 - changed "For none of the regions the correlation changes sign between the seasons." to "The correlation does not change sign between the seasons for any of the regions."

P11L26 - changed "for the lead time 2 days, to a very wide range for lead time 9 days." to "for a lead time of 2 days, to a very wide range for a lead time of 9 days."

P12L2/3 - changed "longer lead time" to "longer lead times"

P12L16/17 - changed "this can be an explanation for too high correction" to "which can be an explanation for too high a correction"

P12L25 - changed "forecast" to "forecasts"

P12L29 - changed "show" to "shows"

P13L3 - changed "was seen" to "were seen"

P13L5 - changed "forecasts" to "forecast"

P13L12 - changed "Additional important" to "Of additional importance"

P13L24 - changed "streamflow forecast" to "streamflow forecasts"
P14L19 - changed "shows" to "show"
P14L21/22 - changed "For lead time 2 days" to "For a lead time of 2 days"
P15L11 - changed "catchment" to "catchments"
P15L24 - changed "depending" to "dependent"
P16L7 – Deleted "The conclusions herein are based on a large and relative representative data set from Norway, but w"
P16L10 - changed "in in" to "in"
P16L9 - changed "need" to "needs"
P16 section 7: changed "Raw meteorological data must be required" to "Raw meteorological data can be obtained"
P16 section 8: changed all "contributed in" to "contributed to"
Table 1 caption - changed "Blue color indicates" to "Blue indicates"
Figure 1 caption L1: changed "indicates" to "indicate"
Figure 1 caption L3: changed "the right map" to "the map on the right"
Figure 3 caption L3: changed "The forth column show" to "The fourth column shows"
Figure 8 caption L2: changed "difference" to "differences"
Figure 10 caption L2: changed "m3s-1" to "$m^3 \, s^{-1}$ "

[revised manuscript text omitted]